# Toxic PARP trapping upon cAMP-induced DNA damage reinstates the efficacy of endocrine therapy and CDK4/6 inhibitors in treatment-refractory ER+ breast cancer

Ozge Saatci [1,2], Metin Cetin [1,2], Meral Uner [3], Unal Metin Tokat [4], Ioulia Chatzistamou [5], Pelin Gulizar Ersan [2], Elodie Montaudon [6], Aytekin Akyol [3], Sercan Aksoy [7], Aysegul Uner [3], Elisabetta Marangoni [6], Mathew Sajish [2] & Ozgur Sahin [1,2] ✉

Resistance to endocrine therapy and CDK4/6 inhibitors, the standard of care (SOC) in estrogen receptor-positive (ER+) breast cancer, greatly reduces patient survival. Therefore, elucidating the mechanisms of sensitivity and resistance to SOC therapy and identifying actionable targets are urgently needed. Here, we show that SOC therapy causes DNA damage and toxic PARP1 trapping upon generation of a functional BRCAness (i.e., *BRCA1/2* deficiency) phenotype, leading to increased histone parylation and reduced H3K9 acetylation, resulting in transcriptional blockage and cell death. Mechanistically, SOC therapy downregulates phosphodiesterase 4D (PDE4D), a novel ER target gene in a feedforward loop with ER, resulting in increased cAMP, PKA-dependent phosphorylation of mitochondrial COXIV-I, ROS generation and DNA damage. However, during SOC resistance, an ER-to-EGFR switch induces PDE4D overexpression via c-Jun. Notably, combining SOC with inhibitors of PDE4D, EGFR or PARP1 overcomes SOC resistance irrespective of the *BRCA1/2* status, providing actionable targets for restoring SOC efficacy.

Breast cancer is the most common cancer and the leading cause of cancer-related deaths among women worldwide[1]. ER-positive (ER+) breast cancer has the highest incidence rate and accounts for around 75% of all cases[2]. Endocrine therapies, including the selective ER modulator (SERM), tamoxifen and the selective ER degrader (SERD), fulvestrant have been the mainstay therapy for the treatment of both early and late-stage ER+ breast cancer for decades. Although most ER+ breast cancer patients initially respond well to endocrine therapy, resistance is common and significantly reduces patient survival[3]. The addition of CDK4/6 inhibitors to endocrine therapy in ER+/HER2- early or metastatic patients led to significant improvements in clinical outcome, and they are now considered one of the standard of care (SOC) therapies. However, a significant proportion of patients still suffer from disease relapse upon prolonged use of endocrine therapy in combination with CDK4/6 inhibitors, representing a major clinical challenge that reduces the long-term benefit and patient survival[4,5].

[1]Department of Biochemistry and Molecular Biology, Hollings Cancer Center, Medical University of South Carolina, Charleston, SC 29425, USA. [2]Department of Drug Discovery and Biomedical Sciences, University of South Carolina, Columbia, SC 29208, USA. [3]Department of Pathology, Faculty of Medicine, Hacettepe University, 06100 Ankara, Turkey. [4]Department of Molecular Biology and Genetics, Bilkent University, Ankara 06800, Turkey. [5]Department of Pathology, Microbiology & Immunology, University of South Carolina, Columbia, SC 29208, USA. [6]Translational Research Department, Institut Curie, PSL Research University, Paris 75005, France. [7]Department of Medical Oncology, Hacettepe University Cancer Institute, 06100 Ankara, Turkey. ✉e-mail: sahin@musc.edu

Multifaceted mechanisms of resistance have been associated with relapse under endocrine therapy[6] some of which may be reversed by CDK4/6 inhibitors, such as activated cell cycle progression by cyclin D1 or CDK4 overexpression[7]. However, not all endocrine-resistant patients respond well to CDK4/6 inhibitors, probably due to alterations that commonly confer resistance to endocrine and CDK4/6 inhibitor therapies, such as RB dysfunction, activation of FGFR/ERBB receptors or activating *AKT* or *RAS* mutations[8]. Despite being the standard of care in ER+ breast cancer, the molecular underpinnings of the benefit achieved with the use of endocrine therapies and CDK4/6 inhibitors as well as the mediators of resistance to these agents remain largely unknown.

Defects in DNA damage response (DDR) pathways are frequently acquired by cancer cells during tumor evolution[9]. The presence of mutations and/or copy number alterations in specific DNA repair genes, such as *BRCA1/2* often create a synthetic lethality, rendering such tumors susceptible to cell death when treated with DNA damage inducers or inhibitors of DNA repair proteins, such as PARP1[10]. In addition, some tumors may exhibit BRCAness phenotype, i.e., acquisition of molecular features of *BRCA1/2*-mutant tumors and display increased sensitivity to DNA damage-induced cell death. One of the major mechanisms of cell death under excessive DNA damage is transcriptional blockage[11]. Transcriptional recovery may also be impaired by prolonged/aberrant accumulation of DNA repair proteins, such as PARP1 at the unrepaired damage site, known as PARP1 trapping, causing inhibition of global transcription[12]. In the clinical landscape, agents that can activate DDR have proven effective owing to the variety of responses that they elicit, such as inhibition of cancer cell proliferation, transcription inhibition and induction of cell death. Regarding endocrine and CDK4/6 inhibitor therapies, little is known if they can commonly alter the landscape of DNA repair proteins and activate DDR, and if so, what could be the upstream and downstream molecular mechanisms controlling these events in sensitive and resistant tumors, and if these controllers may be targeted in the resistant tumors to reinstate SOC-induced efficacy irrespective of their *BRCA* status.

Here, we show, for the first time, that SOC therapies used in ER+ breast cancer (here: tamoxifen, fulvestrant and palbociclib) induce DNA damage along with inhibition of homologous recombination (HR) and toxic PARP1 trapping. Therefore, SOC therapies generate a functional BRCAness phenotype in *BRCA1/2*-wt cells by downregulating key DNA repair proteins that ultimately leads to inhibition of global transcription and cell death in a PARP1-dependent manner. Mechanistically, we found that SOC therapies induce the accumulation of cAMP through PDE4D depletion, generating mitochondrial reactive oxygen species (ROS) and DNA damage. Importantly, we identified PDE4D as a novel ER target gene that in turn stimulates ER activity in a feedforward loop in endocrine-responsive models and regulates *BRCA1/2* expression. Consistently, targeting PDE4D with a clinically tested inhibitor alone in drug-responsive settings mimics SOC-induced effects. We show that EGFR-mediated c-Jun activation facilitates the overexpression of PDE4D during SOC resistance, and upregulation of PDE4D confers SOC resistance. Notably, combining SOC with inhibitors of PDE4D, EGFR or PARP restores G1 arrest and apoptosis irrespective of the *BRCA1/2* status, leading to drug sensitization in vitro and in vivo.

## Results

### SOC therapy commonly induces DNA damage, BRCAness and toxic PARP1 trapping, leading to transcriptional blockage and growth inhibition in ER+ breast cancer

To determine the common gene expression changes governed by the SOC therapies used in ER+ breast cancer, i.e., tamoxifen, fulvestrant and palbociclib, which will further uncover signatures of sensitivity and resistance, we analyzed the Connectivity Map database[13]. We obtained a list of genes that are commonly up or downregulated upon

treatment of the ER+ breast cancer cell line, MCF-7 with SOC (Fig. 1a), which we named the 'SOC sensitivity' signature (Supplementary Data 1). Intriguingly, we observed that genes among the SOC sensitivity signature were similar to those regulated by doxorubicin and etoposide, well-known topoisomerase II inhibitors that induce DNA damage. In line with this, pathway enrichment analysis within the 'SOC sensitivity' signature revealed significant enrichment of several DNA damage/repair, cell cycle-related and apoptosis pathways (Fig. 1b). Notably, we observed strong downregulation of *BRCA1* (z = −3.5, Supplementary Data 1), indicating the potential generation of a functional 'BRCAness' phenotype upon SOC treatment in the *BRCA1/2*-wt ER+ MCF-7 breast cancer cells (Fig. 1b). We showed that treatment with tamoxifen, fulvestrant and palbociclib increased the percentage of G1-arrested (Supplementary Fig. 1a) and apoptotic (Supplementary Fig. 1b) ER+ breast cancer cells. Importantly, SOC treatment induced γ-H2AX, a marker of DNA damage, without activating homologous recombination (HR) as shown by the lack of RAD51 foci formation upon short-term treatment when there is not yet a change in cell cycle distribution (Fig. 1c, d, Supplementary Fig. 1c, d). On the other hand, the DNA damaging agent, etoposide activated HR as demonstrated by the qRT-PCR-based HR reporter assay, as well as the RAD51 foci formation[14,15], demonstrating the HR-proficiency of the *BRCA1/2*-wt T47D cells (Fig. 1c, d, Supplementary Fig. 1e, f). To determine if the lack of RAD51 foci formation which is the indicator of a functional BRCAness phenotype is due to reduced levels of *BRCA1*, we first examined the *BRCA1* expression upon SOC treatment in the *BRCA1/2*-wt T47D cells. In line with the strong downregulation in our 'SOC sensitivity' signature (Supplementary Data 1), BRCA1 protein expression was reduced upon SOC treatment (Fig. 1e). To further validate the causal role of BRCA1 downregulation in terms of reducing DNA repair capacity, we overexpressed *BRCA1* together with SOC treatment for 4 hours and observed reduced levels of SOC-induced γ-H2AX and p-Chk2 (Fig. 1e) without a significant change in the cell cycle distribution (Supplementary Fig. 1c, d). Importantly, in addition to reducing BRCA1 levels, SOC treatment also reduced RAD51 expression, in line with the lack of foci formation which was rescued by BRCA1 overexpression (Fig. 1e). Overall, these results suggest a functional BRCAness phenotype in the *BRCA1/2*-wt ER+ breast cancer cells upon SOC treatment that precedes SOC-induced G1 arrest.

To dissect the mechanisms of SOC-induced DNA damage, we first examined the expression of key DNA repair-related proteins from the 'SOC sensitivity' signature (Supplementary Data 1) together with the G1/S transition marker p-RB (S807/811) upon SOC treatment and observed significant depletion of the DNA repair proteins, such as FEN1 and XRCC1, along with BRCA1 in addition to the G1/S progression marker, p-RB in a time-dependent manner (Fig. 1f, Supplementary Fig. 1g). Importantly, BRCA2 protein expression was also strongly reduced upon SOC treatment (Fig. 1f), suggesting that SOC-induced BRCAness is not only restricted to loss of BRCA1, but also involves BRCA2 downregulation, thus preventing any potential compensation or epistatic interaction between the two partners in the HR pathway. Loss of XRCC1 and FEN1 was shown to cause toxic PARP1 trapping on the chromatin upon DNA damage in BRCA-deficient cells[16,17]. This prompted us to test if SOC induces toxic PARP1 trapping as a common mechanism of growth inhibition. Indeed, we detected increased PARP1 trapping on the chromatin upon SOC treatment of T47D cells along with PARP1 interactor histone PARylating factor 1 (HPF1) (Fig. 1g). ADP ribosylation and PARP1-dependent polymer (PAR) formation are hallmarks of increased PARP1 activity[18]. We demonstrated that PARP trapping is accompanied by increased ADP ribosylation (Fig. 1h), PARP1 auto-PARylation (Fig. 1i), as well as increase in histone H3 serine ADP-ribosylation (H3S10-ADPR) (Fig. 1j), a modification known to be mutually exclusive with H3K9 acetylation[19], the marker of active transcription[20]. Along these lines, SOC treatment resulted in transcription inhibition as shown by immediate reduction of H3K9Ac levels that is sustained over time, up

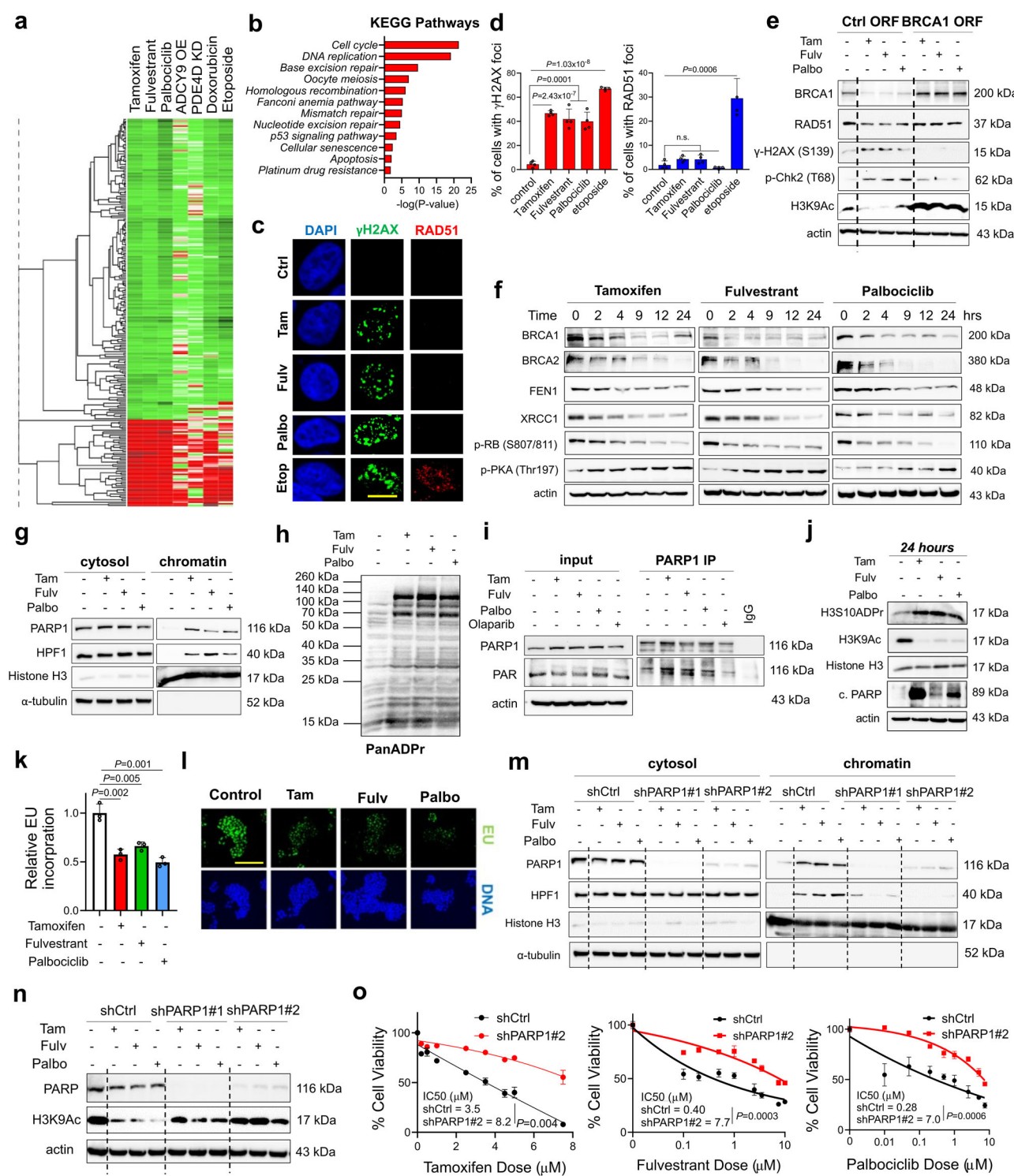

to 24 hours (Fig. 1j, Supplementary Fig. 2a). Furthermore, reduction of global transcription was also demonstrated by decreased 5-ethynyl uridine (EU) staining upon SOC treatment (Fig. 1k, l). To functionally demonstrate that PARP1 trapping upon SOC is indeed toxic, i.e., leading to reduction of cell viability, we silenced PARP1 using shRNAs or siRNAs which resulted in reduced levels of PARP1 as well as its interactor HPF1 on the chromatin (Fig. 1m, Supplementary Fig. 2b) that was followed by the rescue of H3K9Ac (Fig. 1n, Supplementary Fig. 2c), indicating transcriptional recovery. Importantly, upon PARP1 knockdown, the sensitivity of T47D and MCF-7 cells to SOC therapy was significantly reduced in a time and dose-dependent

manner (Fig. 1o, Supplementary Fig. 2d, e), suggesting a critical role of PARP1 in mediating SOC-induced toxicity. Together, these data suggest the existence of a common mechanism of SOC-induced growth inhibition due to DNA damage and toxic PARP1 trapping upon loss of key DNA repair proteins that ultimately causes global transcription inhibition, G1 arrest and apoptosis.

## SOC-induced DNA damage, G1 arrest and apoptosis are mediated by cAMP-regulated mitochondrial ROS generation

Since oxidative stress is known to induce PARP1 trapping[12], we hypothesize that SOC-dependent generation of reactive oxygen

**Fig. 1 | SOC therapy commonly induces DNA damage, BRCAness and toxic PARP1 trapping, leading to transcriptional blockage and growth inhibition in ER+ breast cancer. a** Heatmap of commonly differentially expressed genes in MCF-7 cells treated with SOC (tamoxifen or fulvestrant or palbociclib) for 24 hours from the Connectivity Map database. Green: downregulated genes; red: upregulated genes. **b** The pathway enrichment analysis of the SOC sensitivity signature. **c** IF staining of γ-H2AX (S139) (green) and RAD51 foci (red) in T47D cells upon treatment with SOC for 4 hours. DAPI (blue) was used to stain the nucleus, here and in all relevant figures. Etoposide was used as a positive control. Scale bar = 100 μm. **d** The quantification of γ-H2AX positive cells (left) and those that are also RAD51 foci positive (right) (n = 4 different areas, with at least 100 cells per area). **e** Western blot analyses of BRCA1, RAD51, DNA damage markers and H3K9Ac in T47D cells over-expressing ctrl vs. *BRCA1* ORF and treated with SOC for 4 hours. **f** Western blot analyses of DNA repair proteins, BRCA1, BRCA2, FEN1 and XRCC1 and G1/S transition marker, p-RB (S807/811) and p-PKA (Thr197) in T47D cells treated with SOC in a time-dependent manner. **g** Chromatin occupancy of PARP1 and HPF1 upon treatment of T47D cells with SOC therapies for 2 hours. Histone H3 and α-tubulin were used as the loading controls for nuclear and cytosol fractions, respectively, here

and in all relevant figures. **h** Western blot analysis of ADP ribosylation (ADPR) in T47D cells treated with SOC for 1 hour. **i** PARP1 immunoprecipitation (IP) in SOC-treated T47D cells followed by immunoblotting for PARylation. **j** Western blot analysis of H3S10 ADPR, acetylated H3K9 (H3K9Ac), Histone H3 and cleaved PARP in T47D cells treated with SOC for 24 hours. **k** Relative EU incorporation in SOC-treated cells to show blockage of global transcription (n = 3). **l** Representative images of the EU staining (green) in SOC-treated cells from k. Scale bar = 200 μm. **m** Chromatin occupancy of PARP1 and HPF1 upon treatment of T47D shCtrl vs. shPARP1 cells with SOC therapies for 2 hours. **n** Western blot analysis of H3K9Ac in T47D shCtrl vs. shPARP1 cells treated with SOC for 4 hours. **o** Percentage growth inhibition in T47D cells with shPARP1 and treated with increasing doses of SOC for 5 days (n = 4). Actin is used as a loading control in in all Western blots unless stated otherwise. Data are presented as mean values ± standard deviation (SD). *P*-values were calculated with paired (**o**) or unpaired (**d, k**), two-tailed Student's t test. n.s., not significant (*P* > 0.05). μM: micromolar, for all figures. Experiments in e-j, m are repeated twice with similar results. Source data for this figure are provided as a Source Data file.

species (ROS) would be responsible for DNA damage induction and associated PARP1 trapping. Consistent with our hypothesis, we detected a significant time-dependent increase in the intracellular ROS levels upon tamoxifen, fulvestrant or palbociclib treatment of T47D cells (Fig. 2a, Supplementary Fig. 3a–c). Combining SOC with the ROS scavenger, N-acetylcysteine (NAC) reduced SOC-induced ROS (Fig. 2a) and DNA damage (Supplementary Fig. 3d–g). To identify the source of cytosolic ROS induction, we examined mitochondrial ROS levels upon SOC treatment. Staining of the SOC-treated cells with the mitochondrial ROS-specific dye, MitoSOX demonstrated a significant increase in the mitochondrial ROS levels (Fig. 2b, Supplementary Fig. 3h). As the molecular cause of the mitochondrial stress and ROS generation, we detected a prominent increase in the phosphorylation of Protein Kinase A (PKA), a major kinase regulating mitochondrial function[21]. Intriguingly, the time-dependent increase in PKA phosphorylation accompanied the downregulation of the DNA repair proteins, BRCA1, BRCA2, FEN1 and XRCC1 and loss of RB phosphorylation (Fig. 1f, Supplementary Fig. 1g). PKA is a major downstream effector of cyclic AMP (cAMP) second messenger. Therefore, we first measured the intracellular cAMP levels in SOC-treated MCF-7 and T47D cells and observed a significant accumulation upon treatment with SOC (Fig. 2c, d). Next, we tested if the PKA activation, which is downstream of cAMP induction, is indeed responsible for mitochondrial ROS generation. Inhibition of PKA almost completely blocked SOC-induced mitochondrial ROS generation (Fig. 2e) as well as activation of DDR (Fig. 2f) that ultimately rescued G1/S progression (Supplementary Fig. 3i) and blocked apoptosis (Fig. 2g–i). Notably, SOC treatment induced PKA-dependent phosphorylation of the mitochondrial COXIV subunit I (COXIV-1), which is an indicator of loss of its enzymatic activity, leading to ROS generation (Fig. 2j).

To validate the pathological relevance of our in vitro-derived 'SOC sensitivity' score and its association with the processes we identified, we re-analyzed a neo-adjuvant treated ER+ breast cancer patient dataset, GSE93204[22] (Supplementary Fig. 4a). We found that while SOC sensitivity score positively correlates with cAMP score, it negatively correlates with the proliferation and DNA repair scores over the course of treatment with first-line endocrine therapy, followed by the addition of palbociclib, and finally surgery (Fig. 2k, l, Supplementary Fig. 4b). Importantly, when the response to endocrine therapy or endocrine therapy in combination with palbociclib is high, both SOC sensitivity and cAMP scores are also high, while proliferation and DNA repair scores are low. Likewise, when there is resistance to endocrine therapy or endocrine therapy in combination with palbociclib, SOC sensitivity and cAMP scores are low, while proliferation and DNA repair scores are high (Fig. 2k, l), further validating our in vitro findings. Importantly,

these results were also validated by analyzing an independent dataset of endocrine therapy-treated ER+ breast cancer patients, GSE87411[23]. As shown in Fig. 2m–p, the SOC sensitivity score increased in paired samples of treatment-sensitive patients from baseline to 2 weeks of endocrine therapy treatment, which is accompanied by increased cAMP score and decreased proliferation and DNA repair scores. Overall, these data show a common mechanism of action of SOC therapy where it induces cAMP elevation/PKA activation leading to the generation of ROS that causes DNA damage and toxic PARP trapping-mediated inhibition of global transcription, ultimately leading to G1 arrest and apoptosis.

## SOC therapy activates cAMP-induced DNA damage via down-regulating phosphodiesterase 4D (PDE4D)

PDE4D is one of the major cAMP-specific phosphodiesterases degrading cAMP and known to be overexpressed in cancer[24,25]. We observed a sharp decrease in the levels of PDE4D upon treatment of ER+ breast cancer cells with SOC that accompanies increased PKA phosphorylation and reduced RB phosphorylation (Figs. 1f, 3a, b, Supplementary Fig. 1g). Notably, further supporting the PDE4D-cAMP-DNA damage axis upon SOC treatment, we found that 'SOC sensitivity' genes were similar to those regulated by adenyl cyclase (ADCYC) overexpression or PDE4D knockdown, both of which increase cAMP levels (Fig. 1a). Moreover, inhibition of PDE4D using a specific inhibitor, GebR-7b completely mimicked the effects of SOC by causing a dose-dependent increase in cAMP levels, PKA/CREB phosphorylation and an increase in intracellular and mitochondrial ROS levels (Fig. 3c–g). This further resulted in BRCAness, DNA damage, G1 arrest and apoptosis that ultimately led to a significant growth inhibition in two different ER+ breast cancer cell lines (Fig. 3h–j). Notably, PDE4D inhibition with GebR-7b or BPN14770 only caused a minor growth inhibition in ER-normal mouse or human cells (Supplementary Fig. 5a–c), further supporting the role of PDE4D in regulating ER-dependent cell growth. Supporting these, overexpression of PDE4D in ER+ breast cancer cells almost completely blocked SOC-induced mitochondrial ROS generation (Fig. 3k) and rescued G1/S progression and cell viability by preventing DNA damage and BRCAness in the *BRCA1/2* wt cells (Fig. 3l–o). Altogether, these results indicate that SOC therapy activates cAMP-induced DNA damage via downregulating phosphodiesterase 4D (PDE4D).

## SOC therapy inhibits PDE4D via targeting the ER-PDE4D feedforward loop

Since we observed a reduction in ER levels upon palbociclib treatment similar to fulvestrant (Fig. 3a), we further tested whether palbociclib would reduce ER activity using the ERE reporter assay. As shown in

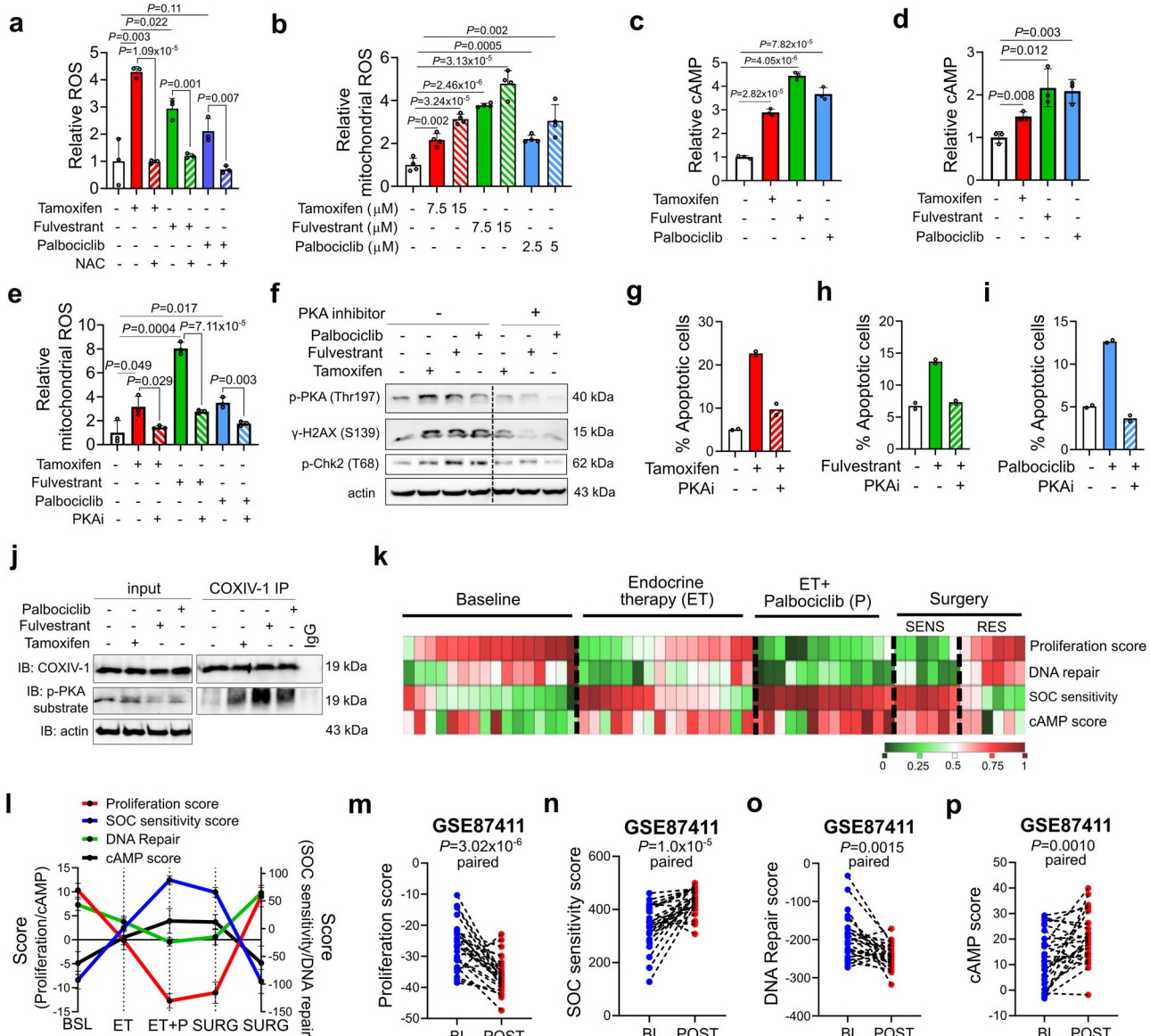

**Fig. 2 | SOC-induced DNA damage, G1 arrest and apoptosis are mediated by cAMP-regulated mitochondrial ROS generation. a** Relative cytoplasmic ROS levels in T47D cells treated with tamoxifen or fulvestrant or palbociclib in the presence or absence of NAC ($n = 3$). **b** MitoSOX staining in T47D cells upon treatment with SOC for 1 hour ($n = 4$). **c, d** Relative cAMP levels in T47D (c) and MCF-7 (d) cells treated with SOC ($n = 3$). **e** MitoSOX staining quantification in T47D cells upon treatment with SOC with or without 1-hour pretreatment with 50 μM of the PKA inhibitor, Rp-Cyclic AMPS ($n = 3$). **f** Western blot analyses of p-PKA (Thr197), γ-H2AX (S139) and p-Chk2 (Thr68) in T47D cells treated with SOC with or without 1-hour pretreatment with 50 μM of the PKA inhibitor, Rp-Cyclic AMPS. The experiment is repeated twice with similar results. **g-i** Percentage of apoptotic cells in T47D cells treated with tamoxifen (**g**), fulvestrant (**h**) or palbociclib (**i**) for 72 hours with or without 20 μM of the PKA inhibitor, Rp-Cyclic AMPS, measured by Annexin V/DAPI staining ($n = 2$). **j** COXIV-1 IP in SOC-treated T47D cells blotted for p-PKA substrate antibody. **k** Heatmap of proliferation score, SOC sensitivity score, cAMP score and DNA repair score in ER+ breast cancer patients from GSE93204 treated with endocrine therapy (ET), followed by addition of palbociclib (P) upon endocrine resistance development and finally underwent surgery. SENS represents sensitive patients at surgery while RES represents resistant patients. **l** Correlations of proliferation score, SOC sensitivity score, cAMP and DNA repair scores in ER+ breast cancer patients from GSE93204 ($n = 16$ for BSL and ET, $n = 13$ for ET + P, n = 6 for SURG-sens and SURG-res). **m–p** Changes of proliferation score (**m**), SOC sensitivity score (**n**), DNA repair score (**o**), and cAMP score (**p**) in paired samples of sensitive patients from baseline to 2 weeks of endocrine therapy treatment from GSE87411 ($n = 27$). Data are presented as mean values ± SD. P-values for the bar graphs were calculated with the unpaired, two-tailed Student's t test while the P-value for m-p were calculated with the paired two-tailed Student's t test. Source data for this figure are provided as a Source Data file.

Fig. 4a, all SOC therapies, including palbociclib, reduced ER activity. These results raised an intriguing possibility that PDE4D, which is downregulated upon SOC treatment (Fig. 3a), could be a novel ER target gene. To test this hypothesis, we stimulated MCF-7 cells with 10 nM estradiol and observed a sharp increase in PDE4D expression at both mRNA and protein levels (Fig. 4b, c). Importantly, treatment with the SOC therapies completely blocked the E2-induced PDE4D induction similar to Cyclin D1 expression (Fig. 4c), a known ER target gene[26].

Notably, BRCA1 expression was also induced upon E2 stimulation and inhibited by SOC therapies (Fig. 4c). We validated the binding of ER to the *PDE4D* promoter by ChIP assay (Fig. 4d).

Gene Set Enrichment Analysis (GSEA) in an ER+/HER2- breast cancer dataset revealed significant enrichment of genes upregulated in ER+ as compared to ER- tumors among high PDE4D-expressing patients' tumors (Fig. 4e). Therefore, we hypothesized that E2-induced PDE4D may have a functional role in promoting ER activity

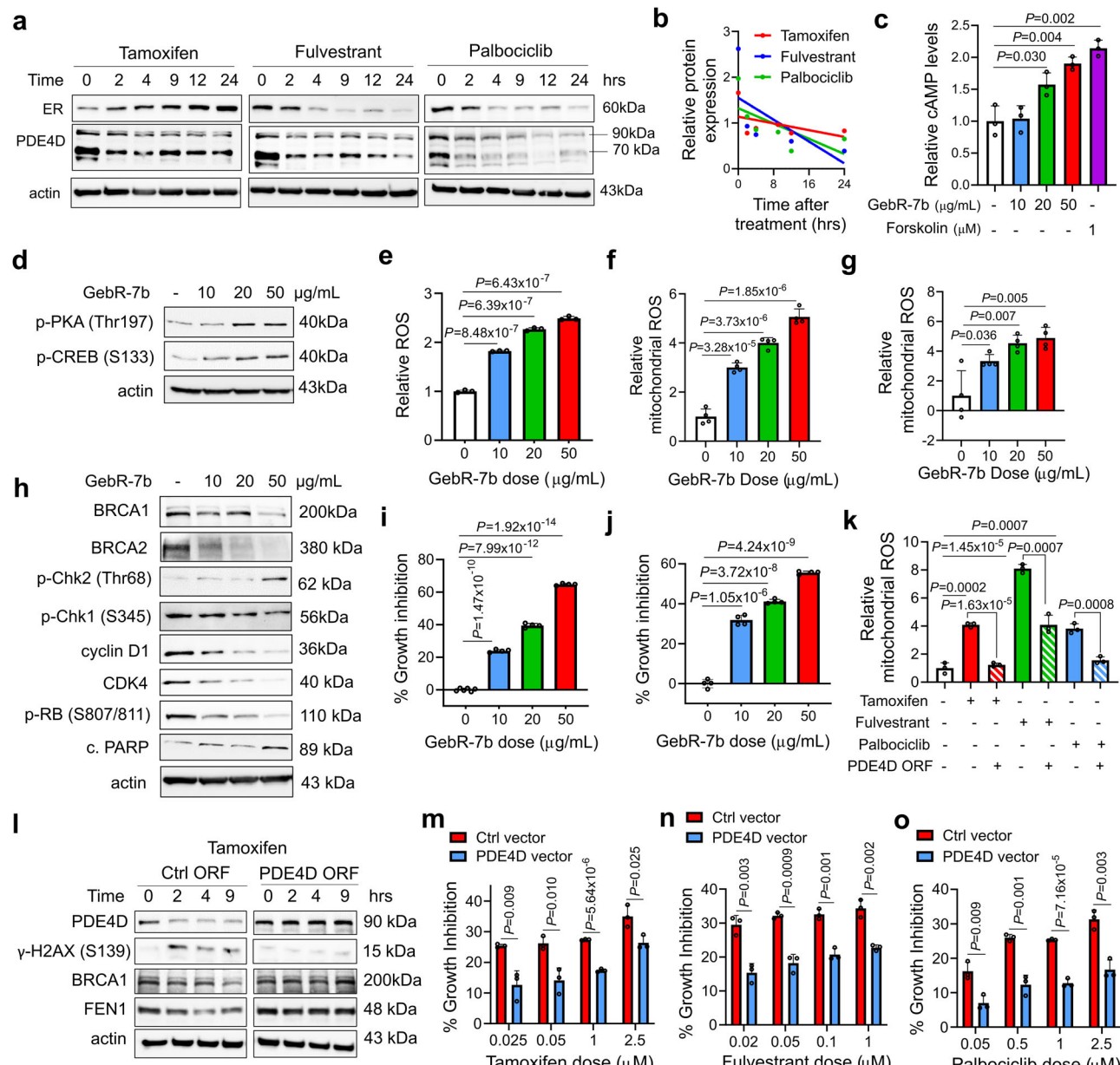

**Fig. 3 | SOC therapy triggers mitochondrial ROS generation, BRCAness and DNA damage via inhibiting PDE4D. a** Western blot analyses of ER and PDE4D in T47D cells treated with SOC in a time-dependent manner. **b** Quantification of PDE4D band intensities relative to actin in SOC-treated T47D cells in a time-dependent manner. **c** Relative cAMP levels in T47D cells treated with increasing doses of the PDE4D inhibitor, GebR-7b or the cAMP inducer, forskolin as a positive control (n = 3). **d** Western blot analyses of p-PKA and p-CREB in T47D cells treated with increasing doses of GebR-7b. **e** Relative ROS levels in T47D cells treated with increasing doses of GebR-7b (n = 3). **f, g** MitoSOX staining in T47D (**f**) and MCF-7 (**g**) cells upon treatment with increasing doses of GebR-7b (n = 4). **h** Western blot analyses of DNA damage, DNA repair, G1 arrest and apoptosis markers in T47D cells treated with increasing doses of GebR-7b for 24 hrs. **i, j** Percent growth inhibition in T47D (**i**) and MCF-7 (**j**) cells treated with increasing doses of GebR-7b for 3 days (n = 6 for control and n = 4 for GebR treatment for **i**; n = 4 for **j**). **k** MitoSOX staining in T47D cells upon treatment with SOC in the presence or absence of the PDE4D ORF (n = 3). **l** Western blot analyses of DNA damage and G1 arrest markers in T47D cells overexpressing PDE4D ORF and treated with tamoxifen. **m-o** Percentage growth inhibition in control (ctrl) vector vs. PDE4D vector-transfected T47D cells treated with increasing doses of tamoxifen (**m**) or fulvestrant (**n**) or palbociclib (**o**) for 2 days (n = 3). Data are presented as mean values ± SD. P-values were calculated with the unpaired, two-tailed Student's t test. Experiments in a, h, l are repeated twice with similar results. Source data for this figure are provided as a Source Data file.

in a feedforward loop and contributes to ER-dependent cellular growth. We showed that similar to the SOC therapies, inhibition of PDE4D using GebR-7b or induction of cAMP using a cAMP inducer, forskolin significantly reduced the E2-induced cell growth (Fig. 4f) as well as ER transcriptional activity (Fig. 4g). Importantly, another PDE4D inhibitor, BPN14770, which is currently in clinical trials (NCT05163808)[27], also caused a similar or greater inhibitory effect on E2-induced cell growth as compared to GebR-7b (Fig. 4f). Next, we

asked if ERK1/2-mediated ER phosphorylation could be regulated by PDE4D/cAMP/PKA as a mechanism of PDE4D-dependent ER activation. E2 stimulation of charcoal-starved MCF-7 cells caused a sharp increase in ER phosphorylation at the S118 residue which was strongly reduced by GebR-7b, along with a decrease in ERK1/2 phosphorylation (Fig. 4h). Tamoxifen also reduced p-ER (S118), albeit independent of ERK1/2 inhibition. The ERK1/2 inhibitor, ulixertinib also caused a strong reduction of the E2-induced ER phosphorylation as expected (Fig. 4h).

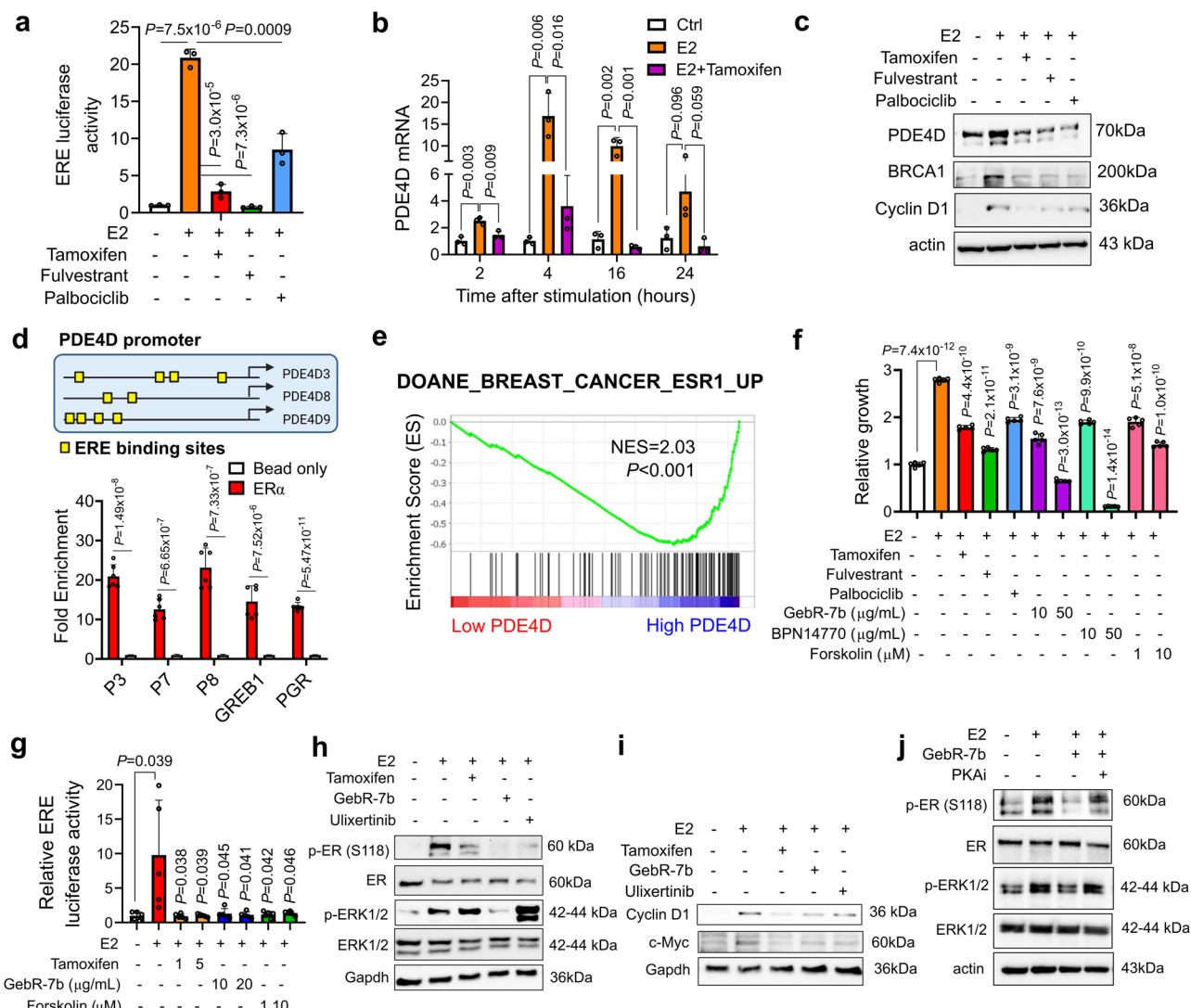

**Fig. 4 | SOC therapy reduces ER signaling via modulating PDE4D/cAMP which in turn regulates ER activity in a feedforward loop. a** ERE reporter assay in E2-stimulated (10 nM) MCF-7 cells with or without SOC (*n* = 3). **b, c** qRT-PCR of PDE4D (**b**) (*n* = 3) and Western blot analyses of PDE4D, Cyclin D1 and BRCA1 (**c**) in MCF-7 cells stimulated with 10 nM E2 in the presence or absence of SOC. **d** ChIP assay of ER in MCF-7 cells showing binding to *PDE4D* promoter (*n* = 6). The putative ER binding sites on the *PDE4D* promoters are depicted. The isoforms PDE4D3, PDE4D8 and PDE4D9 express the 70 kDa protein. **e** GSEA analysis showing enrichment of genes upregulated in ER+ tumors among high PDE4D-expressing ER+/HER2- breast cancer patients' tumors from GSE81538. **f, g** Relative growth (**f**) and ER activity (**g**) in MCF-7 cells stimulated with 10 nM E2 in the presence or absence of SOC, GebR-7b, BPN14770 or forskolin (*n* = 5). **h, i** Western blot analyses in MCF-7 cells treated with tamoxifen, GebR-7b or the ERK inhibitor, ulixertinib for 20 min (**h**) or 24 hours (**i**) in combination with E2. GAPDH is used as the loading control. **j** Western blot analysis of p-ER, ER and p-ERK in MCF-7 cells stimulated with E2 (10 nM) in the presence of PKA inhibitor (100 μM) with or without GebR-7b (20 μg/mL). Data are presented as mean values ± SD. *P*-values were calculated with the unpaired, two-tailed Student's t test. Experiments in h-j are repeated twice with similar results Source data for this figure are provided as a Source Data file.

The reduction of p-ER (S118) upon PDE4D or ERK1/2 inhibition was further accompanied by reduced expression of ER targets, Cyclin D1 and c-Myc (Fig. 4i). Importantly, pretreatment of cells with the PKA inhibitor for 1 hour, followed by PDE4D inhibition and then stimulation with E2 resulted in rescue of ERK1/2 phosphorylation and the subsequent ER phosphorylation (Fig. 4j). Overall, SOC therapy reduces PDE4D via inhibiting ER activity and this reduction in PDE4D, in turn, blocks ER activity by activating cAMP/PKA and inhibiting ERK1/2 in a feedforward loop.

## PDE4D mediates SOC resistance, and inhibiting PDE4D induces BRCAness, DNA damage, PARP1 trapping and transcriptional blockage to restore SOC sensitivity

To test the clinical relevance of PDE4D in SOC resistance, we re-analyzed a metastatic, endocrine therapy-treated ER+ breast cancer patient dataset, GSE124647[28]. An endocrine therapy resistance gene set was strongly enriched in high PDE4D-expressing resistant patients (Fig. 5a), suggesting the importance of PDE4D in SOC resistance. Importantly, higher mRNA expression of *PDE4D* was associated with significantly worse overall survival (OS) among metastatic endocrine-resistant patients (Fig. 5b). Analysis of a large dataset comprising 2283 patients with primary, ER+ breast cancer treated with endocrine therapy revealed that high mRNA expression of *PDE4D* is associated with significantly worse relapse-free survival (RFS) also in early-stage disease (Supplementary Fig. 4c). Next, we analyzed the effects of PDE4D protein levels on patient survival and disease relapse in our own cohort of 171 early and late-stage ER+ breast cancer patients treated with endocrine therapy. As shown in Fig. 5c, d, there is a significantly higher chance of disease relapse in patients expressing high levels of PDE4D protein. Furthermore, higher PDE4D protein level is associated

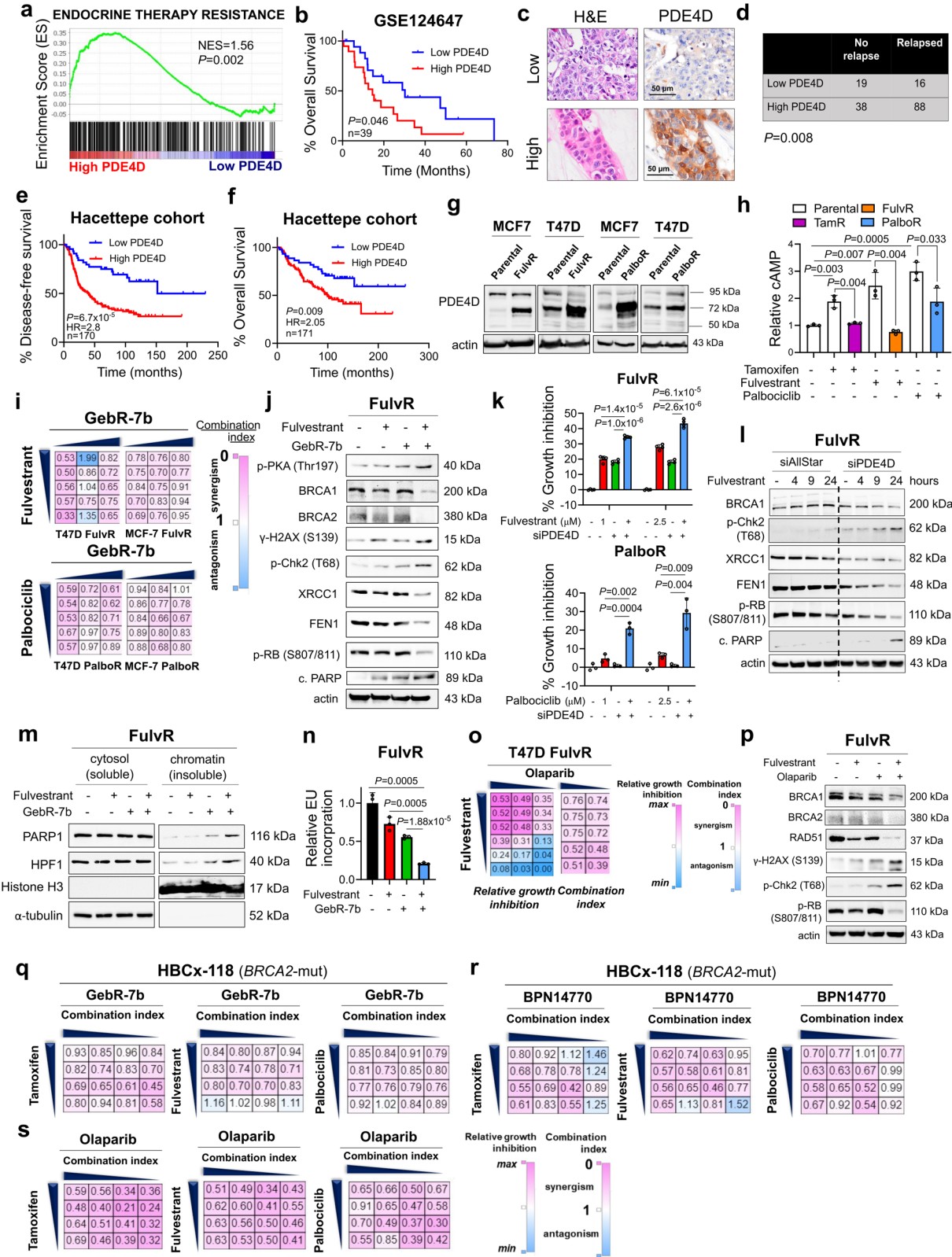

with dramatically worse disease-free survival and overall survival (Fig. 5e, f).

Having validated the clinical relevance of PDE4D as an attractive therapeutic target in SOC-resistant ER+ breast cancer, we then developed tamoxifen, fulvestrant and palbociclib-resistant derivatives of MCF-7 and T47D cells by long-term drug treatment over 9 months (Supplementary Fig. 6a–c) to examine the mechanistic roles of PDE4D

in SOC resistance. The dose-response survival analysis for each model demonstrated that our MCF-7 and T47D-resistant models survive better than the parental counterparts upon treatment with increasing doses of SOC. Furthermore, we also tested the cross-resistance/sensitivity of each of these models to SOC therapies and demonstrated that the tamoxifen and fulvestrant-resistant models are cross-resistant to each other while they were more sensitive to palbociclib as in the

**Fig. 5 | PDE4D mediates SOC resistance and inhibiting PDE4D induces BRCA-ness, DNA damage, PARP1 trapping and transcriptional blockage to restore SOC sensitivity. a** GSEA analysis showing enrichment of endocrine therapy resistance genes among high PDE4D-expressing endocrine resistant patients from GSE124647. **b** Kaplan-Meier overall survival analysis in endocrine-resistant metastatic ER+ breast cancer patients from GSE124647 based on *PDE4D* mRNA expression. **c** IHC images of ER+ breast cancer patient tissues with low and high PDE4D protein expression, together with the corresponding H&E staining in the Hacettepe cohort. **d** Chi-square testing showing significant association of high PDE4D protein expression with disease relapse. **e, f** Kaplan-Meier disease-free (**e**) and overall (**f**) survival analyses in endocrine-treated ER+ breast cancer patients from the Hacettepe cohort based on PDE4D protein expression. **g** Western blot analysis of PDE4D in SOC parental vs. resistant MCF-7 and T47D cells. **h** Relative cAMP levels in SOC parental vs. resistant T47D cells treated with SOC therapy (*n* = 3). **i** The heatmap of combination indices upon treatment of SOC resistant cells with the combination of SOC (fulvestrant: 0.05, 0.1, 05, 2.5, 10 μM for T47D and 1, 5, 10, 12.5, 25 μM for MCF-7; palbociclib: 0.25, 0.5, 1, 2.5, 3.5 μM for T47D and 5, 10, 12.5, 15, 16.5 μM for MCF-7) and the PDE4D inhibitor, GebR-7b (10, 15, 20 μg/mL). The scale bar for the combination index matrices is provided at the right-hand side, here and for all heatmaps. **j** Western blot analyses of DNA damage, DNA repair, G1 arrest and apoptosis markers in T47D FulvR cells treated with the combination of fulvestrant with GebR-7b. **k** Percent growth inhibition in T47D FulvR and PalboR cells upon PDE4D knockdown and treatment with fulvestrant or palbociclib, respectively for 3 days (*n* = 4). **l** Western blot analyses of DNA damage, G1 arrest and apoptosis markers in T47D FulvR cells upon PDE4D knockdown and treatment with fulvestrant. **m** Chromatin occupancy of PARP1 and HPF1 in T47D FulvR cells treated with fulvestrant in combination with GebR-7b. **n** Relative EU incorporation in T47D FulvR cells treated with the combination of fulvestrant and GebR-7b (*n* = 3). **o** Heatmaps of relative growth inhibition and combination indices in T47D FulvR cells treated with the combination of fulvestrant (0.05, 0.1, 0.5, 2.5, 10 μM) with olaparib (5, 7.5 μM). **p** Western blot analysis of DNA damage, DNA repair and G1 arrest markers in T47D FulvR cells treated with fulvestrant, olaparib or their combination. **q-s** Heatmaps of combination indices in primary cell cultures of HBCx-118 (*BRCA2*-mut) model treated with the combination of different SOC therapies (tamoxifen: 2, 3, 4, 4.5 μM; fulvestrant: 1, 5, 10, 20 μM; palbociclib: 0.5, 1, 1.5, 2.5 μM), and PDE4D inhibitors, GebR-7b (30, 40, 50, 60 μg/mL) (**q**) or BPN14770 (35, 40, 45, 50 μg/mL) (**r**) or with olaparib (2.5, 5, 7.5, 10 μM) (**s**). Data are presented as mean values ± SD. *P*-values for the bar graphs were calculated with the unpaired, two-tailed Student's t test. Significance for the Kaplan-Meier survival graphs was calculated with Log-rank test. Chi-square test was used for d. NES: normalized enrichment score. Experiments in j, l, p are repeated twice with similar results. Source data for this figure are provided as a Source Data file.

clinics. However, palbociclib-resistant models were found to be less responsive to endocrine therapy (Supplementary Fig. 6d, e). These results demonstrate the resistance phenotypes of our models and their clinical relevance. We found that the long PDE4D isoforms (70 kDa and 90 kDa) are overexpressed in SOC-resistant cells at both protein and mRNA levels (Fig. 5g, Supplementary Fig. 7a). Long PDE4D isoforms contain the regulatory upstream conserved regions (UCR1, UCR2) which are critical for dimerization, catalytic activity, and inhibitor binding[29]. Overexpression of PDE4D was further accompanied by reduced SOC-induced cAMP induction in the resistant models as compared to the sensitive counterparts (Fig. 5h). Inhibiting PDE4D using GebR-7b caused mostly a synergistic growth inhibition (combination index (CI) < 1) in combination with SOC in resistant models (Fig. 5i). At the molecular level, PDE4D inhibition restored cAMP signaling activity, BRCAness, SOC-induced ROS generation, DNA damage, G1 arrest and apoptosis in endocrine resistant cells with wt *BRCA1/2* (Fig. 5j). Importantly, siRNA-mediated knockdown of PDE4D increased SOC-mediated growth inhibition and activation of DDR, G1 arrest and apoptosis in resistant models similar to PDE4D inhibitor (Fig. 5k, l, Supplementary Fig. 7b). Furthermore, PDE4D inhibition in combination with fulvestrant caused PARP1 trapping along with increased levels of its interactor HPF1 on the chromatin (Fig. 5m). These ultimately resulted in the blockage of global transcription (Fig. 5n, Supplementary Fig. 7c).

Importantly, combining SOC with the PARP1 inhibitor, olaparib which is known to cause PARP1 trapping on the chromatin[30], led to synergistic growth inhibition at multiple doses (Fig. 5o, Supplementary Fig. 8a), leading to induction of DNA damage, loss of BRCA1, BRCA2 and RAD51 in the *BRCA1/2*-wt cells and induction of G1 arrest (Fig. 5p), phenocopying the PDE4D inhibition (Fig. 5j). Combination of SOC with olaparib in two different ER+/HER2+ cell line models with *BRCA2* mutations of unknown clinical significance and allelic loss of *BRCA1* and *BRCA2*[31–33], MDA-MB-361 and ZR-75-30 (Supplementary Fig. 8c, f) that are resistant to SOC (Supplementary Fig. 8b) also caused mostly synergistic growth inhibition (Supplementary Fig. 9a, b) along with increased DNA damage, G1 arrest and apoptosis (Supplementary Fig. 9c). These *BRCA2*-mut cells fail to induce HR activity (Supplementary Fig. 8d, g) and form RAD51 foci (Supplementary Fig. 8e, h) compared to the *BRCA1/2*-wt T47D cells (Supplementary Fig. 1e, f), consistent with the allelic loss of *BRCA1* and *BRCA2* in these cell lines (Supplementary Fig. 8c, f). The *BRCA1*-mut MDA-MB-436 cells were also unable to activate HR and form RAD51 foci as expected (Supplementary Fig. 8i-k). The functionality of the mutant *BRCA2* was also demonstrated by higher sensitivity to olaparib in the *BRCA2*-mut ZR-75-30 cells compared to T47D. Here, MDA-MB-436 cells with known sensitivity to olaparib[34,35] were used as control and demonstrated the highest sensitivity as expected (Supplementary Fig. 10a–c). Notably, PDE4D inhibition in combination with SOC phenocopied the effects of PARP inhibition also in *BRCA2*-mut cell lines (Supplementary Fig. 9a, b, d). These data demonstrate that inhibiting PDE4D or PARP1 overcomes SOC resistance irrespective of *BRCA1/2* status via inducing DNA damage, G1 arrest and apoptosis.

To test the role of PDE4D targeting to overcome SOC resistance in a more clinically relevant setup, we examined the expression of PDE4D in an in vivo-derived tamoxifen-resistant ER+ PDX, HBCx22 TamR as well as in three different PDX models collected from endocrine resistant, metastatic ER+ breast cancer patients (Supplementary Fig. 11a, Supplementary Table 1)[36]. Notably, one of these models, HBCx-118 has *BRCA2* mutation and allelic loss of *BRCA2*[36], indicative of defective DNA repair (Supplementary Fig. 11b), and the other PDX, HBCx-131 has amplified *CCND1* and deletion of *CDKN2A/B* (p16/p15), indicative of high CDK4/6 activity. PDE4D was upregulated at the protein level in HBCx22 TamR as compared to its sensitive counterpart (Supplementary Fig. 11c). Notably, PDE4D was upregulated in the endocrine-resistant, metastatic PDXs (HBCX-118, HBCX-131 and HBCX-139) (Supplementary Fig. 11c). The upregulation of PDE4D in these models was also validated at mRNA level (Supplementary Fig. 11d). Next, we isolated primary cells from these PDXs overexpressing PDE4D and first validated the lack of HR activity in the *BRCA2*-mut HBCx-118 PDXs compared to the *BRCA1/2*-wt HBCx-131 PDXs by HR reporter assay and RAD51 foci staining (Supplementary Fig. 11e–j). In line with the lack of HR activity, the *BRCA2*-mut HBCx-118 PDX cells were more sensitive to olaparib compared to the *BRCA1/2*-wt HBCx-131 PDX cells (Supplementary Fig. 11k). After the characterization of the PDX cell lines, we next tested the combination of SOC with the PDE4D inhibitors, GebR-7b and BPN14770, and PARP inhibitor, olaparib in terms of growth inhibition. Strikingly, the combination therapies caused mostly a synergistic growth inhibition at multiple doses of all the drugs tested in two of the PDX models, HBCX-118 and HBCX-131 independent of their *BRCA1/2* status (Fig. 5q–s, Supplementary Fig. 12a, b), while it had only a minor effect on HBCX-139 (Supplementary Fig. 12c) potentially due to lower PDE4D expression, heterogeneity, and different drivers of resistance in this PDX model. Overall, our results show that in addition to PDE4D downregulation being a major determinant of response to SOC therapy, overexpression of PDE4D confers SOC resistance and has clinical relevance, and inhibition of PDE4D overcomes SOC therapy

resistance by inducing PARP1 trapping irrespective of the *BRCA1/2* status.

### A switch from ER to EGFR dependence characterizes SOC resistance, upstream of PDE4D, and targeting EGFR overcomes SOC resistance

It is known that cells that are resistant to endocrine therapy may lose their ER dependence and can instead rely on the activation of compensatory pathways[6,37–39]. To test the ER dependency of our SOC-resistant models, (i) we cultured both sensitive and resistant cells in E2-deprived conditions using charcoal-stripped FBS media, or (ii) we silenced ER using an siRNA under normal serum condition to eliminate potential effects of using charcoal-stripped FBS on other pathways. As a result, we observed a significantly higher growth rate of the resistant cells as compared to sensitive counterparts under E2 deprivation by charcoal stripping or upon ER knockdown (Supplementary Fig. 13a, b). The E2-deprived cells were also less responsive to E2-mediated transcription (Supplementary Fig. 13c), suggesting that SOC-resistant cells grow in an ER-independent manner. Strikingly, inhibition of PDE4D using GebR-7b or BPN14770 under E2-deprived conditions by charcoal stripping or when ER is depleted by siRNAs (both of which mimic SOC treatment that inhibits ER), significantly reduced the growth of all SOC-resistant cells in a dose-dependent manner (Fig. 6a–c, Supplementary Fig. 13e), further validating the effectiveness of our combination therapy on overcoming SOC resistance. Among the compensatory signaling pathways that replace ER is different receptor tyrosine kinases (RTKs). GSEA analysis in an ER+/HER2- breast cancer dataset revealed significant enrichment of genes upregulated upon EGFR overexpression among high PDE4D-expressing patients' tumors (Fig. 6d). In line with this, we demonstrated that EGFR has been phosphorylated at multiple residues in FulvR and PalboR cells (Fig. 6e–g, Supplementary Fig. 13d), together with activation of downstream AKT and ERK1/2 signaling (Fig. 6g). Inhibition of EGFR using gefitinib or neratinib in E2-deprived condition upon charcoal stripping or in ER-depleted condition upon siRNA knockdown of ER significantly reduced cell growth in SOC resistant cells in a dose-dependent manner (Supplementary Fig. 13e, f) similar to PDE4D inhibition. We next asked if the switch from ER to EGFR in the SOC-resistant cells could be responsible for PDE4D overexpression in SOC resistance. We stimulated ER+ breast cancer cells with EGF and observed a significant increase in PDE4D levels at both mRNA and protein levels (Fig. 6h, i). Furthermore, silencing c-Jun, which is among the major downstream effector transcription factors of EGF signaling[40], blocked EGF-induced PDE4D expression (Fig. 6h, i). We then performed c-Jun ChIP assay and observed a significant enrichment of c-Jun on the promoter regions of 90 and 70 kDa long *PDE4D* isoforms upon EGF stimulation similar to the known targets of c-Jun, ERCC1[41] and Bcl-xL[42] (Fig. 6j).

To test if targeting EGFR, upstream of PDE4D overcomes SOC resistance, we inhibited EGFR using the EGFR inhibitor, gefitinib in combination with SOC. We observed mostly synergistic growth inhibition (combination index <1) in all SOC resistant cell line models upon combination treatment with SOC and gefitinib (Fig. 6k). At the molecular level, EGFR inhibition downregulated PDE4D expression and restored fulvestrant-induced DNA damage, BRCAness and G1 arrest in *BRCA1/2*-wt cells (Fig. 6l). Furthermore, combination of fulvestrant with gefitinib caused toxic PARP1 trapping and increased occupancy of HPF1 on the chromatin, mimicking PDE4D inhibition (Fig. 6m). Lastly, we tested SOC combination with gefitinib in the primary cultures of refractory ER+ PDXs (HBCx-118 and HBCx-131) which also have activated EGFR signaling (Supplementary Fig. 11c) and showed mostly a synergistic growth inhibition (Fig. 6n, Supplementary Fig. 13g). Combining SOC with gefitinib in *BRCA2*-mut ER+/HER2+ breast cancer cell lines also caused mostly synergistic growth inhibition, accompanied by induction of DNA damage, G1 arrest and apoptosis (Supplementary

Fig. 13h–j). Overall, these data suggest that the ER-to-EGFR switch and the subsequent c-Jun activation increases PDE4D in SOC-resistant cells, leading to SOC resistance, and inhibition of EGFR signaling overcomes SOC resistance via downregulation of PDE4D, and restoring SOC-induced DNA damage, PARP trapping, G1 arrest and apoptosis irrespective of the *BRCA1/2* status.

### Targeting PDE4D or its upstream EGFR or its downstream PARP1 overcomes resistance to SOC in PDX organoids and in vivo

Given the strong synergistic effect that we observed in the 2D primary cultures of the PDXs, we further examined the effects of our combination therapies on the growth of organoids of both HBCx-118 (*BRCA2*-mut) and HBCx-131 *(BRCA1/2-wt)* PDXs, to better recapitulate in vivo conditions. Targeting PDE4D using GebR-7b significantly increased the growth inhibition of the HBCx-118 organoids in combination with two different doses of fulvestrant or palbociclib (Fig. 7a–c). Next, we tested the combination of GebR-7b with fulvestrant or palbociclib in vivo and showed a significant decrease in tumor growth upon combination therapies as compared to single agent treatments (Fig. 7d, Supplementary Fig. 14a) without a significant body weight change as compared to SOC alone at the end of treatment (Supplementary Fig. 15a). Combination-treated tumors had lower proliferation rate and higher apoptosis, as shown by Ki67 and TUNEL, respectively (Fig. 7e–g).

Next, we showed significant growth inhibition of both HBCx-118 and HBCx-131 organoids upon combination of multiple doses of fulvestrant+palbociclib combination (F + P), which is now the standard of care for advanced ER+ patients[43], with the clinically tested PDE4D inhibitor, BPN14770[27]; or pan-HER inhibitor, neratinib or PARP1 inhibitor, olaparib (Fig. 8a–d). We then tested these combinations in vivo. Addition of BPN14770 or neratinib to F + P combination significantly reduced the growth of HBCx-118 PDX tumors (Fig. 8e, f, Supplementary Fig. 14b) without a significant body weight change at the end of treatment (Supplementary Fig. 15b). In addition, western blot analysis and IHC staining revealed induction of DNA damage, G1 arrest and apoptosis upon combination treatments (Fig. 8g, Supplementary Fig. 14c). Importantly, staining of the tumors showed a significant increase in ROS levels in tumors treated with SOC in combination with BPN14770 or neratinib (Supplementary Fig. 14d, e). Next, we tested the combination of SOC with olaparib in vivo. Here, we chose a relatively low-to-moderate dose of olaparib that will enable us to observe the synergistic effects of the combination therapy without major toxicity rather than the effects of individual drugs. As a result, we demonstrated reduced tumor growth (Fig. 8h, i, Supplementary Fig. 14f) accompanied by induction of DNA damage and G1 arrest (Fig. 8j) without a significant body weight change in the combination-treated group at the end of treatment (Supplementary Fig. 15c). Here, the tumors were more resistant to SOC treatment compared to previous cohort, probably due to tumor heterogeneity. Despite greater resistance, a combination of SOC with olaparib almost completely blocked tumor growth with 78% tumor growth inhibition (TGI) while treatment with low-dose (35 mg/kg) olaparib alone achieved 14% TGI in the HBCx-118 PDXs. Notably, all combination therapies were effective not only in the *BRCA2*-mut PDX model (HBCx-118), but also in *BRCA1/2*-wt HBCx-131 PDXs in vivo (Fig. 8k–m, Supplementary Fig. 14g), without causing a significant body weight change (Supplementary Fig. 15d) or change in blood counts at the end of treatment (Supplementary Fig. 15e–g). Overall, we show that targeting PDE4D or its upstream EGFR or its downstream PARP1 overcomes resistance to SOC in PDX organoids and in vivo irrespective of the *BRCA1/2* status.

## Discussion

Endocrine therapies modulating ER level and/or activity have been the mainstay therapy for the largest subtype of breast cancer (ER+ breast cancer) and improved the quality of patients' lives and survival. However, resistance to endocrine therapy is still common even after

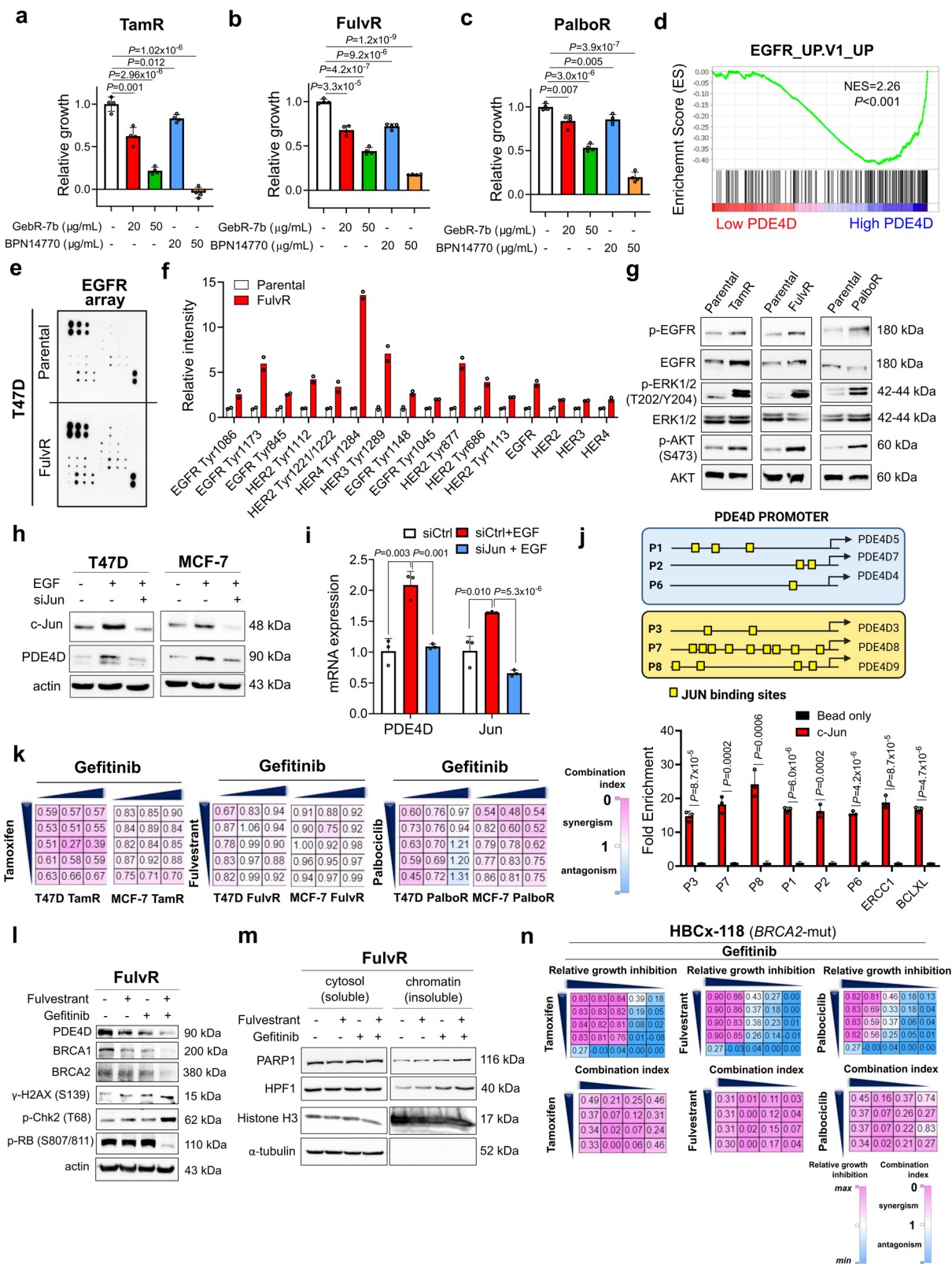

decades of initial therapy. In recent years, CDK4/6 inhibitors have been revolutionary for the treatment of endocrine-resistant disease. However, patients eventually develop resistance also to CDK4/6 inhibitors, reducing patient survival. Therefore, it is of utmost importance to elucidate how SOC therapies work and identify key drivers of resistance that can be targeted to restore SOC sensitivity and ultimately achieve prolonged patient survival. Here, we demonstrate, for the first

time, that SOC therapies used in ER+ breast cancer induce DNA damage, and toxic PARP1 trapping upon generation of a functional BRCAness phenotype through loss of key DNA repair proteins, including BRCA1, BRCA2 and RAD51. Aberrant PARP1 activity on the chromatin then increases H3S10 ADP ribosylation that competes with H3K9 acetylation, thus blocking global transcription. Mechanistically, this is achieved via SOC-induced PDE4D downregulation, cAMP

**Fig. 6 | A switch from ER to EGFR dependence characterizes SOC resistance, upstream of PDE4D, and inhibition of EGFR signaling overcomes SOC resistance. a-c** Relative growth of tamoxifen (**a**), fulvestrant (**b**) and palbociclib (**c**) resistant MCF-7 cells under E2-deprived media, treated with GebR-7b or BPN14770 (*n* = 4). **d** GSEA analysis showing enrichment of genes upregulated upon EGFR overexpression among high PDE4D-expressing ER+/HER2- breast cancer patients' tumors from GSE81538. **e, f** EGFR array image (**e**) and its quantification (**f**) (*n* = 2) in T47D parental vs. fulvestrant resistant cells. **g** Western blot analysis of EGFR and its downstream pathways in parental vs. SOC-resistant T47D cells. For TamR and FulvR cells, EGFR/HER2 (Y1173/Y1248) is shown while for PalboR cells p-EGFR (Y845) is shown. **h** Western blot analysis of PDE4D and c-Jun in T47D and MCF-7 cells transfected with siJun and stimulated with EGF (20 nM) for 24 hours. **i** qRT-PCR analysis of PDE4D in T47D cells transfected with siJun and stimulated with EGF (*n* = 3). **j** ChIP assay of c-Jun in T47D cells stimulated with EGF for 16 hours to show binding to *PDE4D* promoter (*n* = 3 different wells). The putative c-Jun binding sites on the *PDE4D* promoters are depicted. The isoforms, PDE4D5, PDE4D7 and PDE4D4 express the 90 kDa protein, while PDE4D3, PDE4D8 and PDE4D9 express the

70 kDa protein. **k** Heatmaps of combination indices in SOC resistant cells treated with the combination of different SOC therapies (tamoxifen: 0.1, 1, 2.5, 5, 7.5 μM for T47D and 1, 2.5, 5, 7.5, 10 μM for MCF-7; fulvestrant: 0.05, 0.1, 05, 2.5, 10 μM for T47D and 1, 5, 10, 12.5, 25 μM for MCF-7; palbociclib: 0.25, 0.5, 1, 2.5, 3.5 μM for T47D and 5, 10, 12.5, 15, 16.5 μM for MCF-7) and gefitinib (5, 7.5, 10 μM). **l** Western blot analyses of PDE4D, DNA damage, DNA repair and G1 arrest markers in T47D FulvR cells treated with the combination of fulvestrant and gefitinib. **m** Chromatin occupancy of PARP1 and HPF1 in T47D FulvR cells treated with fulvestrant in combination with gefitinib. **n** Heatmaps of relative growth inhibition (upper panel) and combination indices (lower panel) in primary cell cultures of HBCx-118 model treated with the combination of different SOC therapies (tamoxifen: 2, 3, 4, 4.5 μM; fulvestrant: 1, 5, 10, 20 μM; palbociclib: 0.5, 1, 1.5, 2.5 μM) and gefitinib (1, 5, 7.5, 10 μM). Data are presented as mean values ± SD. *P* values were calculated with the unpaired, two-tailed Student's *t* test. NES: normalized enrichment score. Experiments in **g, l, m** are repeated twice with similar results. Source data for this figure are provided as a Source Data file.

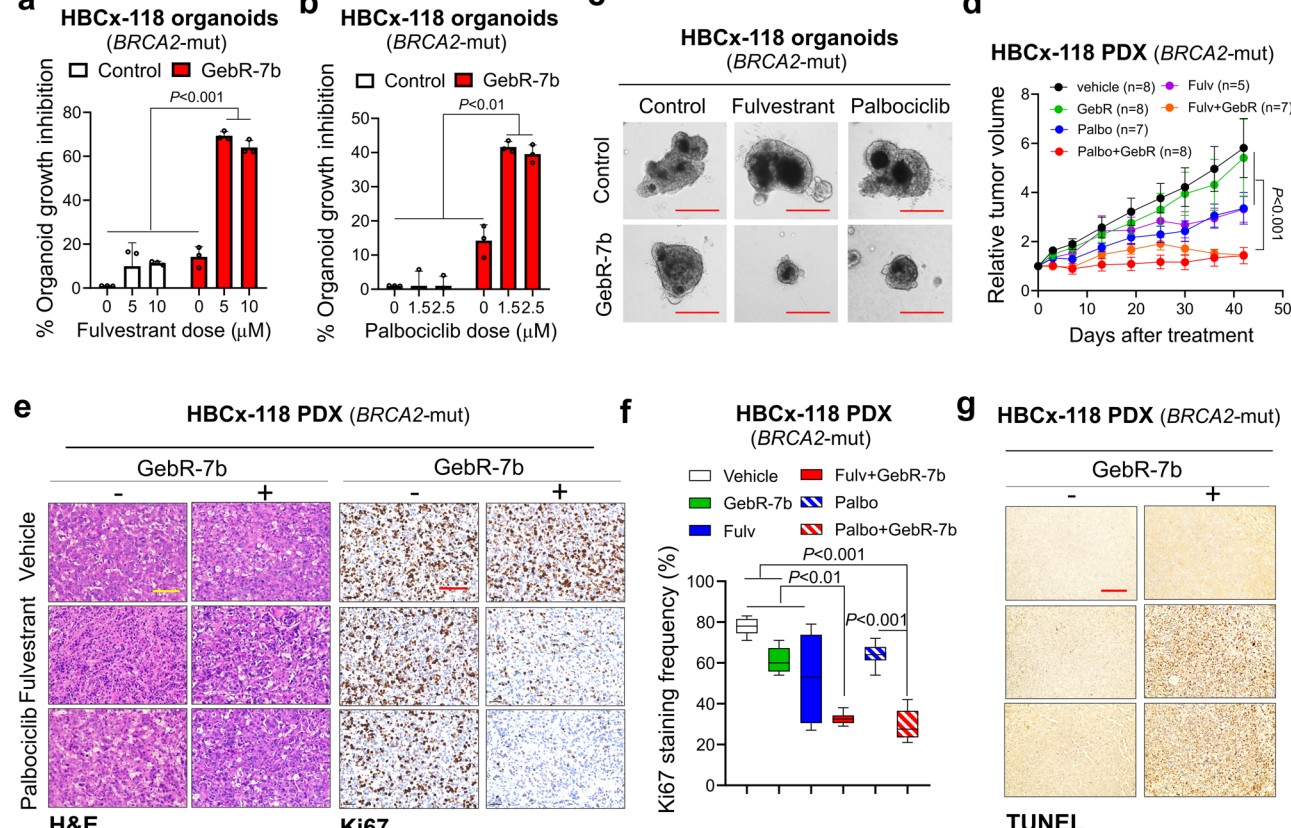

**Fig. 7 | Targeting PDE4D or EGFR overcomes resistance to fulvestrant and palbociclib in PDX organoids and in vivo. a, b** HBCx-118 organoid growth upon the combination of PDE4D inhibitor, GebR-7b (30 μg/mL) and SOC therapies (fulvestrant (**a**) or palbociclib (**b**)) for a week (*n* = 3 different wells). **c** Representative images from HBCx-118 organoids treated with the combination of SOC therapies (fulvestrant or palbociclib) and GebR-7b. Scale bar=50 μm. **d** Tumor growth curves of HBCx-118 PDXs upon treatment with fulvestrant (35 mg/kg, subcutaneous) or palbociclib (35 mg/kg, oral gavage) in combination with PDE4D inhibitor, GebR-7b (3 μg/kg, intraperitoneal) (*n* = 5−8). **e** H&E and Ki67 staining of HBCx-118 PDX tumors treated with fulvestrant or palbociclib in combination with GebR-7b. Scale

bar=100 μm. **f** Quantification of Ki67 staining in tumors from **d** (*n* = 10). The center line shows the median, the box limits show the 75th and 25th percentiles and the whiskers show minimum-maximum values. **g**. TUNEL staining of tumors from d. Scale bar=100 μm. Data for the bar graphs and box plots are represented as mean values ± SD, while data for the tumor volume graph are represented as mean values ± standard error of the mean (SEM). *P* values for the bar graphs were calculated with the unpaired, two-tailed Student's *t* test. The significance for the tumor volume graph and box plot were calculated with two-way and one-way ANOVA, respectively. Source data for this figure are provided as a Source Data file.

elevation and generation of mitochondrial ROS that ultimately results in G1 arrest and apoptotic cell death. Notably, we identified PDE4D as a novel ER target gene that, in turn, modulates ER activity in a feedforward loop in endocrine-responsive models (Fig. 9a, left panel). The high responsiveness to SOC or PDE4D inhibition in SOC-sensitive models is abrogated upon acquisition of resistance due to reduced ER

dependency and higher activation of compensatory pathways (here: EGFR) which upregulates PDE4D via c-Jun, therefore reducing PKA activation and the subsequent ROS-mediated DNA damage, leading to transcriptional recovery and cell survival (Fig. 9a, right panel). During resistance, the response to SOC therapies or PDE4D inhibition alone is minimal, potentially due to reduced dependency on ER. However, only

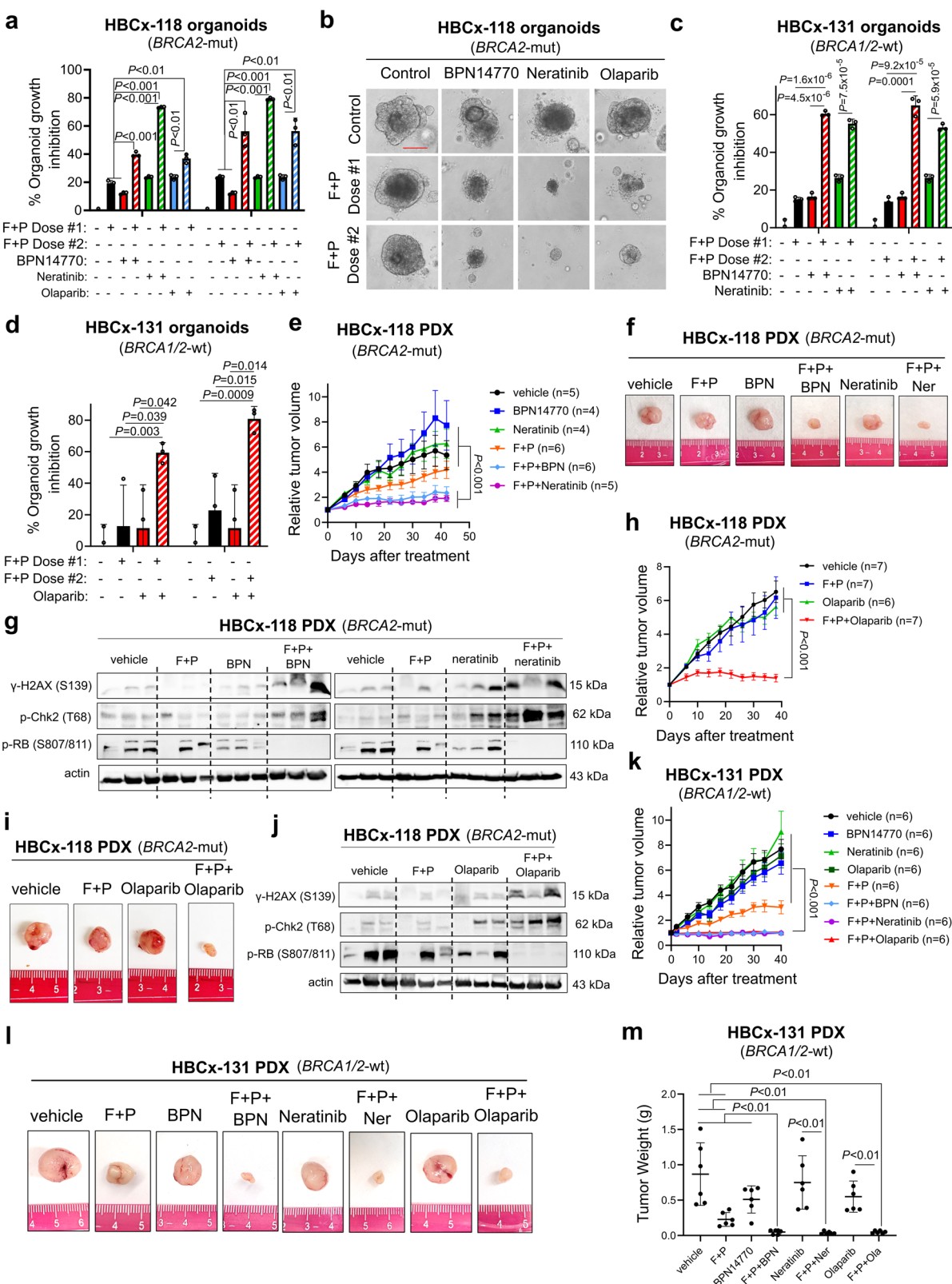

when inhibition of PDE4D or EGFR is combined with SOC therapies or is utilized under E2-deprivation or ER-knockdown, the DNA damage induction and PARP1 trapping are reinstated, thus triggering G1 arrest and apoptosis irrespective of the *BRCA1/2* status. This ultimately leads to reduced tumor growth in multiple models of SOC resistance. Furthermore, using PARP1 inhibitor in combination with SOC as well phenocopied the effects of PDE4D or EGFR inhibition with respect to

SOC sensitization in both *BRCA1/2*-wt and mutant models, validating the key roles of PARP1 trapping in SOC sensitivity (Fig. 9b).

The growing body of evidence has provided some insight into the potential roles of estrogen signaling in modulating DDR, in addition to its canonical functions[44]. Activated ER signaling facilitates DNA repair via direct interaction of ER with the DNA repair proteins, such as DNA-PK[45] and BRCA1[46], or increasing the mRNA expressions of DNA repair

**Fig. 8 | Targeting PDE4D, EGFR or PARP1 using clinically tested or approved inhibitors overcomes resistance to standard-of-care fulvestrant+palbociclib therapy in PDX organoids and in vivo. a** HBCx-118 organoid growth under the combination of fulvestrant and palbociclib (F + P, Dose #1: 2.5 μM fulvestrant and palbociclib; Dose #2: 5 uM fulvestrant and 2.5 μM palbociclib) with the PDE4D inhibitor, BPN-14770 (50 μg/mL) or pan-HER inhibitor, neratinib (0.5 μM), or PARP1 inhibitor, olaparib (5 μM) for a week (*n* = 3 different wells). **b** Representative images from HBCx-118 organoids from a. Scale bar=50 μm. **c** HBCx-131 organoid growth upon the combination of fulvestrant and palbociclib (F + P) with the PDE4D inhibitor, BPN-14770 (50 μg/mL) or pan-HER inhibitor, neratinib (0.5 μM) for a week (*n* = 3 different wells). **d** Percentage organoid growth inhibition of HBCx-131 model treated with fulvestrant (5 μM) + palbociclib (5 μM) in combination with olaparib (10 μM) (*n* = 3 different wells). **e** Tumor growth curves of HBCx-118 PDXs upon treatment with fulvestrant (20 mg/kg, subcutaneous) plus palbociclib (20 mg/kg, oral gavage) in combination with PDE4D inhibitor, BPN14770 (0.75 mg/kg, oral gavage) or the pan-HER inhibitor, neratinib (15 mg/kg, oral gavage) (*n* = 4-6). **f** Tumor pictures at the end of the experiment from e. **g** Western blot analysis of DNA damage and G1 arrest markers in 3 separate HBCx-118 tumors, collected from different mice treated with fulvestrant plus palbociclib with or without BPN14770

or neratinib. **h, i** Tumor growth curves (h) and representative tumor images (i) of HBCx-118 (*BRCA2*-mut) model upon treatment with fulvestrant (F, 25 mg/kg, subcutaneous) + palbociclib (P, 25 mg/kg, p.o.) in combination with PARP inhibitor, olaparib used at a low-to-moderate dose (35 mg/kg) (*n* = 7 different mice for vehicle, F + P and F + P+olaparib, and *n* = 6 different mice for olaparib in h). **j** Western blot analysis of the markers in 3 separate HBCx-118 PDX tumors, collected from different mice treated with the combination of F + P and olaparib. **k, l** Tumor growth curves (**k**) and representative tumor images (**l**) of HBCx-131 (*BRCA1/2*-wt) model upon treatment with fulvestrant (F), 25 mg/kg, subcutaneous + palbociclib (P), 25 mg/kg, p.o. in combination with BPN14770 (0.75 mg/kg, p.o.), neratinib (15 mg/kg, p.o.) or olaparib (35 mg/kg, p.o.) (*n* = 6 different mice for each group in k). **m** Tumor weight measurements of tumors from k (*n* = 6 different tumors). Data for the bar graphs and box plots are represented as mean values ± SD, while data for the tumor volume graph are represented as mean values ± standard error of the mean (SEM). *P*-values for the bar graphs and box plots were calculated with the unpaired, two-tailed Student's t test. Significance for the tumor volume graphs was calculated with two-way ANOVA. Source data for this figure are provided as a Source Data file.

genes, such as *BRCA1*[47], *BRCA2*[48] and *TP53*[49]. However, the questions of whether endocrine therapy or CDK4/6 inhibitors are able to induce DNA damage, alter the chromatin landscape of DNA repair proteins and modify histones and importantly, how these might affect global transcription in ER+ breast cancer cells still remain. We tackled these questions by (1) analyzing gene expression changes upon SOC treatment in an unbiased manner, (2) assessing the effects of SOC on the expression of DNA repair proteins, their chromatin levels and DNA repair functionality and (3) measuring global transcription rates upon treatment with SOC therapies. Based on the data gathered, we propose, for the first time, that SOC therapies used in ER+ breast cancer reduce the expressions of DNA repair proteins, XRCC1, FEN1, BRCA1, BRCA2 and RAD51, caretakers in maintaining genomic integrity. Furthermore, we demonstrated that PARP1 trapping which has so far been majorly associated with PARP1 inhibitors is also one of the hallmarks of SOC treatment that further increases H3S10 ADPR levels while reducing H3K9Ac, thus leading to cell death. PARylation during DNA damage has dual functions; 1) facilitating the recruitment of DNA repair proteins, and 2) inhibiting transcription to avoid further accumulation of DNA damage[50,51]. However, removal of PARP1 from DNA is required both for the resolution of DNA damage repair and also transcriptional re-activation. Therefore, PARP1 trapping on the broken DNA ultimately leads to transcriptional blockage[52]. Along these lines, we demonstrated that SOC-induced PARP1 trapping leads to strong transcriptional suppression that was reversed by BRCA1 over-expression or PARP1 knockdown in sensitive models while in SOC resistance, PARP1 inhibition restores sensitivity. Olaparib is currently being tested in recent clinical trials in combination with palbociclib and fulvestrant in *BRCA1/2* mutation-associated ER+/HER2- metastatic breast cancer (e.g., NCT03685331). Importantly, as we showed that SOC therapy generates a functional BRCAness phenotype in the *BRCA1/2*-wt cells by downregulating BRCA1, BRCA2 and RAD51, PARP inhibitors e.g., olaparib may be tested in combination with SOC therapies not only in the *BRCA1/2*-mut patients, but also in *BRCA1/2*-wt patients.

ROS upon sustained and high levels of cellular stress may lead to DNA damage and ultimately results in the execution of apoptotic cell death[53]. We demonstrated that SOC therapies used in ER+ breast cancer significantly increases mitochondrial and intracellular ROS levels. Furthermore, using the ROS scavenger, NAC reduced SOC-induced DNA damage, clearly indicating that increased ROS is the major source of SOC-induced DNA damage. Induction of ROS has been shown to increase ADP ribosylation which is a substrate of PARP1 enzyme, dependent on functional HPF1[54]. Along these lines, we demonstrated that upon SOC-induced ROS, ADP ribosylation is strongly induced, specifically at H3S10, leading to a reduction of

H3K9Ac and transcriptional blockage. Importantly, unlike existing PARP1 inhibitors that do not trap PARP1 for a prolonged period of time[55], SOC-induced ROS causes sustained H3S10 ADPR levels and loss of H3K9Ac. It was previously reported that ROS levels might correlate with response to PARP1 inhibitors. For instance, a recent study showed that hypoxia-mediated olaparib resistance was associated with lower ROS levels, and induction of ROS has led to olaparib sensitization[56]. Along these lines, we demonstrated that the combination of SOC (inducing ROS levels) with PARP inhibitor promoted strong PARP1 trapping and mediated synergistic growth inhibition.

cAMP is a second messenger that plays crucial roles in many cancer-associated processes, such as cell growth, survival, migration and invasion[57]. An excessive cAMP generation has been shown to cause apoptotic cell death and tumor growth inhibition in vivo in multiple myeloma[58], and in vitro in breast cancer cells as a mechanism of IL-24-induced cell death[59]. PKA is the major downstream of cAMP and is known to phosphorylate mitochondrial proteins and alter their activity[60]. Complex IV, also known as cytochrome c oxidase, is a member of the respiratory chain and is one of the main PKA substrates. PKA-dependent phosphorylation of COXIV subunit I (COXIV-1) at tyrosine 304 inhibits its enzymatic activity[61] and inhibition of COXIV may further lead to ROS generation[62]. Supporting these, we provide a novel link between SOC therapy and DNA damage that involves down-regulation of PDE4D, increase in cAMP and activation of PKA which further phosphorylates and inhibits COXIV, leading to mitochondrial stress and ROS generation. Intriguingly, this axis of ROS generation, DNA damage, PARP1 trapping and transcriptional blockage may potentially be exploited beyond SOC treatment and can be generalized to drugs that can induce cAMP.

PDE4D is a cAMP-degrading phosphodiesterase that is known to have strong anti-inflammatory effect, and targeting PDE4D has been shown to be beneficial for the treatment of respiratory diseases and neurological disorders[63,64]. Higher PDE4D levels has also been shown to be correlated with worse survival in solid tumors, such as lung, prostate, melanoma, ovarian, endometrial, colorectal and gastric cancers[24,25]. Here we identified PDE4D as a novel ER target gene. Most importantly, we showed a regulatory feedforward loop between PDE4D and ER such that while estrogen signaling increases PDE4D expression, PDE4D in turn triggers ER activity via inducing ERK1/2-mediated ER phosphorylation at S118. Likewise, when ER signaling is inhibited by endocrine therapy, PDE4D levels are reduced, initiating the cascade of cAMP/ROS/DNA damage-induced cell death. Interestingly, we showed that palbociclib also reduces ER activity in sensitive cells, similar to endocrine therapy, thus lowering PDE4D expression. These results suggest that PDE4D could be a highly attractive therapeutic target also in early-stage breast cancer given

**a**

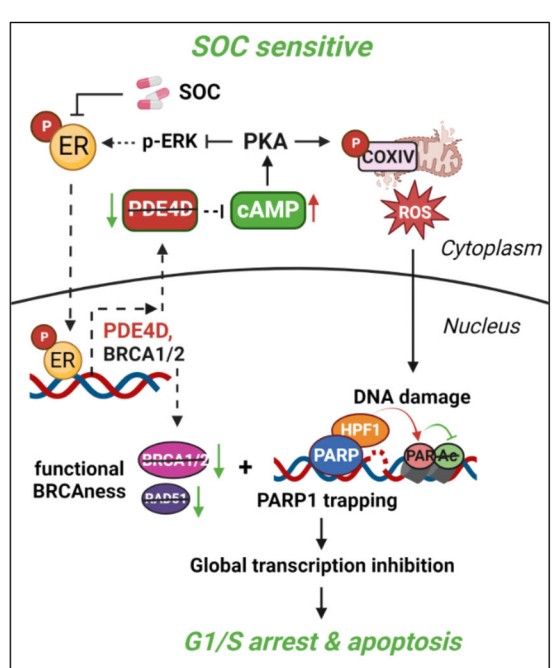
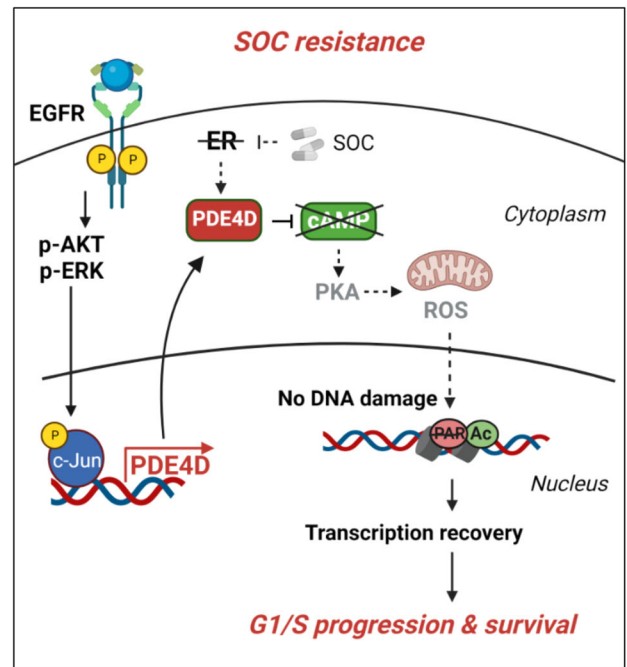

**b**

**Fig. 9 | Schematic summary of the proposed mechanism of SOC sensitivity/ resistance and overcoming resistance via targeting EGFR/PDE4D/PARP1 axis.**
**a** In SOC-sensitive ER+ breast cancer cells, SOC therapies (here: tamoxifen, fulvestrant, and palbociclib) target ER and thereby reducing PDE4D expression, which is a novel ER target gene. Upon reduced PDE4D, cAMP accumulates that further activates PKA, leading to phosphorylation of mitochondrial COXIV subunit I (COXIV-1), mitochondrial stress, generation of reactive oxygen species (ROS) and DNA damage. Meanwhile, PDE4D acts in a feedforward loop with ER, increasing the ER transcriptional activity by inhibiting PKA and activating ERK1/2-mediated ER phosphorylation, leading to the expression of ER targets, including *BRCA1* and *BRCA2*. PDE4D reduction upon SOC treatment, on one hand, facilities the inhibitory effects of SOC on ER, and on the other hand, generates a BRCAness phenotype, rendering the *BRCA1/2*-wt cells sensitive to ROS-induced DNA damage. The SOC-induced DNA damage and BRCAness are followed by PARP1 trapping in complex with its interactor, HPF1. Ultimately, toxic PARP1 accumulation on the chromatin counteracts with transcription shown by increased H3S10 ADPR (PAR) and reduced H3K9Ac (acetylation), leading to blockage of global transcription, induction G1 arrest and apoptosis (left panel). In SOC-resistant cells, the ER to EGFR switch mediates PDE4D transcription via c-Jun, and thereby counteracts the SOC-induced cAMP induction, ROS generation, DNA damage, transcription blockage, G1 arrest and apoptosis (right panel). Dashed lines represent inhibited events. **b** Targeting either PDE4D or EGFR in combination with SOC therapy in SOC-resistant cells restores cAMP induction and the downstream ROS-DNA damage-PARP1 trapping axis, leading to SOC sensitization irrespective of the *BRCA1/2* status. Similarly, targeting PARP1 with PARP inhibitors overcomes SOC resistance, showing the key roles of PARP1 trapping in SOC sensitivity. The figure is created with BioRender.com.

its functional contribution to ER signaling. Furthermore, given the ROS-mediated DNA damage and BRCAness observed upon PDE4D inhibition as single-agent therapy, targeting PDE4D may represent a novel approach that can effectively activate DDR irrespective of BRCA1/2 status.

Beyond the novel roles of PDE4D in SOC sensitivity, we also identified PDE4D as a common mediator of SOC resistance that is overexpressed via c-Jun-mediated transcription as a result of the ER-to-EGFR switch upon acquisition of resistance. Interestingly, the ER-to-EGFR switch was also present in CDK4/6 inhibitor-resistant cells. This may, in part, be explained by the inhibition of ER activity upon palbociclib treatment in sensitive cells which might then activate compensatory pathways, such as EGFR upon long-term treatment. Here, we propose a mechanism underlying c-Jun-mediated cell proliferation in SOC resistance that involves PDE4D as a major transcriptional target of c-Jun. Interestingly, c-Jun has been shown to reprogram ER chromatin binding and ER-dependent transcription in ER+ breast cancer cells[65]. Future studies may investigate whether PDE4D, which we showed to be a novel ER target and a regulator of ER signaling, is also involved in c-Jun-mediated ER reprogramming in estrogen-sensitive cells. It is also yet to be determined whether PDE4D is in a feed-forward loop with EGFR in SOC-resistant cells, similar to the PDE4D-ER loop we identified in sensitive models.

BPN14770 is a selective small molecule allosteric inhibitor of PDE4D that was granted Orphan Drug Designation by the Food and Drug Administration (FDA) upon successful completion of a Phase 2 trial in Fragile X syndrome (FXS). It is currently being tested in a Phase III trial in FXS (NCT05358886). Despite the proven benefits in neurological diseases[66], there has been no studies testing BPN14770 or any other PDE4D inhibitor in the context of cancer treatment. Our PDX data showing the strong efficacy of the combination of fulvestrant/palbociclib cocktail with BPN14770 in endocrine-resistant ER+ PDXs is the first preclinical evidence suggesting a potential use of this highly promising inhibitor in refractory ER+ breast cancer. This is further supported by our patient data showing a strong correlation of PDE4D mRNA and protein expression with endocrine therapy response and patient survival. These data pave the way for clinical testing of BPN14770 with SOC therapies in treatment-refractory patients.

Overall, our results shed light on the roles of SOC therapies used in ER+ breast cancer in modulating DDR and global transcription. Furthermore, we identified the cAMP-induced oxidative DNA damage and PARP1 trapping leading to global transcription inhibition as a key factor in determining response/resistance to SOC therapies. These data further provide preclinical evidence for targeting the cAMP modulator PDE4D or EGFR or PARP1 using clinically available inhibitors as potential strategies to overcome SOC resistance in ER+ breast cancer.

## Methods

This research complies with all relevant ethical regulations, including the Non-Interventional Clinical Research Ethics Committee of Hacettepe University, and Institutional Animal Care and Use Committee of the University of South Carolina and Medical University of South Carolina.

### Cell lines, drugs, and culture conditions

Human breast cancer cell lines, MCF-7 (HTB-22), T47D (HTB-133), ZR-75-30 (CRL-1504), MDA-MB-436 (HTB-130) normal human breast epithelial cell lines MCF-12A (CRL-3598) and MCF-10A (CRL-10317), and normal mouse fibroblast cell line, NIH3T3 (CRL-1658) were purchased from ATCC. The human breast cancer cell line MDA-MB-361 was purchased from Tissue Culture Facility (TCF) Shared Resource of MUSC. PDX primary cells were isolated from the PDX tumors of the models of interest. MCF-7 and T47D cells were cultured in phenol

red–free DMEM (Gibco) with 10% FBS, 0.1% insulin, 50 U/mL penicillin/streptomycin, 1% nonessential amino acids (Gibco). ZR-75-30 and MDA-MB-361 were cultured in RPMI-1640 with 20% FBS, 0.1% insulin, 50 U/mL penicillin/streptomycin, 1% nonessential amino acids (Gibco). NIH3T3 and MDA-MB-436 cells were cultured in DMEM (Gibco) with 10% FBS, 50 U/mL penicillin/streptomycin, 1% nonessential amino acids (Gibco). MCF-12A and MCF-10A cells were cultured in DMEM supplemented with 20 ng/ml epidermal growth factor (EGF) and 500 ng/ml hydrocortisone. Tamoxifen-resistant (TamR) MCF-7 and T47D cells were generated previously[67]. MCF-7 and T47D FulvR and PalboR cells were grown in the presence of 1 μM of fulvestrant (Tocris) and 1 μM of palbociclib (Selleckchem) over 9 months. In parallel, parental MCF-7 and T47D cells were maintained under identical conditions without the drugs. Olaparib, neratinib and ulixertinib were purchased from MedChem Express. Gefitinib was purchased from LCLabs. The PKA inhibitor, Rp-Cyclic AMPS was purchased from Cayman Chemicals. GebR-7b and tamoxifen were purchased from Sigma, and BPN14770 was purchased from Axon MedChem. Cells were routinely tested for mycoplasma contamination using MycoAlert detection kit (Lonza) and were authenticated by STR sequencing. Except parental and resistant cell lines, other cells were cultured for less than 20 passages.

### ER+ human tumor samples

To analyze the effects of PDE4D protein expression on the survival of endocrine therapy-treated female ER+ breast cancer patients, we performed IHC staining of PDE4D in primary tumor samples from 171 ER+ breast cancer patients that were diagnosed between 2000 and 2016 at Hacettepe University School of Medicine, Ankara, Turkey and treated with endocrine therapy with or without radiotherapy and chemotherapy. The study was approved by the Non-Interventional Clinical Research Ethics Committee of Hacettepe University (approval no: 2020/02-40). Informed consent was obtained from all patients.

### Stable transfections using lentiviral vectors

MISSION lentiviral shRNA against PARP1 vectors were purchased from Sigma (TRCN0000007932, TRCN0000007929) (Supplementary Table 2). To generate viral particles carrying shRNA vectors, 6 μg of vectors along with the psPAX2 and pMD2.G packaging plasmids were co-transfected into HEK293FT cells in 6-well plate using lipofectamine (Invitrogen). After 48 hours incubation, the viral particles were collected and transduced into T47D cells. Further selection was done by treating cells with the medium containing 2 μg/mL puromycin for 10 days.

### EdU, Annexin V/DAPI staining and cell cycle assay

EdU staining was done using Click-iT™ EdU Cell Proliferation Imaging Kit (ThermoFisher Scientific) according to manufacturer's instructions. Annexin V/DAPI staining was done as previously described[68,69]. For assessing the effects of SOC on G1 arrest, T47D cells treated with 5 μM of SOC for 24 hours were fixed in ethanol for 30 min on ice and stained with DAPI. For analyzing cell cycle distribution upon SOC treatment at early time points, BrdU/7AAD Flow kit (BD Biosciences) was utilized based on manufacturer's instructions.

### EU staining

EU staining was done using the Click-iT™ RNA Alexa Fluor™ 488 Imaging Kit (ThermoFisher Scientific) according to manufacturer's instructions. Briefly, T47D parental or FulvR cells were treated with fulvestrant or the combination of fulvestrant with GebR-7b, respectively for 12 hours, followed by 1 hour incubation with 1 mM EU in culture media. Then, cells were fixed, permeabilized and the EU was detected by incubation with Click-iT® reaction cocktail for 30 min. Quantification was done using the SpectraMax microplate reader

with an excitation/emission wavelengths of 495/519 nm. The representative images were taken using Nikon Eclipse TS2R fluorescence microscopy.

## Chromatin fractionation

T47D cells were treated with SOC or PARP1 inhibitor for 2 hours. T47D FulvR cells were treated with fulvestrant and GebR-7b combination for 2 hours, or fulvestrant and gefitinib combination for 4 hours. Then, cells were lyzed in the lysis buffer containing 150 mM KCl, 50 mM HEPES pH=7.4, 2.5 mM $MgCl_2$, 5 mM EDTA pH=8, 3 mM DTT, 0.5% Triton X-100, 10% glycerol and protease inhibitor for 45 min on ice. Centrifugation was done at 16,000 g for 15 min to pellet the chromatin. Supernatant was collected as the cytosolic fraction, while the insoluble chromatin was washed once with lysis buffer, resuspended in lysis buffer and then sonicated. Protein concentration was measured using BCA Protein Assay (Thermo Scientific) and equal amounts of protein from cytosol and chromatin factions were mixed with SDS loading dye and boiled at 95 °C for 10 min.

## Generation of PDX-derived organoids

ER+ PDX organoids were established as previously described[69]. For drug testing studies, organoids were dissociated with Tryple (Gibco, MA, USA) at 37 °C for 30 mins in the presence of 10 µM Rock inhibitor (Selleckchem, TX, USA). After counting the single cells, they were plated into 96-well plate (20,000 cells/well) on a matrigel-coated surface with media containing 2% matrigel (Corning, NY, USA). SOC therapies with or without PDE4D or EGFR or PARP1 inhibitors were added 72 hours after seeding. Organoids were grown in the presence of drugs or vehicle for 7 days, and the organoid viability was measured using 3D Cell Titer Glo (Promega, WI, USA).

## In vivo mice experiments

All the in vivo studies were carried out in accordance with the Institutional Animal Care and Use Committee of the University of South Carolina and Medical University of South Carolina. Six-to-eight-week-old female Nu/J mice (Jackson Lab, 002019, homozygous) were housed with a temperature-controlled and 12-hour light/12-hour dark cycle environment and received a standard diet and water ad libitum. The animal facilities maintain centrally controlled and monitored humidity and light cycles. The HBCX-118 (BC1060) or HBCx-131 (BC1101) ER+ PDX tumors that were freshly excised from carrier mice were cut into equal pieces of $2 \times 2 \times 2$ mm to $3 \times 3 \times 3$ mm size. The tumor fragments were immediately transplanted to the neck region of female Nu/J mice. Water was supplemented with 2 ug/ml estradiol until tumors become palpable. After the average tumor volume reached 80-85 mm³, mice were equally distributed to treatment groups with similar mean and median across different groups. Treatments with vehicle, fulvestrant (15-35 mg/kg, weekly, s.c.), palbociclib (15-35 mg/kg, daily, p.o.), neratinib (15 mg/kg, daily, p.o.), olaparib (35 mg/kg, daily, p.o.), BPN14770 (0.75 mg/kg, daily, p.o.) or GebR-7b (3 µg/kg, daily, i.p.) were done for 6 weeks. The estradiol concentration was reduced to 0.5 µg/mL during treatments. The tumor volume was measured twice weekly, and body weight was measured once a week. The ethical tumor size cut-off for both institutions is 20 mm in diameter. Mice were sacrificed 6 weeks after initiation of the treatment, and the tumors were collected and stored for subsequent analysis.

## Bioinformatics analysis

The connectivity map database[13] was used to generate the SOC sensitivity signature. To this end, differentially expressed mRNAs upon treatment of MCF-7 cells with 10 µM of SOC (tamoxifen, fulvestrant, and palbociclib) for 24 hours were downloaded, mRNAs that commonly down- or up-regulated were shortlisted using a z-score cutoff of <−1 or >1, respectively. This yielded 184 commonly downregulated and 53 commonly upregulated genes which altogether constitute the SOC sensitivity signature. z-scores of the genes in the doxorubicin signature obtained from MCF-7 cells treated with 500 nM doxorubicin for 24 hours; etoposide signature obtained from A549 cells treated with 10 µM etoposide for 24 hours, shPDE4D signature obtained from MCF-7 cells transfected for 96 hours and ADCY9 overexpression signature obtained from A375 cells transfected for 96 hours were compared with the SOC sensitivity signature. The neoadjuvant microarray datasets, GSE93204 and GSE87411 and the cohorts of endocrine-treated ER-positive breast cancer patients (GSE124647, GSE81538, GSE202203) were downloaded from the GEO database.

## Statistics and reproducibility

All the results are represented as mean ± standard deviation (SD) or mean ± standard error of the mean (SEM), as indicated in the figure legends. All statistical analyses were performed in GraphPad Prism Software. The significance for survival analysis was done using Log-rank test. Separation of patients in GSE124647 and GSE202203 was done based on median and lower vs. upper 25 percentile *PDE4D* mRNA expression, respectively. In Hacettepe cohort, patients with a PDE4D protein expression at the lowest 25 percentile or higher were classified as low vs. high PDE4D expressers. GSEA analysis was done using the GSEA software and the datasets were downloaded from the Molecular Signatures Database (MSigDB) (https://www.gsea-msigdb.org/gsea/index.jsp). Separation of patients as low vs high PDE4D-expressers for the GSEA analysis was done based on median expression. Comparisons between two groups were done using unpaired two-sided Student's t-test. The comparison of paired tumor samples from pre- and post-treated samples from GSE87411 was done using paired two-sided Student's t-test. Tumor volumes between combination groups vs. single agent or vehicle-treated groups were compared using two-way ANOVA.

All other methods, including transfection, drug treatment, qRT-PCR, ChIP, Western blotting, EGFR array, cAMP and ROS measurements, HR reporter assay, immunofluorescence staining and IHC staining are provided in Supplementary Methods. The Supplementary Tables of sequences of siRNAs, qRT-PCR primers and CHIP qPCR primers (Supplementary Table 3) and list of antibodies (Supplementary Table 4) are also provided in the Supplementary Information file.

## Reporting summary

Further information on research design is available in the Nature Portfolio Reporting Summary linked to this article.

## Data availability

Source data are provided with this paper. Biological materials are available from the corresponding author upon reasonable request. Data presented on Fig. 1a was generated by analyzing the Connectivity Map Database. Data presented on Fig. 2k, l and Supplementary Fig. 4b were generated by analyzing the data available under the accession number GSE93204[22]. Data presented on Fig. 2m-p were generated by analyzing the data from GSE87411[23]. Data presented on Fig. 4e was generated by analyzing the data from GSE81538[70]. Data presented on Fig. 5a, b were generated by analyzing the data from GSE124647[28]. Data presented on Fig. 6d was generated by analyzing the data from GSE81538[70]. Data presented on Supplementary Fig. 4c was generated by analyzing the data from GSE202203[71]. These GEO dataset are available at GEO depository (https://www.ncbi.nlm.nih.gov/geo/). Source data are provided with this paper.

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

## Acknowledgements

We are thankful to the members of Ozgur Sahin laboratory for invaluable discussion and advice. We are also thankful to Dr. David Long (Medical University of South Carolina) for his valuable comments and edits on the manuscript. This work was supported by research funding from American Cancer Society Research Scholar Grant RSG-19-194-01-CSM (O.S.), and in part, from the National Institutes of Health (NIH; R01CA267101 and R01CA251374 to O.S.), SmartState Endowment in Lipidomics and Drug Discovery (O.S.) and Susan G. Komen Interdisciplinary Graduate Training to Eliminate Cancer Disparities (IGniTE-CD) GTDR17500160 (Ozge.S.). The core facilities utilized are supported by NIH (C06 RR015455), Hollings Cancer Center Support Grant (P30 CA138313), or Center of Biomedical Research Excellence (Cobre) in Lipidomics and Pathobiology (P30 GM103339). The Zeiss 880 microscope was funded by a Shared Instrumentation Grant (S10 OD018113).

## Author contributions

Ozge.S. designed and performed experiments, acquired and analyzed data, interpreted data, and prepared the paper; M.C. contributed to organoid generation and ChIP experiments; M.U. performed the IHC staining and evaluation of ER+ patient tissues; U.M.T. contributed to in silico identification of c-Jun and ER binding sites on PDE4D promoter. I.C. performed the pathological analyses and quantification of PDX tumor samples; P.G.E. contributed to PDX experiments; E.M. contributed to PDX data; A.A. contributed to TMA preparation, IHC staining and data interpretation; S.A. provided clinical information of ER+ patients and contributed to data analyses; A.U. performed the IHC staining of ER+ patient tissues and contributed to data interpretation; E.M. provided PDXs and associated clinicopathological data; S.M. contributed to the design of DNA damage mechanistic experiment; O.S. designed the study, oversaw experiments and data analyses, and prepared the paper. All authors reviewed and commented on the paper.

## Competing interests

O. Sahin is the co-founder and manager of OncoCube Therapeutics LLC, developing TACC3 inhibitors, and the founder and president of LoxiGen, Inc, developing lysyl oxidase inhibitors. The other authors declare no potential conflicts of interest.
