## [Peer Review File · Nature Communications]

nature portfolio

Peer Review FileReviewer comments, first round

Reviewer #1 - expertise in PARP inhibition and breast cancer (Remarks to the Author):

The question of how to target treatment resistant ER positive breast cancer is an important one. Generally the data is of good quality and this is a comprehensive study. Much of the data is observational however, meaning that the proposed mechanism is not completely supported.

In particular, the data suggesting that the ER/CDK4/6i therapy effectiveness is driven by "toxic PARP1 trapping and BRCAness" is weak. There is a reduction in levels of BRCA1 protein, but functional "BRCAness" would really be defined by a lack of functional homologous recombination activity. This would usually be assessed by a lack of RAD51 foci - normally in response to ionising radiation, but the levels of damage suggested by Fig 1D may be sufficient to see this without. At the moment the paper does not convincingly show that the cells have a functional BRCA1 deficient phenotype, and the changes in BRCA1 levels could be coincidental (e.g. resulting from cell cycle distribution).

Fig 1G suggests that there is some PARP1 association with chromatin, but again it is not clear if this is really "toxic PARP1 trapping". There is already appreciable PARP1 chromatin localisation without drug (why isn't this toxic trapping?), and the increase could simply be a result of increased damaged DNA PARP1 substrate. Furthermore, there is no data that this is actually what is killing the cells. This could be easily assessed by genetic disruption of PARP1 - if this phenotype is driven by trapping, it should be alleviated by PARP1 loss. It would also help the case if cell survival/viability data for the drug treatments was shown - currently only indirect readouts are shown. The whole paper would be strengthened by showing the original sensitivity and later resistance phenotypes clearly in the cell line models used, using dose-response survival assays. Showing that the SOC treatments are synergistic with olaparib would be another key experiment, as this would support the assertion of BRCAness induction and PARP1-trapping dependent lethality. On the basis of the current data I am not really convinced that the DNA repair aspects have anything to do with the resistance/sensitivity to SOC +/- EGFR/PDE4D inhibitors.

The dependence on PDE4D in resistant cells is more convincing as the effect is rescued by overexpression and mimicked by inhibition. This part of the paper is more well-developed and identifies PDE4D as an ER target gene.

The results of the in vivo experiments are promising. Full error bars should be plotted for tumor measurements and the type of error bars stated. Since the numbers of mice in each group are small, it would be better to show individual mouse measurements. Although body weight data do not show severe toxicity with the PDE4D inhibitor combinations, the PDE4D inhibitors seem to kill most of the cell line models in the paper. BPN14770 at least does not inhibit mouse Pde4d so the relevance of body weight data is limited - what about GebR-7b? From reading the paper and the proposed mechanism, I am not completely clear about the basis of tumour selectivity - is this due to the sustained PDE4D upregulation in the resistant tumours? In which case, wouldn't these be expected to be sensitive to PDE4D inhibition alone? Why is the SOC chemotherapy still necessary, bearing in mind the mechanism suggested in Figure 7? This could be made clearer in the discussion and summary figures.

In summary the in vivo data provides some qualified rationale to pursue PDE4D as a target in resistant breast cancer, and this is backed up by reasonable mechanistic data. The earlier observations regarding DNA repair phenotypes are potentially interesting but lack a clear causal link with the drug response phenotype.

Reviewer #2 - expertise in DNA damage (Remarks to the Author):

Authors present the results of their study which show that in ER+ breast cancer, SOC treatment

with hormonal therapy and CDK4/6 inhibition results in PARP1 trapping, reduced BRCA1 expression, cAMP elevation through PDE4D depletion, generation of mitochondrial reactive oxygen species (ROS) and DNA damage accumulation. In SOC resistance, they show that upstream EGFR signaling activates PDE4D and that targeting PDE4D or EGFR restores the pathways affected by SOC therapy and recapitulates SOC sensitivity.

Major comments:

1. authors correlate effects of SOC on PARP1 trapping, reduced H3K9Ac, and significant global reduction in transcription to BRCAness induced as observed by reduced BRCA1, FEN1, and XRCC1. However, these effects may simply be a result of G1 accumulation due to the effects of blockade of ER-stimulated proliferation and/or inhibition of the CDK4/6 axis (and hence E2F transcriptional activity). Furthermore, many of these effects may also indicate a level of replication stress induced by G1 accumulation (supported by pCHK2). Is there a reduction of DNA repair capacity when BRCA1 protein levels are reduced but cell cycle changes are not apparent (perhaps at 2 or 4 hrs after treatment based on Westerns)? along those same lines is there a synergistic effect if SOC is combined with pharmacologically achievable doses of PARP inhibition?

2. What happens in BRCA mutated ER+ cells?

3. Cells are under charcoal stripped serum conditions, which also deprives the cells of many growth factors that may interact with similar pathways in the opposite manner given the starved condition of cells. Are similar results obtained in normal serum conditions?

4. SOC now combines endocrine therapy with CDK4/6 inhibition, but the combination is not tested.

5. Is PARylation of PARP1 reduced with treatment?

6. The role of EGFR activation of MAPK leading to ligand independent ER activation is not new. Can this explain the role of cAMP/PDE4D results observed by the group?

7. Use of a PDE4D targeting agent to restore estrogen sensitivity is very nice translational work that has potential impact in the clinic.

Overall, while the overall manuscript reports interesting results tying estrogen resistance and cAMP/PDE4D, and an inhibitor of PDE4D restores estrogen sensitivity, the connection to DDR and BRCAness is correlative and may be an indirect effect of cell cycle effects.

Rebuttal Letter for manuscript NCOMMS-23-08866

We thank both reviewers for their insightful comments, and we greatly appreciate that they found the original manuscript interesting, comprehensive and very nice translational work and that it addresses an important question. We have performed substantial additional experimental work, both in vitro and in vivo, to address each and every comment of the reviewers. We believe that addressing the reviewers' comments with additional data supported the mechanisms proposed and has significantly improved the manuscript. We hope that the reviewers now find our revised manuscript suitable for publication in *Nature Communications*.

Reviewer #1 - expertise in PARP inhibition and breast cancer (Remarks to the Author):

The question of how to target treatment resistant ER positive breast cancer is an important one. Generally the data is of good quality and this is a comprehensive study. Much of the data is observational however, meaning that the proposed mechanism is not completely supported.

Response: We thank the reviewer for finding our manuscript significant, our data of good quality and for stating that it is a comprehensive study. To provide additional support for the proposed mechanisms we identified, we performed several lines of in vitro and in vivo experiments. We have now revised the manuscript as explained in detail below. We believe that these additional data overall strengthened the manuscript significantly and strongly support our proposed mechanisms.

In particular, the data suggesting that the ER/CDK4/6i therapy effectiveness is driven by "toxic PARP1 trapping and BRCAness" is weak. There is a reduction in levels of BRCA1 protein, but functional "BRCAness" would really be defined by a lack of functional homologous recombination activity. This would usually be assessed by a lack of RAD51 foci - normally in response to ionising radiation, but the levels of damage suggested by Fig 1D may be sufficient to see this without. At the moment the paper does not convincingly show that the cells have a functional BRCA1 deficient phenotype, and the changes in BRCA1 levels could be coincidental (e.g. resulting from cell cycle distribution).

Response: We thank the reviewer for this highly insightful comment. In the original manuscript, we showed BRCA1 downregulation and DNA damage induction starts 2-4 hours after treatment with SOC (i.e., tamoxifen, fulvestrant, and palbociclib). To demonstrate that BRCA1 downregulation is indeed functional with respect to the loss of DNA repair capacity rather than just being coincidental with cell cycle distribution, we first performed γ -H2AX and RAD51 co-staining in T47D cells upon SOC treatment. As shown in new **Fig. 1c, d** of the revised manuscript and also below, SOC treatment increased the % of γ -H2AX-positive cells without inducing RAD51 foci formation and any significant cell cycle arrest (new **Fig. S1c, d**) at 4 hours of treatment. On the other hand, etoposide, a DNA damaging agent used as a positive control increased RAD51 foci formation as expected^{1,2}. To further validate the causal role of BRCA1 downregulation in terms of reducing DNA repair capacity, we performed a rescue experiment with BRCA1 overexpression followed by SOC treatment. SOC treatment for 4 hours in Ctrl ORF-expressing cells increased the γ -H2AX and p-Chk2 levels (new **Fig. 1e**) without causing G1 or S phase arrest

(new Fig. S1c, d). On the other hand, BRCA1 overexpression under SOC treatment prevented DNA damage induction (new Fig. 1e) without affecting cell cycle distribution (new Fig. S1c, d). Importantly, in addition to reducing BRCA1 levels, SOC treatment also reduced RAD51 expression which was rescued by BRCA1 overexpression (new Fig. 1e). These results altogether suggest a functional BRCAness phenotype upon SOC treatment that was not resulting from cell cycle distribution. These are now provided in Pages 6,7, Lines 137-151 in the revised manuscript.

Figure. BRCA1 downregulation upon SOC treatment leads to lack of functional homologous recombination activity and it precedes G1 arrest. **1c** IF staining of γ -H2AX (S139) (green) and RAD51 (red) in T47D cells upon treatment with SOC for 4 hours. DAPI (blue) was used to stain the nucleus. Etoposide was used as a positive control. Scale bar = 100 μ m. **d** Quantification of the % of γ -H2AX+ cells that are RAD51 low or RAD51 high (n=8 different areas, 400 cells). **1e** Western blot analyses of BRCA1, RAD51, DNA damage markers and H3K9Ac in T47D cells overexpressing ctrl vs. BRCA1 ORF and treated with SOC for 4 hours. Actin is used as a loading control in all Western blots unless stated otherwise. **S1c** Percentage of cells found in G1, S and G2/M phases of the cell cycle upon SOC treatment for 4 hours in the absence or presence of BRCA1 ORF as determined by BrdU/7AAD staining (n=4). **S1d** Dot plots of cells stained with BrdU/7AAD after 4 hours treatment with SOC in the absence or presence of BRCA1 ORF. Error bars correspond to mean values \pm standard deviation (SD). *P*-values were calculated with the unpaired, two-tailed Student's *t* test. * *P*<0.05, ** *P*<0.01.

Fig 1G suggests that there is some PARP1 association with chromatin, but again it is not clear if this is really "toxic PARP1 trapping". There is already appreciable PARP1 chromatin localization without drug (why isn't this toxic trapping?), and the increase could simply be a result of increased damaged DNA PARP1 substrate. Furthermore, there is no data that this is actually what is killing the cells. This could be easily assessed by genetic disruption of PARP1 - if this phenotype is driven by trapping, it should be alleviated by PARP1 loss.

Response: This is a highly relevant and insightful comment by the reviewer. To investigate the functionality of increased PARP1 trapping in terms of induction of cell killing under SOC therapy, we performed several lines of experiments. First, we repeated the chromatin fractionation experiments by increasing the frequency of the washing steps and loading less amount of protein to reduce the background PARP1 signal on the chromatin in control cells. The newer blots are

now provided in new **Fig. 1g** and also below. ADP ribosylation and PARP1-dependent polymer (PAR) formation are hallmarks of increased PARP1 activity³. Upon increased PARP1 levels on the chromatin compared to only a weak binding in control cells, we detected a subsequent increase in ADP ribosylation (new **Fig. 1h**). Furthermore, PARP1 auto-PARylation was also increased upon SOC treatment while the PARP inhibitor, olaparib that is known to reduce PARP1-dependent polymer (PAR) formation upon inhibiting PARP1 activity⁴ reduced the auto-PARylation (new **Fig. 1i**). PARylation during DNA damage has dual functions; 1) facilitating the recruitment of DNA repair factors, and 2) inhibiting transcription to avoid further accumulation of DNA damage⁵⁻⁷. However, removal of PARP1 from DNA is required both for resolution of DNA damage repair and also transcriptional re-activation⁸. Therefore, PARP1 trapping on DNA ultimately leads to transcriptional blockage^{5,7}. Along these lines, we demonstrated that SOC-induced PARP trapping caused transcription inhibition at 4 hours of treatment (new **Fig.S2a**) and sustained over 24 hours (new **Fig. 1j**), coinciding with apoptosis. Mechanistically, we showed increased histone H3 serine ADP-ribosylation of H3S10 (H3S10 ADPr), a modification known to be mutually exclusive with H3K9 acetylation, the marker of active transcription^{9,10}. Increase in H3S10 ADPr is also consistent with increased levels of the PARP1 interactor histone PARylating factor, HPF1 on the chromatin (new **Fig. 1k, Fig. S2a**). This shows that the PARP1/HPF1 complex trapped on damaged DNA triggers H3S10 ADP ribosylation, leading to loss of H3K9Ac and blockage of global transcription. These are now provided in Pages 7, Lines 161-167 in the revised manuscript.

Figure. SOC treatment traps PARP1 on the chromatin along with increased histone ADP-ribosylation and reduced transcriptional activation marker. 1g Chromatin occupancy of PARP1 and HPF1 upon treatment of T47D cells with SOC therapies for 2 hours. Histone H3 was used as the loading control for chromatin fraction, and α -tubulin was used as the loading control for cytosol fraction. **1h** Western blot analysis of ADP ribosylation (ADPr) in T47D cells treated with SOC for 1 hour. **1i** PARP1 immunoprecipitation (IP) in SOC-treated T47D cells followed by immunoblotting for PARylation. **1j** Western blot analysis of H3S10 ADPr, acetylated H3K9 (H3K9Ac), Histone H3 and cleaved PARP in T47D cells treated with SOC for 24 hours. **S2a** Western blot analysis of H3S10 ADPr and H3K9Ac in T47D cells treated with SOC for 4 hours.

To functionally demonstrate that PARP1 trapping upon SOC is indeed toxic, i.e., leading to reduction of cell viability, we silenced PARP1 using both shRNAs or siRNAs. Upon PARP1 silencing with shRNAs or siRNAs, levels of PARP1 as well as its interactor HPF1 on the chromatin were reduced (new **Fig. 1m, Fig. S2b**), along with the restoration of H3K9 acetylation levels (new **Fig. 1n, Fig. S2c**), indicating transcriptional recovery. Furthermore, under PARP1 knockdown, the sensitivity of T47D and MCF-7 cells to SOC therapy was significantly reduced (new **Fig. 1o, S2d, e**). Overall, our data demonstrate that increased PARP1 trapping is not simply be a result of increased damaged DNA but is functionally important to block transcription that ultimately results in cell death under SOC therapy. These are now provided in Pages 7,8, Lines 172-179 in the revised manuscript.

Figure. PARP1 knockdown alleviates SOC-mediated cell death. **1m, S2b** Chromatin occupancy of PARP1 and HPF1 in T47D cells expressing two different sequences of shPARP1 (1m) or an siPARP1 (s2b) and treated with SOC for 2 hours. Histone H3 was used as the loading control for chromatin fraction, and α -tubulin was used as the loading control for cytosol fraction. **1n, S2c** Western blot analysis of H3K9Ac in T47D cells expressing shPARP1 (1n) or siPARP1 (S2c) treated with SOC for 4 hours. **1o** Percentage growth inhibition in T47D cells with shPARP1 and treated with SOC for 72 hours ($n=3$). **S2d, e** Percentage growth inhibition in MCF-7 (d) and T47D (e) cells transfected with siPARP1 and treated with SOC for 72 hours ($n=4$). Error bars correspond to mean values \pm SD. P -values were calculated with the unpaired, two-tailed Student's t test. ** $P < 0.01$.

It would also help the case if cell survival/viability data for the drug treatments was shown - currently only indirect readouts are shown. The whole paper would be strengthened by showing the original sensitivity and later resistance phenotypes clearly in the cell line models used, using dose-response survival assays.

Response: For the synergy experiments where we combined two drugs at multiple doses, we provided the heatmaps of combination indices (CI) which is a direct and quantitative readout for synergy assessment. The calculation of CI was done using the following formula based on the Bliss Independence model¹¹, one of the most popular models: $CI = (EA+EB-EA*EB)/EAB$. Here, EA and EB are the effects of reagents A and B on cell viability alone, while EAB is the inhibitory effect when A and B are combined. The use of heatmaps to represent synergy data is preferred since it directly shows the CI value for each dose pair and allows the assessment of synergy by comparing the CI value with the cut-off for synergy¹² and potentially provide more quantitative and clear visualization of data than dose response curves for multiple dose combination studies. For the revised manuscript, we renewed the CI heatmaps by improving the readability of the numbers and the contrast between colors on the scale. We also provided the color coding and the cut-off for synergy next to every heatmap for easier interpretation. As an example, we provided below one of the relative growth inhibition (a) and updated CI (b) heatmaps as well as the corresponding dose response curve (c) for palbociclib + gefitinib combination in HBCx-118 cells (gefitinib doses: 1, 5, 7.5 and 10 μ M; and palbociclib doses: 0.5, 1, 1.5, 2.5 μ M) (**Fig. 6n, Figure for Reviewer 1** below).

Figure for Reviewer 1. The relative growth inhibition and combination index heatmaps and the corresponding dose response graph for palbociclib + gefitinib combination in HBCx-118 cells. **a, b** Heatmaps of relative growth inhibition (a) and combination indices (b) in HBCx-118 cells treated with increasing doses of palbociclib (0.5, 1, 1.5, 2.5 μ M) and gefitinib (1, 5, 7.5 and 10 μ M). **c** Dose response curves in HBCx-118 cells from a, b. Error bars correspond to mean values \pm SD.

Regarding the reviewer's comment on showing the original sensitivity and later resistance phenotypes, we had indeed provided the dose-response survival analysis for each model in the original manuscript and also provided below (provided in the original manuscript as **Fig. S6a-c**). However, we now mention more explicitly that our MCF-7 and T47D resistant models survive better than the parental counterparts upon treatment with increasing doses of SOC. For example, IC50 comparison of Parental vs Resistant cells are clearly explained: for tamoxifen (IC50 2.8 μ M vs. 8.0 μ M), for fulvestrant (IC50 4.4 μ M vs. 11.5 μ M) and for palbociclib (IC50 0.97 μ M vs. 4.25 μ M) for T47D parental and resistant cells, respectively. Furthermore, we also tested cross-resistance of our resistant models to other SOC therapies, again at multiple doses (provided in

the original manuscript as **Fig. S6d, e** and also provided below) and demonstrated that tamoxifen and fulvestrant resistant models are cross-resistant to each other while they were more sensitive to palbociclib as in the clinics. However, palbociclib resistant models were found to be less responsive to endocrine therapy. These data demonstrate the resistance phenotypes of our models and their clinical relevance. These are now explained better in Page 12, Lines 295-302 in the revised manuscript.

Figure. Dose response curves of parental vs. resistant T47D and MCF-7 cells and their cross-resistance to other SOC therapies. S6a-c Percent cell viability in parental vs. resistant T47D and MCF-7 cells treated with increasing doses of tamoxifen (a), fulvestrant (b) or palbociclib (c) for 72 hours (n=4, 6). S6d, e Percent growth inhibition in parental and resistant T47D (S6d) and MCF-7 (S6e) cells treated with all SOC therapies at increasing doses to assess cross resistance (tamoxifen: 0.01-12.5 μM ; fulvestrant: 0.05-50 μM ; palbociclib: 0.05-20 μM) (n=4, 6). The numbers indicate % growth inhibition which are colored based on the scale bar provided below the figure. Error bars correspond to mean values \pm SD.

Showing that the SOC treatments are synergistic with olaparib would be another key experiment, as this would support the assertion of BRCAness induction and PARP1-trapping dependent lethality. On the basis of the current data I am not really convinced that the DNA repair aspects have anything to do with the resistance/sensitivity to SOC +/- EGFR/PDE4D inhibitors.

Response: We thank the reviewer for suggesting combining SOC with the PARP inhibitor that is highly relevant. To address this question, we did several lines of both in vitro and in vivo experiments supporting our hypothesis. We first demonstrated that PARP inhibition with olaparib overcomes SOC resistance via causing synergistic growth inhibition in all our SOC resistant models i.e., TamR, FulvR and PalboR cells (new Fig. 5o, Fig. S8a). We further demonstrated that combination of SOC with Olaparib reduces BRCA1 and RAD51 levels that was accompanied by DNA damage induction and G1 arrest in these cells (new Fig. 5p). Next, we tested combining SOC with olaparib as well as Gefitinib and gefitinib also in two different BRCA-mutant ER+ cell lines, MDA-MB-361 and ZR-75-30 that we demonstrated to be resistant to SOC (new Fig. S8b).

Combination treatments caused mostly synergistic growth inhibition in MDA-MB-361 (new Fig. S8c, S11h) and ZR-75-30 (new Fig. S8d, S11i) cells, accompanied by induction of DNA damage and G1 arrest upon combination treatment (new Fig. S8e, S11j).

Figure. PARP1 inhibition overcomes SOC resistance in TamR, FulvR and PalboR cells via inducing BRCAness, DNA damage and G1 arrest. 5o, S8a Heatmaps of relative growth inhibition (left) and combination indices (right) in T47D FulvR (5o), T47D TamR and PalboR (S8a) cells treated with the combination of different SOC therapies (fulvestrant: 0.05, 0.1, 0.5, 2.5, 10 μM; tamoxifen: 2, 3, 4, 5, 6 μM; palbociclib: 0.25, 0.5, 1, 2.5, 3.5 μM) with olaparib (5, 7.5 μM). The scale bar for the growth inhibition and combination index matrices are provided at the right-hand side. 5p Western blot analysis of the markers in T47D FulvR cells treated with fulvestrant or olaparib or their combination.

Figure. SOC treatments are synergistic with PDE4D, EGFR and PARP inhibition in BRCA-mut ER+HER2+ MDA-MB-361 cells. S8b Table of IC50 values of SOC therapies in BRCA-mut MDA-MB-361 and ZR-75-30 cells. S8c, 11h Heatmaps of relative growth inhibition (left) and combination indices (right) in MDA-MB-361 cells treated with different SOC therapies (tamoxifen: 0.5, 1, 1.5, 2, 3.5 μM; fulvestrant: 0.5, 1, 2.5, 5, 10 μM; palbociclib: 2.5, 5, 10, 20, 30 μM) and Gefir-7b (50, 75, 100 μg/ml) or olaparib (10, 25, 50 μM) (S8c) or gefitinib (0.1, 0.2, 0.5 μM) (S11h).

S8d

ZR-75-30 (*BRCA*-mut)

S11i

SOC treatments are synergistic with PDE4D, EGFR and PARP inhibition in *BRCA*-mut ER+HER2+ ZR-75-30 cells. S8d, S11i Heatmaps of relative growth inhibition (left) and combination indices (right) in ZR-75-30 cells treated with different SOC therapies (tamoxifen: 2, 3, 4, 5, 7.5 μ M; fulvestrant: 5, 10, 15, 25, 35 μ M; palbociclib: 15, 20, 25, 30, 35 μ M) and GebR-7b (5, 10, 20 μ g/ml) or olaparib (10, 20, 30 μ M) (S8d) or gefitinib (0.1, 0.5, 1 μ M) (S11i).

S8e

S11j

Figure. The combination of SOC therapies with PDE4D, EGFR and PARP inhibitors in *BRCA*-mut cells causes DNA damage, G1 arrest and apoptosis. S8e, S11j Western blot analysis of the markers in ZR-75-30 cells treated with tamoxifen/fulvestrant in combination with GebR-7b, olaparib (S8e) or gefitinib (S11j).

Next, the combination of SOC with olaparib was also tested in primary cell cultures of 2 different PDX models (*BRCA*-mut HBCx-118 and the *BRCA*-WT HBCx-131), and synergistic growth inhibition was observed (new Fig. 5s, Fig. S10a, b). Furthermore, we tested the combination in PDX organoids and observed significant increase in growth inhibition upon PARP inhibition together with SOC (new Fig. 8a, b, d).

Figure. PARP inhibition overcomes SOC resistance in ER+ PDX cells and organoids irrespective of the *BRCA* status. **5s, S10a** Heatmaps of combination indices (5s) and relative growth inhibition (S10a) in HBCx-118 cells treated with the combination of different SOC therapies (tamoxifen: 2, 3, 4, 4.5 μ M; fulvestrant: 1, 5, 10, 20 μ M; palbociclib: 0.5, 1, 1.5, 2.5 μ M) with olaparib (2.5, 5, 7.5, 10 μ M). **S10b** Heatmaps of combination indices (right) and relative growth inhibition (left) in HBCx-131 cells treated with the combination of different SOC therapies (tamoxifen: 2, 3, 4, 4.5 μ M; fulvestrant: 1, 5, 10, 20 μ M; palbociclib: 0.5, 1, 1.5, 2.5 μ M) with olaparib (2.5, 5, 7.5, 10 μ M). **8a, b** HBCx-118 organoid growth (8a) and representative images (8b) under the combination of fulvestrant and palbociclib (F+P, Dose #1: 2.5 μ M fulvestrant and palbociclib; Dose #2: 5 μ M fulvestrant and 2.5 μ M palbociclib) with the PDE4D inhibitor, BPN-14770 (50 μ g/mL) or pan-HER inhibitor, neratinib (0.5 μ M) for a week (n=3). **8d** Percentage organoid growth inhibition of HBCx-131 model treated with fulvestrant (5 μ M) + palbociclib (5 μ M) in combination with olaparib (10 μ M) (n=3). Error bars correspond to mean values \pm SD. P-values were calculated with the unpaired, two-tailed Student's t test. * $P < 0.05$, ** $P < 0.01$

Finally, we tested PARP inhibition in combination with SOC in the HBCx-118 and HBCx-131 PDXs in vivo and observed significantly better tumor growth inhibition in both the *BRCA*1-WT and *BRCA*-mutant PDX tumors (new **Fig. 8h-i**). Western blot analysis revealed induction of DNA damage and G1 arrest upon PARP inhibition with loss of *BRCA*1 and *RAD51* in combination with SOC (new **Fig. 8j**). Overall, our results show that SOC treatments are synergistic with olaparib with the assertion of *BRCAness* induction and PARP1-trapping dependent cell death. These are now provided in Page 13, Lines 316-326 and Page 17, Lines 416-425 in the revised manuscript.

Figure. PARP inhibition overcomes SOC resistance in ER+ PDXs in vivo irrespective of the BRCA status. **8h, i** Tumor growth curves (8h) and representative tumor images (8i) of HBCx-118 (*BRCA*-mutant) model upon treatment with fulvestrant (F, 25 mg/kg, subcutaneous) + palbociclib (P, 25 mg/kg, p.o.) in combination with PARP inhibitor, olaparib (35 mg/kg) (n=6, 7). **8j** Western blot analysis of the markers in HBCx-118 PDX tumors treated with combination of F+P and olaparib. **8k, l** Tumor growth curves (8k) and representative tumor images (8l) of HBCx-131 (*BRCA*-WT) model upon treatment with fulvestrant (F), 25 mg/kg, subcutaneous + palbociclib (P), 25 mg/kg, p.o. in combination with the PARP inhibitor, olaparib (35 mg/kg) (n=6). Error bars correspond to mean values \pm SD. Significance for the tumor volume graphs was calculated with two-way ANOVA. * $P < 0.05$, ** $P < 0.01$.

The dependence on PDE4D in resistant cells is more convincing as the effect is rescued by overexpression and mimicked by inhibition. This part of the paper is more well-developed and identifies PDE4D as an ER target gene.

Response: We thank the reviewer for finding the PDE4D dependence of resistant cells convincing and stating that it is well-developed.

The results of the in vivo experiments are promising. Full error bars should be plotted for tumor measurements and the type of error bars stated. Since the numbers of mice in each group are small, it would be better to show individual mouse measurements.

Response: As suggested by the reviewer, we have plotted the full error bars with the type of error bars stated in the legends (standard error of mean, SEM) for the in vivo graphs (**Fig. 7d, 8e, h, k**). Please see Page 40, Lines 1016-1017 and Page 41, Lines 1042-1043 in the revised manuscript. Furthermore, we now provide the actual tumor volumes in mm³ in the **Supplementary Figure S12a, b, f, g** of the revised manuscript and also provide the raw data of tumor volumes for each individual mouse in the raw data file.

Figure. Relative tumor volumes (left) and absolute tumor volumes in mm³ (right) from the in vivo experiments shown in the manuscript. Error bars correspond to mean values \pm standard error of the mean (SEM). Significance for the tumor volume graphs was calculated with two-way ANOVA. ** $P < 0.01$.

Although body weight data do not show severe toxicity with the PDE4D inhibitor combinations, the PDE4D inhibitors seem to kill most of the cell line models in the paper. BPN14770 at least

does not inhibit mouse Pde4d so the relevance of body weight data is limited - what about GebR-7b? From reading the paper and the proposed mechanism, I am not completely clear about the basis of tumour selectivity - is this due to the sustained PDE4D upregulation in the resistant tumours? In which case, wouldn't these be expected to be sensitive to PDE4D inhibition alone? Why is the SOC chemotherapy still necessary, bearing in mind the mechanism suggested in Figure 7? This could be made clearer in the discussion and summary figures.

Response: We thank the reviewer for bringing up the potential toxicity issue. As the reviewer stated, BPN14770 displays around 15-fold less potency against mouse PDE4D compared to human PDE4D¹³, whereas GebR-7b inhibits both human and mouse PDE4D¹⁴. To test whether PDE4D inhibition with GebR-7b or BPN14770 causes toxicity to normal cells, we treated the normal mouse cell line, NIH3T3, and two normal human cells, MCF10A and MCF12A with increasing doses of GebR-7b and BPN14770. As shown in new Fig. S5a-c, there was only minor growth inhibition in normal cells upon PDE4D inhibition with any of the inhibitors as compared to ER-positive cancer cells (Fig. 3i, j). These data suggest that normal cells do not rely on PDE4D for cell growth.

Figure. Testing potential cytotoxicity of PDE4D inhibition in normal mouse and human cells. S5a-c Percentage growth inhibition in normal mouse NIH3T3 (S5a) and normal human ER- MCF10A (S5b) and MCF12A (S5c) cells treated with PDE4D inhibitor, GebR-7b and BPN14770. **3i, j** Percentage growth inhibition in ER+ T47D (3i) and MCF-7 (3j) cells treated with PDE4D inhibitor, GebR-7b. Error bars correspond to mean values \pm SD. P-values were calculated with the unpaired, two-tailed Student's t test. ** $P < 0.01$

For in vivo testing of toxicity, we had already provided body weight changes as a measure of toxicity. As shown in **Supplementary Fig. 13a-d**, we have not detected any significant body weight change upon combination of SOC with GebR-7b, BPN14770, neratinib or olaparib. There was also no significant body weight change in BPN14770 or GebR-7b-treated mice compared to vehicle. To further assess the potential in vivo toxicity of PDE4D inhibition, we performed blood cell counting in the latest in vivo experiment we performed using HBCx-131 PDXs over 40 days treatment with SOC in combination with BPN14770, neratinib or olaparib, and observed no significant change in neither combination nor single agent treatment groups as compared to vehicle (new **Fig. S13e-g**). Importantly, the clinical trial testing BPN14770 in Fragile X-syndrome successfully met the primary end point of BPN14770 safety and tolerability. No major treatment-emergent adverse events (TEAEs) were reported except vomiting and upper respiratory tract infection, with no meaningful differences between the treatment arms¹⁵. There was one serious adverse event (SAE), which is severe septic olecranon bursitis, but it was assessed as an intercurrent illness unrelated to BPN14770 usage. There were overall no changes in electrocardiograms, and also no trigger of suicidal thoughts or clinically notable self-injurious behaviors. These altogether suggests that PDE4D inhibition is safe and tolerable as single agent as well as in combination with SOC therapy. These are now provided in Page 10, Lines 236-239, Page 16, Lines 402-403 and Page 17, Lines 411-412 and Lines 415-425 in the revised manuscript.

Figure. Body weight change and blood cell counts in ER+ PDXs under treatment with SOC therapies in combination with PDE4D, EGFR or PARP inhibitors. S13a Percentage body weight change in HBCx-118 PDXs upon treatment with fulvestrant (35 mg/kg, subcutaneous) or palbociclib (35 mg/kg, oral gavage) in combination with PDE4D inhibitor, GebR-7b (3 µg/kg, intraperitoneal) (n = 5-8). **S13b** Percentage body weight change in HBCx-118 PDXs upon treatment with fulvestrant (F, 20 mg/kg, subcutaneous) plus palbociclib (P, 20 mg/kg, oral gavage) in combination with PDE4D inhibitor, BPN14770 (0.75 mg/kg, oral gavage) or the pan-HER inhibitor, neratinib (15 mg/kg, oral gavage) (n = 4-6). **S13c** Percentage body weight change in HBCx-118 PDXs upon treatment with fulvestrant (F, 25 mg/kg, subcutaneous) + palbociclib (P, 25 mg/kg, oral gavage) in combination with olaparib (35 mg/kg, oral gavage) (n=6,7). **S13d** Percentage body weight change in HBCx-131 PDXs upon treatment with fulvestrant (F, 15 mg/kg, subcutaneous) + palbociclib (P, 15 mg/kg, oral gavage) in combination with BPN14770 (0.75 mg/kg, p.o.), neratinib (15 mg/kg, p.o.) or olaparib (35 mg/kg, p.o.) (n=6). **S13e-g** Blood counts from mice in S13d at the end of treatment. Error bars correspond to mean values ±SD. *P*-values were calculated with one-way ANOVA. n.s., not significant.

Regarding the comment of the reviewer about the sensitivity/dependence of parental and resistant cells to PDE4D inhibition alone, we now provide a better discussion. We showed for the first time that PDE4D is an ER target gene and it regulates ER signaling in a feedforward loop. Given the key role of PDE4D in regulating ER signaling, the endocrine-sensitive ER+ cells are also sensitive to PDE4D inhibition compared to ER- normal cells (e.g., MCF10A or MCF12A) as shown above. Upon SOC treatment, PDE4D is downregulated as a result of reduced ER signaling that further contributes to SOC mechanism of action in a feed-forward loop, leading to cell death. In other words, downregulation of PDE4D is a novel mechanism of sensitivity to SOC in sensitive cells. The high responsiveness to SOC or PDE4D inhibition in SOC sensitive models is abrogated upon acquisition of resistance due to reduced ER dependency and higher activation of compensatory pathways, (here: EGFR) which upregulates PDE4D via c-Jun, therefore reducing PKA activation and the subsequent ROS-mediated DNA damage, leading to transcriptional recovery and cell survival. During resistance, the response to SOC therapies or PDE4D inhibition alone is minimal, potentially due to reduced dependency on ER. However, only when inhibition of PDE4D or EGFR is combined with SOC therapies or is utilized under E2-deprivation or ER-knockdown, the DNA damage induction, BRCAness, PARP1 trapping, G1 arrest and apoptosis are reinstated that ultimately leads to reduced tumor growth in multiple models of SOC resistance. Furthermore, targeting PARP1 as well phenocopied the effects of PDE4D or EGFR inhibition with respect to SOC sensitization, validating the key roles of PARP1 trapping in SOC sensitivity. This is now provided in the Discussion section (Page 18, Lines 443-453 in the revised manuscript) when describing the summary figure.

In summary the in vivo data provides some qualified rationale to pursue PDE4D as a target in resistant breast cancer, and this is backed up by reasonable mechanistic data. The earlier observations regarding DNA repair phenotypes are potentially interesting but lack a clear causal link with the drug response phenotype.

Response: We thank the reviewer for this statement and also for the suggestions which improved the manuscript substantially, especially for causally linking the DNA damage and drug response. Importantly, we are now in the process of translating our findings to patients. We hope that we could address all the points of the reviewer adequately.

Reviewer #2 - expertise in DNA damage (Remarks to the Author):

Authors present the results of their study which show that in ER+ breast cancer, SOC treatment with hormonal therapy and CDK4/6 inhibition results in PARP1 trapping, reduced BRCA1 expression, cAMP elevation through PDE4D depletion, generation of mitochondrial reactive oxygen species (ROS) and DNA damage accumulation. In SOC resistance, they show that upstream EGFR signaling activates PDE4D and that targeting PDE4D or EGFR restores the pathways affected by SOC therapy and recapitulates SOC sensitivity.

Response: We thank the reviewer for evaluating our manuscript and providing constructive comments which significantly improved our manuscript.

Major comments:

1. authors correlate effects of SOC on PARP1 trapping, reduced H3K9Ac, and significant global reduction in transcription to BRCAness induced as observed by reduced BRCA1, FEN1, and XRCC1. However, these effects may simply be a result of G1 accumulation due to the effects of blockade of ER-stimulated proliferation and/or inhibition of the CDK4/6 axis (and hence E2F transcriptional activity). Furthermore, many of these effects may also indicate a level of replication stress induced by G1 accumulation (supported by pCHK2). Is there a reduction of DNA repair capacity when BRCA1 protein levels are reduced but cell cycle changes are not apparent (perhaps at 2 or 4 hrs after treatment based on Westerns)? along those same lines is there a synergistic effect if SOC is combined with pharmacologically achievable doses of PARP inhibition?

Response: We thank the reviewer for these highly insightful comments. In the original manuscript, we showed BRCA1 downregulation and DNA damage induction start 2-4 hours after treatment with SOC (i.e., tamoxifen, fulvestrant, and palbociclib). As the reviewer also pointed out, at these early time points, no change in cell cycle is expected. To test whether there is a reduction of DNA repair capacity when BRCA1 protein levels are reduced at 4 hours without any change in cell cycle distribution, we first performed γ -H2AX and RAD51 co-staining in T47D cells upon 4 hours of SOC treatment. As shown in new **Fig. 1c, d** of the revised manuscript and also below, SOC treatment increased the % of γ -H2AX-positive cells without inducing RAD51 foci formation and any significant cell cycle arrest (new **Fig. S1c, d**) at 4 hours of treatment. On the other hand, etoposide, a DNA damaging agent used as a positive control increased RAD51 foci formation as expected^{1,2}. To further validate the causal role of BRCA1 downregulation in terms of reducing DNA repair capacity, we performed a rescue experiment with BRCA1 overexpression followed by SOC treatment. SOC treatment for 4 hours in Ctrl ORF-expressing cells increased the γ -H2AX and p-Chk2 levels (new **Fig. 1f**) without causing G1 or S phase arrest (new **Fig. S1c, d**). On the other hand, BRCA1 overexpression under SOC treatment prevented DNA damage induction (new **Fig. 1f**) without affecting cell cycle distribution (new **Fig. S1c, d**). Importantly, in addition to reducing BRCA1 levels, SOC treatment also reduced RAD51 expression which was rescued by BRCA1 overexpression (new **Fig. 1g**). These results altogether suggest that there is a functional reduction of DNA repair capacity upon downregulation of BRCA1 before cell cycle changes are not apparent. These are now provided in Pages 6,7, Lines 137-151 in the revised manuscript.

Figure. BRCA1 downregulation upon SOC treatment leads to lack of functional homologous recombination activity and it precedes G1 arrest. **1c** IF staining of γ -H2AX (S139) (green) and RAD51 (red) in T47D cells upon treatment with SOC for 4 hours. DAPI (blue) was used to stain the nucleus. Etoposide was used as a positive control. Scale bar = 100 μ m. **d** Quantification of the % of γ -H2AX+ cells that are RAD51 low or RAD51 high (n=8 different areas, 400 cells). **1e** Western blot analyses of BRCA1, RAD51, DNA damage markers and H3K9Ac in T47D cells overexpressing ctrl vs. BRCA1 ORF and treated with SOC for 4 hours. Actin is used as a loading control in all Western blots unless stated otherwise. **S1c** Percentage of cells found in G1, S and G2/M phases of the cell cycle upon SOC treatment for 4 hours in the absence or presence of BRCA1 ORF as determined by BrdU/7AAD staining (n=4). **S1d** Dot plots of cells stained with BrdU/7AAD after 4 hours treatment with SOC in the absence or presence of BRCA1 ORF. Error bars correspond to mean values \pm standard deviation (SD). *P*-values were calculated with the unpaired, two-tailed Student's *t* test. * *P*<0.05, ** *P*<0.01.

Regarding the question of the reviewer on whether there is a synergistic effect if SOC is combined with pharmacologically achievable doses of PARP inhibition, we did several lines of both in vitro and in vivo experiments. First, we demonstrated that PARP inhibition with olaparib at doses ranging from 2.5 to 7.5 μ M (the plasma concentration of olaparib is 2.5 μ M and it is given twice daily) overcomes SOC resistance via causing synergistic growth inhibition in our TamR, FulvR and PalboR cells (new **Fig. 5o**, **Fig. S8a**). We further demonstrated that combination of SOC with olaparib reduces BRCA1 and RAD51 levels that was accompanied by DNA damage induction and G1 arrest (new **Fig. 5p**). The combination of SOC with olaparib was also tested in primary cell cultures of 2 different PDX models (the *BRCA*-WT HBCx-131 and *BRCA*-mut HBCx-118), and synergistic growth inhibition was observed (new **Fig. 5s**, **Fig. S10a, b**). Furthermore, we tested the combination in PDX organoids and observed significant increase in growth inhibition upon PARP1 inhibition together with SOC (new **Fig. 8a, b, d**).

Figure. PARP1 inhibition overcomes SOC resistance in TamR, FulvR and PalboR cells via inducing BRCAness, DNA damage and G1 arrest. **5o**, **S8a** Heatmaps of relative growth inhibition (left) and combination indices (right) in T47D FulvR (5o), T47D TamR and PalboR (S8a) cells treated with the combination of different SOC therapies (fulvestrant: 0.05, 0.1, 0.5, 2.5, 10 μ M; tamoxifen: 2, 3, 4, 5, 6 μ M; palbociclib: 0.25, 0.5, 1, 2.5, 3.5 μ M) with olaparib (5, 7.5 μ M). The scale bar for the growth inhibition and combination index matrices are provided at the right-hand side. **5p** Western blot analysis of the markers in T47D FulvR cells treated with fulvestrant or olaparib or their combination.

Figure. PARP inhibition overcomes SOC resistance in ER+ PDX cells and organoids irrespective of the BRCA status. **5s**, **S10a** Heatmaps of combination indices (5s) and relative growth inhibition (S10a) in HBCx-118 cells treated with the combination of different SOC therapies (tamoxifen: 2, 3, 4, 4.5 μ M; fulvestrant: 1, 5, 10, 20 μ M; palbociclib: 0.5, 1, 1.5, 2.5 μ M) with olaparib (2.5, 5, 7.5, 10 μ M). **S10b** Heatmaps of combination indices (right) and relative growth inhibition (left) in HBCx-131 cells treated with the combination of different SOC therapies (tamoxifen: 2, 3, 4, 4.5 μ M; fulvestrant: 1, 5, 10, 20 μ M; palbociclib: 0.5, 1, 1.5, 2.5 μ M) with olaparib (2.5, 5, 7.5, 10 μ M). **8a** HBCx-118 organoid growth under the combination of fulvestrant and palbociclib (F+P, Dose #1: 2.5 μ M fulvestrant and palbociclib; Dose #2: 5 μ M fulvestrant and 2.5 μ M palbociclib) with the PDE4D inhibitor, BPN-14770 (50 μ g/mL) or pan-HER inhibitor, neratinib (0.5 μ M) for a week (n=3). **8b** Percentage organoid growth inhibition of HBCx-131 model treated with fulvestrant (5 μ M) + palbociclib (5 μ M) in combination with olaparib (10 μ M) (n=3). Error bars correspond to mean values \pm SD. P-values were calculated with the unpaired, two-tailed Student's t test. * $P < 0.05$. ** $P < 0.01$

Finally, we tested PARP inhibition in combination with SOC in vivo. Combining SOC with olaparib in the *BRCA*-mut HBCx-118 PDXs completely blocked tumor growth (new Fig. 8h, i) and triggered DNA damage induction and G1 arrest upon loss of *BRCA*1 and *RAD*51 expressions (new Fig. 8j). Furthermore, combination of SOC with olaparib also effectively blocked tumor growth in the *BRCA*-WT HBCx-131 PDXs (new Fig. 8k, l). Combination therapy did not increase body weight loss compared to SOC alone (new Fig. S13c, d). Combination therapy or any single agent treatment also did not cause any significant change in the blood counts (new Fig. S13g). These are now provided in Page 13, Lines 316-326 and Page 17, Lines 416-425 in the revised manuscript.

Figure. PARP inhibition overcomes SOC resistance in ER+ PDXs in vivo irrespective of the *BRCA* status. **8h, i** Tumor growth curves (8h) and representative tumor images (8i) of HBCx-118 (*BRCA*-mutant) model upon treatment with fulvestrant (F, 25 mg/kg, subcutaneous) + palbociclib (P, 25 mg/kg, p.o.) in combination with PARP inhibitor, olaparib (35 mg/kg) (n=6, 7). **8j** Western blot analysis of the markers in HBCx-118 PDX tumors treated with combination of F+P and olaparib. **8k, l** Tumor growth curves (8k) and representative tumor images (8l) of HBCx-131 (*BRCA*-WT) model upon treatment with fulvestrant (F), 25 mg/kg, subcutaneous + palbociclib (P), 25 mg/kg, p.o. in combination with the PARP inhibitor, olaparib (35 mg/kg) (n=6). Error bars correspond to mean values \pm SD. Significance for the tumor volume graphs was calculated with two-way ANOVA. * $P < 0.05$, ** $P < 0.01$.

Figure. Body weight change and blood cell counts in ER+ PDXs under treatment with SOC therapies in combination with PARP inhibitor. S13c Percentage body weight change in HBCx-118 PDXs upon treatment with fulvestrant (F, 25 mg/kg, subcutaneous) + palbociclib (P, 25 mg/kg, oral gavage) in combination with olaparib (35 mg/kg, oral gavage) (n=6,7). **S13d** Percentage body weight change in HBCx-131 PDXs upon treatment with fulvestrant (F, 15 mg/kg, subcutaneous) + palbociclib (P, 15 mg/kg, oral gavage) in combination with BPN14770 (0.75 mg/kg, p.o.), neratinib (15 mg/kg, p.o.) or olaparib (35 mg/kg, p.o.) (n=6). **S13g** Blood counts from mice in S13d at the end of treatment. Error bars correspond to mean values ±SD. P-values were calculated with one-way ANOVA. n.s., not significant.

2. What happens in BRCA mutated ER+ cells?

Response: We tested the combination of SOC with BPN14770 or gefitinib or olaparib in two different *BRCA*-mut ER+/HER2+ cell lines, MDA-MB-361 and ZR-75-30. We showed that these cells are resistant to SOC therapies (new Fig. S8b). Similar to our *BRCA*-WT models, combination therapies caused mostly a synergistic growth inhibition in MDA-MB-361 (new Fig. S8c, S11h) and ZR-75-30 (new Fig. S8d, S11i) cells. At molecular level, combination therapy induced DNA damage, G1 arrest and apoptosis via reducing *BRCA*1 and *RAD*51 expression in the *BRCA*-mut cells similar to the results we obtained in *BRCA*-WT cell lines (new Fig. S8e, S11j). These data suggest that combination of SOC with EGFR or PDE4D or PARP inhibition is effective irrespective of the *BRCA* status. These are now provided in Page 13, Lines 316-326 in the revised manuscript.

S8b

IC50 (μM)	MDA-MB-361	ZR-75-30
Tamoxifen	10.7	10.5
Fulvestrant	>50	>50
Palbociclib	>20	>20

S8c

MDA-MB-361 (*BRCA*-mut)

S11h

Figure. SOC treatments are synergistic with PDE4D, EGFR and PARP inhibition in *BRCA*-mut ER+HER2+ MDA-MB-361 cells. S8b Table of IC50 values of SOC therapies in *BRCA*-mut MDA-MB-361 and ZR-75-30 cells. S8c, 11h Heatmaps of relative growth inhibition (left) and combination indices (right) in MDA-MB-361 cells treated with different SOC therapies (tamoxifen: 0.5, 1, 1.5, 2, 3.5 μM ; fulvestrant: 0.5, 1, 2.5, 5, 10 μM ; palbociclib: 2.5, 5, 10, 20, 30 μM) and GebR-7b (50, 75, 100 $\mu\text{g/ml}$) or olaparib (10, 25, 50 μM) (S8c) or gefitinib (0.1, 0.2, 0.5 μM) (S11h).

S8d

ZR-75-30 (*BRCA*-mut)

S11i

SOC treatments are synergistic with PDE4D, EGFR and PARP inhibition in *BRCA*-mut ER+HER2+ ZR-75-30 cells. S8d, S11i Heatmaps of relative growth inhibition (left) and combination indices (right) in ZR-75-30 cells treated with different SOC therapies (tamoxifen: 2, 3, 4, 5, 7.5 μ M; fulvestrant: 5, 10, 15, 25, 35 μ M; palbociclib: 15, 20, 25, 30, 35 μ M) and GebR-7b (5, 10, 20 μ g/ml) or olaparib (10, 20, 30 μ M) (S8d) or gefitinib (0.1, 0.5, 1 μ M) (S11i).

S8e

S11j

Figure. The combination of SOC therapies with PDE4D, EGFR and PARP inhibitors in *BRCA*-mut cells causes DNA damage, G1 arrest and apoptosis. S8e, S11j Western blot analysis of the markers in ZR-75-30 cells treated with tamoxifen/fulvestrant in combination with GebR-7b, olaparib (S8e) or gefitinib (S11j).

3. Cells are under charcoal stripped serum conditions, which also deprives the cells of many growth factors that may interact with similar pathways in the opposite manner given the starved condition of cells. Are similar results obtained in normal serum conditions?

Response: We thank the reviewer for bringing up this question. We would like to clarify that except in experiments where we studied the characteristics of parental vs. resistant cells in terms of dependency on ER signaling, we used regular media with normal FBS (10% FBS). However, in order to test whether resistant cells grow in an E2-independent but PDE4D/EGFR-dependent manner, it was necessary to deplete the growth media of E2 which can be achieved by using charcoal-stripped FBS which is widely accepted method for E2-depletion¹⁶. In experiments where we used charcoal-stripped FBS in SOC resistant cells and then perform treatment with PDE4D and EGFR inhibitors (**Fig. 6a-c, Supplementary Fig. 11f**), we mimicked the conditions for our combination therapy of SOC (ER inhibition) and PDE4D/EGFR inhibitors. We now better explain the motivation of these experiments in the manuscript. Please see Page 15, Lines 358-361 in the revised manuscript. However, as the reviewer also pointed out, using charcoal-stripped FBS might have impacts on other pathways. To rule out these potential effects, we silenced ER to mimic E2-deprivation. As shown in new **Fig. S11b**, while the parental cells exhibit significant growth inhibition upon loss of ER, resistant cells continue growing, similar to results obtained with charcoal-stripped FBS (**Fig. S11a** in the revised manuscript). Furthermore, the ER-depleted resistant cells, a condition mimicking SOC treatment, exhibit significantly more growth inhibition under treatment with PDE4D or EGFR inhibitors (new **Fig. S11e**), further demonstrating the effectiveness of our combination therapy. These results are also highly similar to those obtained with charcoal-stripped FBS (**Fig. S11f** in the revised manuscript). Therefore, we conclude that similar data to those obtained under charcoal-stripped FBS conditions can be obtained under normal serum conditions upon depletion of ER with siRNAs. These are now provided in Page 14-15, Lines 351-361 in the revised manuscript.

Figure. Dependency on ER is reduced in SOC resistance and SOC sensitivity is restored upon inhibiting PDE4D or EGFR under ER knockdown. S11b Percentage growth inhibition in T47D parental vs. SOC resistant cells upon ER knockdown with siRNA for 72 hours. Western blot validation of ER knockdown is shown on the graph (n=4). **S11e** Percentage growth inhibition in SOC resistant cells upon ER knockdown with siRNA and under treatment with GebR-7b, BPN14770 or neratinib (n=4). Error bars correspond to mean values \pm SD. P-values were calculated with the unpaired, two-tailed Student's t test. ** $P < 0.01$

4. SOC now combines endocrine therapy with CDK4/6 inhibition, but the combination is not tested.

Response: In the original manuscript and now in **Figure 8** of the revised manuscript, we indeed have tested the combination of endocrine therapy (fulvestrant (F)) and CDK4/6 inhibitor (palbociclib (P)) in organoids as well as in vivo. Our results show that targeting PDE4D or EGFR or PARP1 significantly improves the efficacy of F+P therapy which is now SOC both in organoids and in vivo PDXs.

5. Is PARylation of PARP1 reduced with treatment?

Response: This is a highly relevant question by the reviewer. Before analyzing PARP1 auto-PARylation upon SOC treatment, we first analyzed ADP ribosylation. ADP ribosylation and PARP1-dependent polymer (PAR) formation are hallmarks of increased PARP1 activity³. Upon increased PARP1 levels on the chromatin compared to only a weak binding in control cells (new **Fig. 1g**), we detected a subsequent increase in ADP ribosylation (new **Fig. 1h**). Subsequently, PARP1 auto-PARylation was also increased upon SOC treatment while the PARP inhibitor, olaparib that is known to reduce PARP-1-dependent polymer (PAR) formation upon inhibiting PARP1 activity⁴ reduced the auto-PARylation (new **Fig. 1i**). PARylation during DNA damage has dual functions; 1) facilitating the recruitment of DNA repair factors, and 2) inhibiting transcription to avoid further accumulation of DNA damage^{6,7}. However, removal of PARP1 from DNA is required both for resolution of DNA damage repair and also transcriptional re-activation⁸. Therefore, PARP1 trapping on DNA ultimately leads to transcriptional blockage^{5,7}. Along these lines, we demonstrated that SOC-induced PARP trapping caused transcriptional blockage at 4 hours of treatment (new **Fig.S2a**) and sustained over 24 hours (new **Fig. 1j**). Mechanistically, we showed increased histone H3 serine ADP-ribosylation of H3S10 (H3S10 ADPr), a modification known to be mutually exclusive with H3K9 acetylation, the marker of active transcription^{9,10}. Increase in H3S10 ADPr is also consistent with increased levels of the PARP1 interactor histone PARylating factor (HPF1) on the chromatin (new **Fig. 1k, Fig. S2a**). This shows that the PARP1/HPF1 complex trapped on damaged DNA triggers H3S10 ADP ribosylation, leading to loss of H3K9Ac and blockage of global transcription. These are now provided in Page 7, Lines 161-167 in the revised manuscript.

Figure. SOC treatment traps PARP1 on the chromatin along with increased histone ADP-ribosylation and reduced transcriptional activation marker. 1g Chromatin occupancy of PARP1 and HPF1 upon treatment of T47D cells with SOC therapies for 2 hours. Histone H3 was used as the loading control for chromatin fraction, and α -tubulin was used as the loading control for cytosol fraction. **1h** Western blot analysis of ADP ribosylation (ADPr) in T47D cells treated with SOC for 1 hour. **1i** PARP1 immunoprecipitation (IP) in SOC-treated T47D cells followed by immunoblotting for PARYlation. **1j** Western blot analysis of H3S10 ADPr, acetylated H3K9 (H3K9Ac), Histone H3 and cleaved PARP in T47D cells treated with SOC for 24 hours. **S2a** Western blot analysis of H3S10 ADPr and H3K9Ac in T47D cells treated with SOC for 4 hours.

To further test the functionality of trapped PARP1 and the subsequent parylation and loss of acetylation on cell death under SOC treatment, we silenced PARP1 using both siRNAs or shRNAs. Upon PARP silencing with siRNAs or shRNAs, PARP1 levels on the chromatin were reduced (new **Fig. 1m, Fig. S2b**), along with the restoration of H3K9 acetylation levels (new **Fig. 1n, Fig. S2c**), indicating transcriptional recovery. Furthermore, under PARP1 knockdown, the sensitivity of T47D and MCF-7 cells to SOC therapy was significantly reduced (new **Fig. 1o, p**). Overall, our data demonstrate that increased PARP1 trapping is not simply a result of increased damaged DNA but is functionally important to block transcription that ultimately results in cell death under SOC therapy. These are now provided in Pages, 7 and 8, Lines 172-179 in the revised manuscript.

Figure. PARP1 knockdown alleviates SOC-mediated cell death. **1m, S2b** Chromatin occupancy of PARP1 and HPF1 in T47D cells expressing two different sequences of shPARP1 (1m) or an siPARP1 (s2b) and treated with SOC for 2 hours. Histone H3 was used as the loading control for chromatin fraction, and α -tubulin was used as the loading control for cytosol fraction. **1n, S2c** Western blot analysis of H3K9Ac in T47D cells expressing shPARP1 (1n) or siPARP1 (S2c) treated with SOC for 4 hours. **1o** Percentage growth inhibition in T47D cells with shPARP1 and treated with SOC for 72 hours (n=3). **S2d, e** Percentage growth inhibition in MCF-7 (d) and T47D (e) cells transfected with siPARP1 and treated with SOC for 72 hours (n=4). Error bars correspond to mean values \pm SD. *P*-values were calculated with the unpaired, two-tailed Student's *t* test. ** $P < 0.01$.

6. The role of EGFR activation of MAPK leading to ligand-independent ER activation is not new. Can this explain the role of cAMP/PDE4D results observed by the group?

Response: Although the role of EGFR/MAPK signaling on ligand-independent ER activation is known¹⁷, the role of PDE4D/cAMP axis on ERK1/2-mediated ER phosphorylation in the presence of ER ligand, estradiol has not been tested before. In the original manuscript, we showed that

ERK1/2 activation is the downstream of PDE4D/cAMP that activates ER. For the revised manuscript, we performed a rescue experiment where we pretreated the cells with the PKA inhibitor for 1 hour, followed by PDE4D inhibition and then stimulation with E2. As shown below, PKA inhibition resulted in rescue of ERK1/2 phosphorylation and the subsequent ER phosphorylation (new Fig. 4j). These data suggest that the PKA deactivation by PDE4D (due to decreased cAMP levels) potentially drives ERK1/2 activation and ER phosphorylation that ultimately results in enhanced ER transcriptional activity. These are now provided in Page 11, Lines 272-274 in the revised manuscript.

Figure. PKA inhibition rescues ERK activation and ER phosphorylation. 4j. Western blot analysis of p-ER, ER and p-ERK in MCF-7 cells stimulated with E2 in the presence of PKA inhibitor with or without GebR-7b.

7. Use of a PDE4D targeting agent to restore estrogen sensitivity is very nice translational work that has potential impact in the clinic.

Response: We thank the reviewer for this encouraging statement. Indeed, we are now in the process of translating our findings to patients.

Overall, while the overall manuscript reports interesting results tying estrogen resistance and cAMP/PDE4D, and an inhibitor of PDE4D restores estrogen sensitivity, the connection to DDR and BRCAness is correlative and may be an indirect effect of cell cycle effects.

Response: We thank the reviewer for finding our manuscript interesting. The newly added data provided above based on both reviewer's comments provide a considerable mechanistic support for the proposed mechanisms in relation to the causal connection of SOC sensitivity to BRCAness and DDR. We hope that the reviewer now finds our manuscript suitable for publication.

REFERENCES:

- 1 Sun, J. et al. ATM modulates the loading of recombination proteins onto a chromosomal translocation breakpoint hotspot. *PLoS One*.**5**, e13554 (2010).
- 2 Turchick, A., Hegan, D. C., Jensen, R. B. & Glazer, P. M. A cell-penetrating antibody inhibits human RAD51 via direct binding. *Nucleic Acids Res.***45**, 11782-11799 (2017).
- 3 Luo, X. & Kraus, W. L. On PAR with PARP: cellular stress signaling through poly(ADP-ribose) and PARP-1. *Genes Dev.***26**, 417-432 (2012).
- 4 Min, A. & Im, S. A. PARP Inhibitors as Therapeutics: Beyond Modulation of PARylation. *Cancers (Basel)*.**12**, (2020).
- 5 Adamowicz, M. et al. XRCC1 protects transcription from toxic PARP1 activity during DNA base excision repair. *Nat Cell Biol.***23**, 1287-1298 (2021).
- 6 Awwad, S. W., Abu-Zhayia, E. R., Guttman-Raviv, N. & Ayoub, N. NELF-E is recruited to DNA double-strand break sites to promote transcriptional repression and repair. *EMBO Rep.***18**, 745-764 (2017).
- 7 Chou, D. M. et al. A chromatin localization screen reveals poly (ADP ribose)-regulated recruitment of the repressive polycomb and NuRD complexes to sites of DNA damage. *Proc Natl Acad Sci U S A.***107**, 18475-18480 (2010).
- 8 Ko, H. L. & Ren, E. C. Functional Aspects of PARP1 in DNA Repair and Transcription. *Biomolecules*.**2**, 524-548 (2012).
- 9 Bartlett, E. et al. Interplay of Histone Marks with Serine ADP-Ribosylation. *Cell Rep.***24**, 3488-3502 e3485 (2018).
- 10 Liszczak, G., Diehl, K. L., Dann, G. P. & Muir, T. W. Acetylation blocks DNA damage-induced chromatin ADP-ribosylation. *Nat Chem Biol.***14**, 837-840 (2018).
- 11 Fouquier, J. & Guedj, M. Analysis of drug combinations: current methodological landscape. *Pharmacol Res Perspect.***3**, e00149 (2015).
- 12 Flobak, A. et al. A high-throughput drug combination screen of targeted small molecule inhibitors in cancer cell lines. *Sci Data.***6**, 237 (2019).
- 13 Zhang, C. et al. Memory enhancing effects of BPN14770, an allosteric inhibitor of phosphodiesterase-4D, in wild-type and humanized mice. *Neuropsychopharmacology.***43**, 2299-2309 (2018).
- 14 Bruno, O. et al. GEPR-7b, a novel PDE4D selective inhibitor that improves memory in rodents at non-emetic doses. *Br J Pharmacol.***164**, 2054-2063 (2011).
- 15 Berry-Kravis, E. M. et al. Inhibition of phosphodiesterase-4D in adults with fragile X syndrome: a randomized, placebo-controlled, phase 2 clinical trial. *Nat Med.***27**, 862-870 (2021).
- 16 Vanetti, C., Bifari, F., Vicentini, L. M. & Cattaneo, M. G. Fatty acids rather than hormones restore in vitro angiogenesis in human male and female endothelial cells cultured in charcoal-stripped serum. *PLoS One.***12**, e0189528 (2017).
- 17 El-Tanani, M. K. & Green, C. D. Interaction between estradiol and growth factors in the regulation of specific gene expression in MCF-7 human breast cancer cells. *J Steroid Biochem Mol Biol.***60**, 269-276 (1997).

Reviewer comments, second round

Reviewer #1 (Remarks to the Author):

The DNA repair side of the manuscript has been somewhat strengthened by the new data. The data in new Fig 1c suggest that there is lower engagement of RAD51 in the palbo/fulvestrant/tamoxifen exposed cells. It's unclear to me why this doesn't appear as discrete foci, but perhaps this is related to the doses used. PARP1 knockdown also seems to rescue some of the growth inhibition, suggesting that PARP1 is required for the toxicity of the treatments. I would prefer to see these data plotted as dose-response curves and carried on for longer periods, to get down to low levels of surviving cells - in all the assays shown in Fig 1o, for example, most cells are still alive.

I have some further concerns about the models in light of some of the new data. The BRCA status of the cell lines and PDOs used needs to be stated more clearly in order to interpret these new data. Concentrations of olaparib used are quite high, and there is no activity of olaparib alone in the PDX despite the BRCA2 mutation. It's not sufficient to describe these just as BRCA-mut. The actual mutation should be given and ideally some functional assessment of HR status (e.g. RAD51 foci) provided. This also goes for the BRCAm cell lines used. If the mechanism of action is via BRCA1 and RAD51 as the authors suggest, I am not sure that it follows that this would operate in a BRCA2m background (where RAD51 loading is already defective), unless this is through a non-HR related function of BRCA1. Otherwise we might expect epistasis. It might be that these models are not really functionally BRCA defective, but this is important to know either way.

Reviewer #2 (Remarks to the Author):

Authors have performed additional experiments and addressed my comments satisfactorily.

Rebuttal Letter for manuscript NCOMMS-23-08866A

We thank both reviewers for evaluating our revised manuscript in light of the new data we provided. We also greatly appreciate the additional comments from Reviewer #1 to help us further improve our manuscript. Based on the additional points raised by the reviewer, we now made clarification on the characteristics of our cell line and PDX models in terms of their *BRCA1/2* status and homologous recombination (HR) activity by providing new data. We also further clarify our proposed mechanism of action. We hope that our revised manuscript is now suitable for publication in *Nature Communications*.

Reviewer #1 (Remarks to the Author):

The DNA repair side of the manuscript has been somewhat strengthened by the new data. The data in new Fig 1c suggest that there is lower engagement of RAD51 in the palbo/fulvestrant/tamoxifen exposed cells. It's unclear to me why this doesn't appear as discrete foci, but perhaps this is related to the doses used.

We thank the reviewer for finding our manuscript strengthened compared to the original version upon addition of new data. To address the issue of lack of discrete foci formation of γ -H2AX and RAD51 in Fig. 1c, we tested whether it is a dose-related problem as the reviewer suggested. We firstly tested the homologous recombination (HR) proficiency of the *BRCA1/2*-wild type (wt) T47D cells. To this end, we performed the HR reporter assay using the HR Assay Kit from Norgen Biotek (catalog no: 35600) ^{1,2} along with the RAD51 foci formation assay upon treatment with etoposide. The HR reporter assay is based on co-transfecting the cells with two different plasmids containing defective lacZ α cassettes. The two plasmids form a functional lacZ- α cassette only when they undergo homologous recombination. The amount of the HR product is then determined upon DNA isolation followed by qRT-PCR using the specific assay primers that only amplifies the HR product provided by the kit. As shown below and in new **Supplementary Fig. S1e**, etoposide activated HR in a dose-dependent manner. Similar to the HR reporter assay, etoposide also induced RAD51 foci formation in the T47D cells (**Supplementary Fig. S1f**), altogether demonstrating the HR proficiency of the T47D cells bearing wt *BRCA1/2*.

Figure. Etoposide-induced HR activation and RAD51 foci formation in T47D cells with wild-type *BRCA1/2*. **S8e** The relative homologous recombination (HR) activity in T47D (*BRCA1/2*-wt) cells upon treatment with increasing doses of etoposide (n=2, 3). **S8f** IF staining of γ -H2AX (S139) (green) and RAD51 (red) in T47D cells upon treatment with 10 μ M of etoposide for 4 hours. The quantification of γ -H2AX positive cells and those that are also RAD51 positive is given at the right panel (n=4 different areas, with at least 100 cells per area). DAPI (blue) was used to stain the nucleus. Error bars correspond to mean values \pm SD. *P*-values are calculated with the unpaired, two-tailed Student's *t* test.

Next, to test whether the lack of discrete foci formation is a dose-related issue, we treated T47D cells also with a high dose of etoposide (50 μM). Notably, as the reviewer suggested, the foci formation was more clearly visible at low dose (10 μM) while the staining follows a diffuse pattern as the dose increases to 50 μM (**Figure for Reviewer**). Therefore, in the RAD51 foci formation assays in new **Fig. 1c**, **Supplementary Fig S1f**, **S8e**, **h**, **k** and **S11f**, **g**, **i**, **j**, we used 10 μM of etoposide.

Figure. RAD51 foci formation upon treatment with increasing doses of etoposide in T47D cells with wild type *BRCA1/2*. IF staining of $\gamma\text{-H2AX}$ (S139) (green) and RAD51 (red) in T47D cells upon treatment with 10 or 50 μM of etoposide for 4 hours. The quantification of $\gamma\text{-H2AX}$ positive cells and those that are also RAD51 positive is given at the right panel ($n=4$ different areas, with at least 100 cells per area). DAPI (blue) was used to stain the nucleus. Error bars correspond to mean values $\pm\text{SD}$. P -values are calculated with the unpaired, two-tailed Student's t test.

Based on these results, to show the foci formation more clearly in all drug-treated groups (both SOC and etoposide), we repeated the experiment in **Fig. 1c, d** using low doses of etoposide, i.e., 10 μM . We also lowered the exposure to be able to better visualize discrete foci. As shown below and in new **Fig. 1c, d**, RAD51 foci formation is now clearer in the lower dose etoposide-treated cells. Furthermore, the $\gamma\text{-H2AX}$ foci formation is also clearer now in all SOC-treated groups. In short, we now resolved the issue of lack of discrete foci formation for $\gamma\text{-H2AX}$ and RAD51 and integrated these new data to the revised manuscript (Page 6; Lines: 142, 145-147).

Figure. γ -H2AX and RAD51 foci formation in T47D cells upon SOC treatment. **1c** IF staining of γ -H2AX (S139) (green) and RAD51 (red) in T47D cells (*BRCA1/2*-wt) upon treatment with SOC for 4 hours. Etoposide was used as a positive control. DAPI (blue) was used to stain the nucleus. **1d** The quantification of γ -H2AX positive cells and those that are also RAD51 positive (*n*=4 different areas, with at least 100 cells per area). Error bars correspond to mean values \pm SD. *P*-values were calculated with the unpaired, two-tailed Student's *t* test. n.s., not significant (*P*>0.05).

PARP1 knockdown also seems to rescue some of the growth inhibition, suggesting that PARP1 is required for the toxicity of the treatments. I would prefer to see these data plotted as dose-response curves and carried on for longer periods, to get down to low levels of surviving cells - in all the assays shown in Fig 1o, for example, most cells are still alive.

As suggested by the reviewer, we repeated the experiment in Fig. 1o by using a wider dose range and a longer treatment time (5 days) to be able to achieve higher growth inhibition (i.e., lower surviving cells). We now also provide dose-response curves instead of bar graphs as requested by the reviewer. As shown below, and in new Fig. 1o, PARP1 knockdown leads to a significant increase in the IC₅₀ values of the SOC therapies, clearly suggesting that PARP1 significantly contributes to SOC-mediated toxicity. These data are now integrated to the revised manuscript (Page 8; Lines: 184-186).

1o
Figure. PARP1 knockdown mitigates SOC-mediated toxicity. 1o Percentage growth inhibition in T47D cells with shPARP1 and treated with increasing doses of SOC for 5 days (n=4). P-values were calculated with the paired, two-tailed Student's t test.

I have some further concerns about the models in light of some of the new data. The BRCA status of the cell lines and PDOs used needs to be stated more clearly in order to interpret these new data. Concentrations of olaparib used are quite high, and there is no activity of olaparib alone in the PDX despite the BRCA2 mutation. It's not sufficient to describe these just as BRCA-mut. The actual mutation should be given and ideally some functional assessment of HR status (e.g. RAD51 foci) provided. This also goes for the BRCAm cell lines used.

Firstly, we apologize for our oversight for not providing the *BRCA1/2* mutational status of the cell line models and PDXs in detail and not showing the HR status of the models. We now prepared separate tables for each *BRCA*-mut cell line and PDX models that we utilized, showing the type of mutations in the *BRCA1* or *BRCA2* genes and their functional consequence in terms of HR activity as tested by both qRT-PCR-based HR reporter assay as well as the RAD51 foci formation assay (**Supplementary Fig. S8c, f, i, 11b**). To determine the *BRCA1/2* mutations in our cell line models and PDXs, we utilized the CCLE dataset³ and published literature. In ZR-75-30 cell line, there is a missense mutation in the *BRCA2* gene (c.6966G>T) leading to the p.Met2322Ile change in the protein³. In MDA-MB-361 cells, there is a missense mutation in the *BRCA2* gene (c.4970A>G) leading to the p.Asn1657Ser change in the protein³. There is no mutation in the *BRCA1* gene in neither of the cell lines. We also prepared a table for the MDA-MB-436 cells that bear *BRCA1* mutation and is known to be HR-defective⁴, as we used this cell line as a control in our experiments. In the HBCx-118 PDX model that we use, there is a frameshift mutation in the *BRCA2* gene leading the p.C419fs change in the protein⁵. The HBCx-131 and HBCx-139 PDXs bear wt *BRCA1* and *BRCA2* genes. These aberrations are summarized below and in **Supplementary Fig. S8c, f, i, 11b**.

S8c

ZR-75-30	BRCA1	BRCA2
Mutation	None	p.Met2322Ile
Type of mutation	None	Missense
HR activity	Impaired	

S8f

MDA-MB-361	BRCA1	BRCA2
Mutation	None	p.Asn1657Ser
Type of mutation	None	Missense
HR activity	Impaired	

S8i

MDA-MB-436	BRCA1	BRCA2
Mutation	p.Glu1731del28	None
Type of mutation	deletion	None
HR activity	Impaired	

S11b

HBCx-118	BRCA1	BRCA2
Mutation	None	p.C419fs
Type of mutation	None	frameshift
HR activity	Impaired	

Figure. The *BRCA1/2* mutations in the cell line and PDX models used and their effect on homologous recombination (HR) activity. S8c, f, i, S11b. Tables of the *BRCA1* and *BRCA2* mutations, their types, and the HR activity in ZR-75-30 (S8c), MDA-MB-361 (S8f), MDA-MB-436 (S8i) and HBCx-118 (S11b) cells.

To test the functional importance of these *BRCA1* or *BRCA2* mutations in terms of HR activity, we performed the qRT-PCR-based HR reporter assay as well as the RAD51 foci formation assay upon etoposide treatment of our *BRCA2*-mut cell lines. As a result of the *BRCA2* mutations that are outlined in new **Supplementary Fig. S8c, f**, the ZR-75-30 and MDA-MB-361 cells exhibit impaired HR activity tested at two different doses of etoposide (**Supplementary Fig. S8d, g**) and also failed to form RAD51 foci upon etoposide treatment (**Supplementary Fig. S8e, h**) unlike T47D cells which were able to activate HR (**Supplementary Fig. S1e**) and recruit RAD51 to the damaged DNA (**Supplementary Fig. S1f**). The MDA-MB-436 cells bearing *BRCA1* mutation and with defective HR also failed to activate HR and induce RAD51 engagement, similar to our *BRCA2*-mut cell lines (**Supplementary Fig. 8j, k**).

Figure. Characterization of the *BRCA2*-mut SOC resistant cell line models in terms of homologous recombination (HR) proficiency. S8d, S8g, S1e, S8j The relative homologous recombination (HR) activity in ZR-75-30 (*BRCA2*-mut) (S8d), MDA-MB-361 (*BRCA2*-mut) (S8g), T47D (*BRCA1/2*-wt) (S1e) and MDA-MB-436 (*BRCA1*-mut) (S8j) cells upon treatment with increasing doses of etoposide for 12 hours ($n=3$). **S8e, S8h, S1f, S8k** IF staining of γH2AX (S139) (green) and RAD51 (red) in ZR-75-30 (*BRCA2*-mut) (S8e), MDA-MB-361 (*BRCA2*-mut) (S8h), T47D (*BRCA1/2*-wt) (S1f) and MDA-MB-436 (*BRCA1*-mut) (S8k) cells upon treatment with etoposide for 4 hours. The quantification of γH2AX positive cells and those that are also RAD51 positive ($n=4$ different areas, with at least 100 cells per area). DAPI (blue) was used to stain the nucleus. Error bars correspond to mean values \pm SD. P -values were calculated with the unpaired, two-tailed Student's t test. n.s., not significant ($P > 0.05$).

Similarly, etoposide failed to activate HR and induce RAD51 engagement as we demonstrated by the qRT-PCR-based HR reporter assay (**Supplementary Fig. 11e**) and the RAD51 staining (**Supplementary Fig. 11f, g**), respectively in our *BRCA2*-mut HBCx-118 PDX cells. In contrast, our *BRCA1/2*-wt HBCx-131 PDX cells were able to activate HR and form RAD51 foci as shown below and in new **Supplementary Fig. 11h-j**.

Figure. Characterization of the *BRCA2*-mut SOC resistant PDX cells in terms of homologous recombination (HR) proficiency. **S11e, h** The relative homologous recombination (HR) activity in the *BRCA2*-mut HBCx-118 (S11e) and *BRCA1/2*-wt HBCx-131 (S11h) cells upon treatment with etoposide for 12 hours (n=3). **S11f, i** IF staining of γ -H2AX (S139) (green) and RAD51 (red) in HBCx-118 (*BRCA2*-mut) (S11f) and *BRCA1/2*-wt HBCx-131 (S11i) cells upon treatment with etoposide for 4 hours. **S11g, j** The quantification of γ -H2AX positive cells and those that are also RAD51 positive in HBCx-118 (S11g) and HBCx-131 (S11j) PDXs (n=4 different areas, with at least 100 cells per area). DAPI (blue) was used to stain the nucleus. Error bars correspond to mean values \pm SD. *P*-values were calculated with the unpaired, two-tailed Student's *t* test. n.s., not significant ($P > 0.05$).

Next, to demonstrate that the low-level RAD51 engagement and HR activity are indeed functional in terms of increased olaparib response, we tested the olaparib response of the *BRCA2*-mut ZR-75-30 cells, *BRCA1/2*-wt T47D cells and the *BRCA1*-mut MDA-MB-436 cells, one representative cell line from each *BRCA1/2* status. In the original manuscript, in the synergy experiments (as shown below and in **Supplementary Fig. S9b**), we used sublethal olaparib doses up to 30 μ M for ZR-75-30 to observe the synergistic effects of combination therapies on growth inhibition better and obtained a maximum 32% growth inhibition at the highest dose, i.e., 30 μ M. For the revised manuscript, we treated the cells at higher, lethal doses (50 and 75 μ M), together with the *BRCA1/2*-wt T47D and *BRCA1*-mut MDA-MB-436 cells treated with the same doses as controls. As shown below and in new **Supplementary Fig. S10**, olaparib treatment reduced the viability of ZR-75-30 cells (50% and 67% at 50 and 75 μ M, respectively) at a very similar extent to MDA-MB-436 cells, a known olaparib-responsive cell line⁶. On the contrary, the *BRCA1/2*-wt T47D cells respond to olaparib a much lesser extent (13% and 22% at 50 and 75 μ M, respectively) as expected (**Supplementary Fig. S10**).

For the PDX cells, in the synergy experiments (**Supplementary Fig. S12a, b**), we again used sublethal olaparib doses up to 10 μM in the *BRCA2*-mut HBCx-118 and *BRCA1/2*-wt HBCx-131 cells, respectively. For the revised manuscript, we again isolated primary cells from both PDXs and treated both cell lines at 10 and 25 μM . As a result, we observed a stronger growth inhibition in the *BRCA2*-mut HBCx-118 cells (25% and 50% growth inhibition) compared to HBCx-131 cells with wt *BRCA1/2* (5% and 25% growth inhibition), validating the functionality of its *BRCA2* mutation (**Supplementary Fig. S11k**). These are now integrated to the manuscript (Pages 13, 14; Lines: 329, 331-336, 345, 351-355).

Figure. Olaparib dose response in *BRCA1/2*-wt vs. mutant cell lines and PDX cells. **S10** Percentage cell viability in T47D (*BRCA1/2*-wt), MDA-MB-436 (*BRCA1*-mut) and ZR-75-30 (*BRCA2*-mut) cells upon treatment with olaparib for 3 days (n=3, 4). **S11k** Percentage cell viability in HBCx-131 (*BRCA1/2*-wt) and HBCx-118 (*BRCA2*-mut) cells upon treatment with olaparib for 3 days (n=4). Error bars correspond to mean values \pm SD. *P*-values were calculated with the unpaired, two-tailed Student's *t* test.

To address the comment of the reviewer on the lack of in vivo response to olaparib in the *BRCA2*-mut HBCx-118 PDXs, we now provide our rationale for choosing a dose of 35 mg/kg of olaparib clearer in the revised manuscript. Olaparib induces tumor growth inhibition (TGI) ranging from 30% to 90% when used at a dose range of 50 to 100 mg/kg and with a frequency of daily to twice daily administration in *BRCA1/2*-mut models in vivo⁷⁻¹⁰. Based on these doses and frequency of administration used in literature, we chose a dose of 35 mg/kg, which is a relatively low-to-moderate dose. This is because we would like to observe the effect of the combination of SOC (i.e., fulvestrant+palbociclib) and olaparib on tumor growth which necessitates us to use doses that will not have a profound effect when used alone as single-agent therapies. This would then allow us to test whether the combination of 2 or more drugs would synergize and cause an even greater tumor growth inhibition without causing toxicity. Indeed, the TGI that we achieved in the HBCx-118 PDXs upon daily treatment with 35 mg/kg olaparib alone, at the end of 40 days was 14%, while the combination of fulvestrant+palbociclib and olaparib caused a TGI of 78%. We now write our motivation for choosing a low-to-moderate dose explicitly in the revised manuscript (Pages 17, 18; Lines: 436-438, 444-445). Furthermore, we would like to point out that this PDX model is generated from the bone metastasis of a patient who has prior exposure to endocrine therapy and chemotherapy, including the DNA damaging agents, epirubicin and cyclophosphamide which might cause a certain level of de novo resistance to olaparib. Overall, we believe that we addressed the concerns of the reviewer on the lack of details of the *BRCA1/2* status and HR proficiency of the cell lines/PDXs being used by performing several lines of experiments to characterize our models. We also explained our rationale for deciding on the olaparib dose used in in vivo experiments.

If the mechanism of action is via BRCA1 and RAD51 as the authors suggest, I am not sure that it follows that this would operate in a BRCA2m background (where RAD51 loading is already defective), unless this is through a non-HR related function of BRCA1. Otherwise we might expect epistasis. It might be that these models are not really functionally BRCA defective, but this is important to know either way.

We apologize for not explaining our proposed mechanism clear enough. We now provide better and more explicit description of the proposed mechanism upon rephrasing the key sections of the manuscript, and we also provide more experimental support for our proposed mechanism as summarized below. Our main proposed mechanism of SOC sensitivity involves DNA damage and toxic PARP1 trapping upon ROS induction as a result of the cAMP-PKA-mediated mitochondrial stress. The SOC-mediated DNA damage is accompanied by generation of a functional BRCAness phenotype which enables cytotoxic cell death in the *BRCA1/2*-wt cells which would otherwise activate homologous recombination to repair the DNA damage. The proficiency of the T47D cells to activate HR upon DNA damage is now validated in the revised manuscript using HR reporter assay and RAD51 foci formation upon etoposide treatment. Although T47D is an HR proficient cell line, SOC therapies fail to induce HR in T47D cells, indicative of the generation of a functional BRCAness phenotype upon SOC treatment. In the original manuscript, we analyzed BRCA1 expression upon SOC or combination treatments in sensitive and resistant T47D cells, respectively and observed a strong downregulation along with a decrease in RAD51, suggesting BRCAness. As BRCAness involves both BRCA1 and BRCA2, in the revised manuscript, we also analyzed the BRCA2 expression and demonstrated that it is also strongly downregulated in cells with wt *BRCA1/2* upon treatment with SOC (**Fig. 1f**) or with combination therapies (**Fig. 5j, 6l, 5p**) in sensitive and SOC resistant cells, respectively. BRCA2 expression is also reduced upon PDE4D inhibition in sensitive T47D cells with wt *BRCA1/2* (**Fig. 3h**). Downregulation of both BRCA1 and BRCA2 likely prevents any potential compensation or an epistatic interaction between the two partners.

Figure. Western blot analysis of DNA damage, DNA repair, G1 arrest and apoptosis markers in SOC-treated sensitive cells or combination-treated SOC resistant cells. 1f, 3h, 5j, 6l, 5p Western blot analysis of the markers in SOC-treated T47D cells (1f), GebR-7b-treated T47D cells (3h), in T47D FulvR cells treated with the combination of SOC and GebR (5j), SOC and gefitinib (6l) or SOC and olaparib (5p).

In the revised manuscript, we also characterized our *BRCA2*-mut SOC resistant cell lines and PDXs in terms of HR activity and demonstrated that they are HR deficient using HR reporter assay and RAD51 foci formation. Given the lack of an HR activity in these cells, the downregulation of BRCA1 or RAD51 protein expressions in the *BRCA2*-mut settings upon combination treatments is likely an indicator of reduced ER activity as *BRCA1*, *BRCA2* and *RAD51* are all estradiol-induced genes¹¹. Therefore, in the revised manuscript, we removed the BRCA1 and RAD51 blots from the *BRCA2*-mut cell line and PDX Western blotting experiments (Fig. 8g, j, Supplementary Fig. S9c, d, 13j) to alleviate further confusion.

Overall, we now (1) provide a comprehensive characterization of our *BRCA1/2*-wt and *BRCA2*-mut cell line and PDX models in terms of the *BRCA1/2* mutations and HR activity; (2) examined both BRCA1 and BRCA2 proteins upon SOC treatment and PDE4D inhibition; and (3) provide better explanation of our proposed mechanism in light of the HR proficiency/deficiency of the models. These points are now integrated to the Abstract (Lines: 52-54, 61), Introduction (Lines: 110, 112, 122, 123), Results (Lines: 142, 144-152, 158-159, 163-167, 250, 318, 326, 331-336, 340, 351-355, 364, 401, 411-412, 451) and Discussion (Lines: 461-463, 476, 477, 479-480, 485, 492, 505) sections. We also updated our working model as shown below and in new Fig. 9, showing that SOC therapies in sensitive cells or their combination with PDE4D, EGFR or PARP1 inhibitors in SOC resistant cells effectively induce DNA damage and toxic PARP1 trapping,

leading to G1 arrest and apoptosis irrespective of the *BRCA1/2* status. We hope that the reviewer now finds our revised manuscript suitable for publication.

Figure. Schematic summary of the proposed mechanism of SOC sensitivity/resistance and overcoming resistance via targeting EGFR/PDE4D/PARP1 axis. 9a In SOC sensitive cells, SOC therapies downregulate PDE4D which is in a feedforward loop with ER, resulting in cAMP/PKA activation, causing mitochondrial ROS generation, DNA damage and PARP1 trapping. SOC therapy also downregulates BRCA1/2 and RAD51 levels, causing a functional BRCAness, i.e., HR deficiency that altogether leads to transcriptional blockage and cell death. In SOC resistant cells, PDE4D is upregulated via EGFR/c-Jun, blocking cAMP/PKA activation and thus leading to SOC resistance. Dashed lines represent inhibited events. **9b** In SOC resistant cells, combining SOC with inhibitors of PDE4D, EGFR or PARP leads to DNA damage, PARP1 trapping, transcriptional repression, and cell death irrespective of the *BRCA1/2* status.

References

1. Chen, Y., *et al.* 14-3-3sigma Contributes to Radioresistance By Regulating DNA Repair and Cell Cycle via PARP1 and CHK2. *Mol Cancer Res* **15**, 418-428 (2017).
2. Ohba, S., Mukherjee, J., See, W.L. & Pieper, R.O. Mutant IDH1-driven cellular transformation increases RAD51-mediated homologous recombination and temozolomide resistance. *Cancer Res* **74**, 4836-4844 (2014).
3. Ghandi, M., *et al.* Next-generation characterization of the Cancer Cell Line Encyclopedia. *Nature* **569**, 503-508 (2019).
4. Gu, Y., *et al.* BRCA1-deficient breast cancer cell lines are resistant to MEK inhibitors and show distinct sensitivities to 6-thioguanine. *Sci Rep* **6**, 28217 (2016).
5. Montaudon, E., *et al.* PLK1 inhibition exhibits strong anti-tumoral activity in CCND1-driven breast cancer metastases with acquired palbociclib resistance. *Nat Commun* **11**, 4053 (2020).
6. Johnson, N., *et al.* Stabilization of mutant BRCA1 protein confers PARP inhibitor and platinum resistance. *Proc Natl Acad Sci U S A* **110**, 17041-17046 (2013).
7. Aziz, D., *et al.* Synergistic targeting of BRCA1 mutated breast cancers with PARP and CDK2 inhibition. *NPJ Breast Cancer* **7**, 111 (2021).
8. Osoegawa, A., Gills, J.J., Kawabata, S. & Dennis, P.A. Rapamycin sensitizes cancer cells to growth inhibition by the PARP inhibitor olaparib. *Oncotarget* **8**, 87044-87053 (2017).
9. Sun, K., *et al.* A comparative pharmacokinetic study of PARP inhibitors demonstrates favorable properties for niraparib efficacy in preclinical tumor models. *Oncotarget* **9**, 37080-37096 (2018).
10. Wang, L.M., *et al.* Thioparib inhibits homologous recombination repair, activates the type I IFN response, and overcomes olaparib resistance. *EMBO Mol Med* **15**, e16235 (2023).
11. Spillman, M.A. & Bowcock, A.M. BRCA1 and BRCA2 mRNA levels are coordinately elevated in human breast cancer cells in response to estrogen. *Oncogene* **13**, 1639-1645 (1996).

Reviewer comments, third round

Reviewer #1 (Remarks to the Author):

Thank you for adding the extra functional information, I think this helps bring some clarity to the models. There are still some surprising aspects to these, but I think if fully described the data can be included. The missense mutations in the ZR-75-30 line and MDA-MB-361 lines are of unknown clinical significance (see BRCA Exchange database) and this should be noted - it is not clear if these are the cause of the HR deficiency observed. Are these observed at high allele frequencies (indicating loss of the wild type allele as typical for a pathogenic mutation)?

For MDA-MB-436 the mutation described is different from the published and well-characterised splice donor mutation in this cell line (c.5386+1G>A - see e.g. Elstrodt et al Cancer Res 2006). Did the authors genotype this themselves or otherwise confirm cell line identity in their own stocks, or are they relying on public sequencing data? I apologise for laboring this point but there is potential to create confusion in the literature here so it is important to get it right.

Olaparib concentrations in Supp Fig 10 and 11k still seem extremely high compared to published cell line sensitivity for MDA-MB-436 (see e.g. Johnson et al PNAS 2013) but perhaps this is a technical issue. I think the RAD51 and HR reporter assays provided are sufficient to show the HR deficient phenotypes however.

Answers to the comments of the Reviewer #1 for the manuscript NCOMMS-23-08866B

Thank you for adding the extra functional information, I think this helps bring some clarity to the models. There are still some surprising aspects to these, but I think if fully described the data can be included. The missense mutations in the ZR-75-30 line and MDA-MB-361 lines are of unknown clinical significance (see BRCA Exchange database) and this should be noted - it is not clear if these are the cause of the HR deficiency observed. Are these observed at high allele frequencies (indicating loss of the wild type allele as typical for a pathogenic mutation)?

We thank the Reviewer for finding the extra functional data provided helpful to bring more clarity to the models. As the reviewer pointed out, the missense mutations in ZR-75-30 and MDA-MB-361 cell lines have clinically unknown significance and we now noted this in the manuscript (Page 13, Lines 322, 323). Regarding the question of the reviewer on whether the mutations are observed at high allele frequencies, we indeed found out that both the MDA-MB-361 and ZR-75-30 cell lines have allelic loss for both the *BRCA1* and *BRCA2* genes¹⁻³, and the HBCx-118 PDX model has allelic loss of *BRCA2*⁴, potentially causing HR deficiency as observed. We have now noted this in the manuscript (Pages 13, 14, Lines 328, 329, 342). This is also the case for the MDA-MB-436 cell line^{2,3}. We now updated the **Supplementary Figures S8c, S8f, S8i and S11b** by including this information.

S8c			S8f		
ZR-75-30	BRCA1	BRCA2	MDA-MB-361	BRCA1	BRCA2
Mutation	None	c.6966G>T	Mutation	None	c.4970A>G
Type of mutation	None	Missense	Type of mutation	None	Missense
Allelic Loss	Yes	Yes	Allelic Loss	Yes	Yes
HR activity	Impaired		HR activity	Impaired	

S8i			S11b		
MDA-MB-436	BRCA1	BRCA2	HBCx-118	BRCA1	BRCA2
Mutation	c.5396+1G>A	None	Mutation	None	c.1257del
Type of mutation	Splicing mutation	None	Type of mutation	None	frameshift
Allelic Loss	Yes	Yes	Allelic Loss	None	Yes
HR activity	Impaired		HR activity	Impaired	

Figure. The *BRCA1/2* mutations, type of mutation, allelic changes, and homologous recombination (HR) activity in the cell models used. S8c, S8f, S8i, S11b. Tables of the *BRCA1* and *BRCA2* mutations, mutation types, the allelic loss of *BRCA1* or *BRCA2* and the HR activity in ZR-75-30 (S8c), MDA-MB-361 (S8f), MDA-MB-436 (S8i) and HBCx-118 (S11b) cells.

For MDA-MB-436 the mutation described is different from the published and well-characterised splice donor mutation in this cell line (c.5386+1G>A - see e.g. Elstrodt et al Cancer Res 2006). Did the authors genotype this themselves or otherwise confirm cell line identity in their own stocks, or are they relying on public sequencing data? I apologise for laboring this point but there is potential to create confusion in the literature here so it is important to get it right.

The splice donor mutation that the reviewer has pointed out for MDA-MB-436 is indeed the same as the one we provided in previous revision; however, we happened to provide the predicted amino acid change (p.Glu1731del28) instead of the genomic change (c.5396+1G>A) based on Table 2 of the same reference (Elstrodt et al, 2006²) (please note that it should be 5396, not 5386). To prevent any confusion, we now updated the **Supplementary Figures S8c, S8f, S8i** and **S11b** by providing the nucleotide change in *BRCA1* or *BRCA2* genes in all cell lines and PDX cells as shown above.

Olaparib concentrations in Supp Fig 10 and 11k still seem extremely high compared to published cell line sensitivity for MDA-MB-436 (see e.g. Johnson et al PNAS 2013) but perhaps this is a technical issue. I think the RAD51 and HR reporter assays provided are sufficient to show the HR deficient phenotypes however.

We fully agree with the Reviewer that the doses we showed in **Supplementary Fig. S10** (50 and 75 μM) are higher compared to published cell line sensitivity for MDA-MB-436. Some reported IC50s in publications are 6.4 μM (please see ref⁵), 1.25 μM (please see ref⁶) and 10.2 μM (please see ref⁷) for olaparib for MDA-MB-436 cells which are indeed lower than the doses we used. However, our olaparib IC50 in MDA-MB-436 cell line (1.8 μM as shown below) is also as low as the reported IC50s and it is clear from the dose response curves shown below (new **Supplementary Fig. S10a-c**) that the olaparib response of ZR-75-30 cells is in between MDA-MB-436 and T47D. In the previous revision, we had chosen those two doses (50 and 75 μM) based on the IC50 of ZR-75-30 cell line. To avoid confusion, we have now updated the figure with the dose response curves as shown below and also noted in the main text (Page 14, Lines 332, 333). Furthermore, the assay type performed in Johnson et al⁸ and our assay were different. While they did colony formation assay with 2 weeks treatment, we performed the standard cell viability assay with Sulforhodamine B (SRB)⁹ with 3 days treatment. These could be the reason of difference between the nanomolar range IC50s in Johnson et al, and IC50s we obtained and the others reported (around 1-10 μM^{5-7}). Overall, as the Reviewer appreciates, we characterized all cell lines with two different methods (RAD51 foci formation and the HR reporter assay) to show the HR deficiency, showing the validity of our models.

Figure. Olaparib response of the *BRCA1/2*-wt versus *BRCA1/2*-mut cell lines. S10a-c Percentage cell viability in MDA-MB-436 (*BRCA1*-mut) (S10a), ZR-75-30 (*BRCA2*-mut) (S10b) and T47D (*BRCA1/2*-wt) (S10c) cells upon treatment with olaparib for 3 days (n=3, 4). Error bars correspond to mean values \pm SD.

References

- 1 Barretina, J. *et al.* The Cancer Cell Line Encyclopedia enables predictive modelling of anticancer drug sensitivity. *Nature* **483**, 603-607, doi:10.1038/nature11003 (2012).
- 2 Elstrodt, F. *et al.* BRCA1 mutation analysis of 41 human breast cancer cell lines reveals three new deleterious mutants. *Cancer Res* **66**, 41-45, doi:10.1158/0008-5472.CAN-05-2853 (2006).
- 3 Harkes, I. C. *et al.* Allelotype of 28 human breast cancer cell lines and xenografts. *Br J Cancer* **89**, 2289-2292, doi:10.1038/sj.bjc.6601448 (2003).
- 4 Montaudon, E. *et al.* PLK1 inhibition exhibits strong anti-tumoral activity in CCND1-driven breast cancer metastases with acquired palbociclib resistance. *Nat Commun* **11**, 4053, doi:10.1038/s41467-020-17697-1 (2020).
- 5 Li, Y. *et al.* BKM120 sensitizes BRCA-proficient triple negative breast cancer cells to olaparib through regulating FOXM1 and Exo1 expression. *Sci Rep* **11**, 4774, doi:10.1038/s41598-021-82990-y (2021).
- 6 Moustafa, D., Elwahed, M. R. A., Elsaid, H. H. & Parvin, J. D. Modulation of Early Mitotic Inhibitor 1 (EMI1) depletion on the sensitivity of PARP inhibitors in BRCA1 mutated triple-negative breast cancer cells. *PLoS One* **16**, e0235025, doi:10.1371/journal.pone.0235025 (2021).
- 7 Malka, M. M., Eberle, J., Niedermayer, K., Zlotos, D. P. & Wiesmuller, L. Dual PARP and RAD51 Inhibitory Drug Conjugates Show Synergistic and Selective Effects on Breast Cancer Cells. *Biomolecules* **11**, doi:10.3390/biom11070981 (2021).
- 8 Johnson, N. *et al.* Stabilization of mutant BRCA1 protein confers PARP inhibitor and platinum resistance. *Proc Natl Acad Sci U S A* **110**, 17041-17046, doi:10.1073/pnas.1305170110 (2013).
- 9 Vichai, V. & Kirtikara, K. Sulforhodamine B colorimetric assay for cytotoxicity screening. *Nat Protoc* **1**, 1112-1116, doi:10.1038/nprot.2006.179 (2006).